# Iterative Methods via Locally Evolving Set Process

**Baojian Zhou**[1,2] *    **Yifan Sun**[3]    **Reza Babanezhad Harikandeh**[4]    **Xingzhi Guo**[3]

**Deqing Yang**[1,2]                **Yanghua Xiao**[2]

[1] the School of Data Science, Fudan University,
[2] Shanghai Key Laboratory of Data Science, School of Computer Science, Fudan University
[3] Department of Computer Science, Stony Brook University, [4] Samsung SAIT AI Lab.

## Abstract

Given the damping factor $\alpha$ and precision tolerance $\epsilon$, Andersen et al. [2] introduced Approximate Personalized PageRank (APPR), the *de facto local method* for approximating the PPR vector, with runtime bounded by $\Theta(1/(\alpha\epsilon))$ independent of the graph size. Recently, Fountoulakis & Yang [12] asked whether faster local algorithms could be developed using $\tilde{\mathcal{O}}(1/(\sqrt{\alpha}\epsilon))$ operations. By noticing that APPR is a local variant of Gauss-Seidel, this paper explores the question of *whether standard iterative solvers can be effectively localized*. We propose to use the *locally evolving set process*, a novel framework to characterize the algorithm locality, and demonstrate that many standard solvers can be effectively localized. Let $\overline{\text{vol}}(\mathcal{S}_t)$ and $\overline{\gamma}_t$ be the running average of volume and the residual ratio of active nodes $\mathcal{S}_t$ during the process. We show $\overline{\text{vol}}(\mathcal{S}_t)/\overline{\gamma}_t \leq 1/\epsilon$ and prove APPR admits a new runtime bound $\tilde{\mathcal{O}}(\overline{\text{vol}}(\mathcal{S}_t)/(\alpha\overline{\gamma}_t))$ mirroring the actual performance. Furthermore, when the geometric mean of residual reduction is $\Theta(\sqrt{\alpha})$, then there exists $c \in (0, 2)$ such that the local Chebyshev method has runtime $\tilde{\mathcal{O}}(\overline{\text{vol}}(\mathcal{S}_t)/(\sqrt{\alpha}(2-c)))$ without the monotonicity assumption. Numerical results confirm the efficiency of this novel framework and show up to a hundredfold speedup over corresponding standard solvers on real-world graphs.

## 1 Introduction

Personalized PageRank (PPR) vectors are key tools for graph problems such as clustering [2, 3, 30, 36, 54, 57], diffusion [10, 14, 15, 29], random walks [25, 32, 44], neural net training [7, 27, 20, 21], and many others [17, 48]. The Approximate PPR (APPR) [2] and its many variants [6, 9, 13, 37] efficiently approximate PPR vectors by exploring the neighbors of a specific node at each time, only requiring access to a tiny part of the graph – hence the number of operations needed is independent of graph size. These local solvers are well-suited for large-scale graphs in modern graph data analysis. Specifically, let $\boldsymbol{A}$ and $\boldsymbol{D}$ be the adjacency and degree matrices of a graph $\mathcal{G}$, respectively. Given a source node $s$ and the damping factor $\alpha \in (0, 1)$, this paper studies local solvers for the linear system

$$\left(\boldsymbol{I} - (1 - \alpha)\left(\boldsymbol{I} + \boldsymbol{A}\boldsymbol{D}^{-1}\right)/2\right)\boldsymbol{\pi} = \alpha\boldsymbol{e}_s, \tag{1}$$

where $\boldsymbol{e}_s$ is the standard basis of $s$ and $\boldsymbol{\pi}$ is the PPR vector [2, 12, 37]. Given the error tolerance $\epsilon$, a local solver needs to find $\hat{\boldsymbol{\pi}}$ such that $\|\boldsymbol{D}^{-1}(\hat{\boldsymbol{\pi}} - \boldsymbol{\pi})\|_\infty \leq \epsilon$ without accessing the entire graph $\mathcal{G}$.[2]

---

*Corresponding to: Baojian Zhou, bjzhou@fudan.edu.cn

[2]Local methods for solving Equ. (1) can be naturally extended to other linear systems defined on $\mathcal{G}$.

38th Conference on Neural Information Processing Systems (NeurIPS 2024).

Andersen et al. [2] proposed the local APPR algorithm, which pushes large residuals to neighboring nodes until all residuals are small. Its runtime is upper bounded by $\Theta(1/(\alpha\epsilon))$ independent of graph size. Based on a variational characterization of Equ. (1), Fountoulakis et al. [13] reformulated the problem as optimizing a quadratic objective plus $\ell_1$-regularization and later asked [12] whether there exists a local solver with runtime $\tilde{\mathcal{O}}(1/(\sqrt{\alpha}\epsilon))$. This corresponds to an accelerated rate since $\alpha$ is the strongly convex parameter. Recently, Martínez-Rubio et al. [37] provided a method based on a nested subspace pursuit strategy, and the corresponding iteration complexity is bounded by $\tilde{\mathcal{O}}(|\mathcal{S}^*|/\sqrt{\alpha})$ where $\mathcal{S}^*$ is the support of the optimal solution. This bound deteriorates to $\tilde{\mathcal{O}}(n/\sqrt{\alpha})$ when the solution is dense, with $n$ representing the number of nodes in $\mathcal{G}$, which could be less favorable than that of standard solvers under similar conditions. Moreover, the nested computational structure provides a constant factor overhead, which could be significant in practice.

The bound analysis of the above local methods critically depends on the monotonicity properties of the designed algorithms. These requirements may hinder the development of simpler and faster local linear solvers that lack such monotonicity properties. Specifically, the runtime analysis of APPR relies on the non-negativity and decreasing monotonicity of residuals. Conversely, the runtime bounds developed in Fountoulakis et al. [13] and Martínez-Rubio et al. [37] depend on the monotonicity of variable updates, ensuring that the sparsity of intermediate variables increases monotonically.

**Our contributions.** Based on a refined analysis of APPR, our starting point is to demonstrate that APPR is a local variant of Gauss-Seidel Successive Overrelaxation (GS-SOR) that can be treated as an evolving set process.[3] This insight leads us to explore whether standard solvers can be effectively localized to solve Equ. (1). To develop faster local methods with improved local bounds, we propose a novel *locally evolving set process* framework inspired by its stochastic counterpart [38]. This framework enables the development of faster local methods and circumvents the monotonicity requirement barrier in runtime complexity analysis in the existing literature. For example, our analysis of the local Chebyshev method does not depend on the monotonicity of residual or the active node sets processed. Specifically,

- As a core tool, we propose an algorithm framework based on the *locally evolving set process*. We show that APPR is a local variant of GS-SOR using this process. This framework is powerful enough to facilitate the development of new local solvers. Specifically, standard gradient descent (GD) can be effectively localized for solving this problem and admits $\Theta(1/(\alpha\epsilon))$ runtime bound.

- This local evolving set process provides a novel way to characterize the algorithm locality; hence, new runtime bounds can be derived. Let $\overline{\mathrm{vol}}(\mathcal{S}_t)$ and $\overline{\gamma}_t$ be the running average of volume and the residual ratio of active nodes $\mathcal{S}_t$ during the process; we prove the ratio $\overline{\mathrm{vol}}(\mathcal{S}_t)/\overline{\gamma}_t$ serving as a lower bound of $1/\epsilon$. We further show both APPR and local GD have $\tilde{\mathcal{O}}(\overline{\mathrm{vol}}(\mathcal{S}_t)/(\alpha\overline{\gamma}_t))$ runtime bound mirroring the actual performance of these two methods.

- Using our framework, we show there exists $c \in (0, 2)$ such that both the localized Chebyshev and Heavy-Ball methods admit runtime bound $\tilde{\mathcal{O}}(\overline{\mathrm{vol}}(\mathcal{S}_t)/(\sqrt{\alpha}(2 - c)))$ with the assumption that the geometric mean of active ratio factors is $\Theta(\sqrt{\alpha})$. Importantly, our analysis does not require any monotonicity property. The technical novelty is that we effectively characterize residuals of these two methods by using second-order difference equations with parameterized coefficients.

- We demonstrate, over 17 large graphs, that these localized methods can significantly accelerate their standard counterparts by a large margin. Furthermore, our proposed LOCSOR, LOCCH, and LOCHB are significantly faster than APPR and $\ell_1$-based solvers on two huge-scale graphs.

**Paper structure.** We begin by clarifying notations and reviewing APPR in Sec. 2. Sec. 3 introduces the locally evolving set process. Sec. 4 presents localized Chebyshev and Heavy-Ball methods along with our novel techniques. We discuss open questions in Sec. 5. Experiments and conclusions are covered in Sec. 6 and 7, respectively. *Detailed related works and all missing proofs are included in the Appendix. Our code is available at* `https://github.com/baojian/LocalCH`.

## 2 Notations and Preliminaries

**Notations.** We consider an undirected simple graph $\mathcal{G}(\mathcal{V}, \mathcal{E})$ where $\mathcal{V} = \{1, 2, \ldots, n\}$ and $\mathcal{E} \subseteq \mathcal{V} \times \mathcal{V}$ with $|\mathcal{E}| = m$ are the node and edge sets, respectively. The set of neighbors of $v$ is denoted as

---

[3]The local variant of GS-SOR is defined in Appendix B.1

$\mathcal{N}(v) \subseteq \mathcal{V}$. The adjacency matrix $\boldsymbol{A}$ of $\mathcal{G}$ assigns unit weight $a_{u,v} = 1$ if $(u, v) \in \mathcal{E}$ and 0 otherwise. The $v$-th entry of the degree matrix $\boldsymbol{D}$ is $d_v = |\mathcal{N}(v)|$. Given $\mathcal{S} \subseteq \mathcal{V}$, we define the volume of $\mathcal{S}$ as $\text{vol}(\mathcal{S}) \triangleq \sum_{v \in \mathcal{S}} d_v$. The support of $\boldsymbol{x} \in \mathbb{R}^n$ is the set of nonzero indices $\text{supp}(\boldsymbol{x}) \triangleq \{v : x_v \neq 0, v \in \mathcal{V}\}$. The eigendecomposition of $\boldsymbol{D}^{-1/2} \boldsymbol{A} \boldsymbol{D}^{-1/2} = \boldsymbol{V} \boldsymbol{\Lambda} \boldsymbol{V}^\top$ where each column of $\boldsymbol{V}$ is an eigenvector and $\boldsymbol{\Lambda} = \text{diag}(\lambda_1, \lambda_2, \ldots, \lambda_n)$ with $1 = \lambda_1 \geq \lambda_2 \geq \cdots \geq \lambda_n \geq -1$.

## 2.1 Revisiting Anderson's APPR and its local runtime bound

We use $(\boldsymbol{p}, \boldsymbol{z})$ for solving Equ. (1) while use $(\boldsymbol{x}, \boldsymbol{r})$ for solving Equ. (3) or Equ. (4). With the initial setting $\boldsymbol{p} \leftarrow \boldsymbol{0}, \boldsymbol{z} \leftarrow \boldsymbol{e}_s$, APPR obtains a local estimate of $\boldsymbol{\pi}$ denoted as $\boldsymbol{p}$ by using a sequence of PUSH operations defined as

$$\boxed{\text{APPR}(\alpha, \epsilon, s, \mathcal{G}) : \quad \textbf{Repeat } (\boldsymbol{p}, \boldsymbol{z}) \leftarrow \text{PUSH}(u, \alpha, \boldsymbol{p}, \boldsymbol{z}) \textbf{ Until } \forall v, z_v < \epsilon d_v; \textbf{ Return } \boldsymbol{p}.} \quad (2)$$

---

**Algo. 1** PUSH$(u, \alpha, \boldsymbol{p}, \boldsymbol{z})$

1: $\nu = z_u$
2: $p_u \leftarrow p_u + \alpha \cdot \nu$
3: $z_u \leftarrow (1 - \alpha) \cdot \nu / 2$
4: **for** $v \in \mathcal{N}(u)$ **do**
5: $\quad z_v \leftarrow z_v + \frac{(1-\alpha)\nu}{2d_u}$
6: **Return** $(\boldsymbol{p}, \boldsymbol{z})$

---

At each repeat step $(\boldsymbol{p}, \boldsymbol{z}) \leftarrow \text{PUSH}(u, \alpha, \boldsymbol{p}, \boldsymbol{z})$, it synchronously updates both $\boldsymbol{p}$ and residual $\boldsymbol{z}$ whenever there exists an *active* node $u \in \mathcal{V}$ (a node with a large residual, i.e., $z_u \geq \epsilon d_u$). Specifically, for each active $u$, it updates $p_u$, $z_u$, and $z_v$ for $v \in \mathcal{N}(u)$ by using a PUSH operator illustrated on the left. It stops when no active nodes are left. APPR can be implemented locally so that the runtime is independent of $\mathcal{G}$. In particular, $\text{vol}(\text{supp}(\boldsymbol{p}))$ is locally bounded, demonstrating the sparsity effect. We restate the existing main results as follows.

**Lemma 2.1** (Runtime bound of APPR [2]). *Given $\alpha \in (0, 1)$ and the precision $\epsilon \leq 1/d_s$ for node $s \in \mathcal{V}$ with $\boldsymbol{p} \leftarrow \boldsymbol{0}, \boldsymbol{z} \leftarrow \boldsymbol{e}_s$ at the initial, $\text{APPR}(\alpha, \epsilon, s, \mathcal{G})$ defined in (2) returns an estimate $\boldsymbol{p}$ of $\boldsymbol{\pi}$. There exists a real implementation of (2) (e.g., Algo. 2) such that the runtime $\mathcal{T}_{\text{APPR}}$ satisfies*

$$\mathcal{T}_{\text{APPR}} \leq \Theta\left(1/(\alpha\epsilon)\right).$$

*Furthermore, the estimate $\hat{\boldsymbol{\pi}} := \boldsymbol{p}$ satisfies $\|\boldsymbol{D}^{-1}(\hat{\boldsymbol{\pi}} - \boldsymbol{\pi})\|_\infty \leq \epsilon$ and $\text{vol}(\text{supp}(\hat{\boldsymbol{\pi}})) \leq 2/((1-\alpha)\epsilon)$.*

The main argument for proving Lemma 2.1 is critically based on: 1) $\boldsymbol{z} \geq \boldsymbol{0}$ and $\|\boldsymbol{z}\|_1$ decreases during the updates; 2) for each active $u$, $z_u \geq \epsilon d_u$, implying that $\|\boldsymbol{z}\|_1$ is decreased by at least $\alpha\epsilon d_u$, consecutively leading to $\sum_u d_u \leq 1/(\alpha\epsilon)$.

## 2.2 Problem reformulation

To approximate the PPR vector $\boldsymbol{\pi}$, the original linear system in Equ. (1) can be reformulated as an equivalent symmetric version defined as

$$\boldsymbol{Q}\boldsymbol{x} = \boldsymbol{b}, \quad \text{with } \boldsymbol{Q} \triangleq \boldsymbol{I} - \frac{1-\alpha}{1+\alpha}\boldsymbol{D}^{-1/2}\boldsymbol{A}\boldsymbol{D}^{-1/2} \text{ and } \boldsymbol{b} \triangleq \frac{2\alpha}{(1+\alpha)}\boldsymbol{D}^{-1/2}\boldsymbol{e}_s, \quad (3)$$

where again $\boldsymbol{e}_s$ is the standard basis of $s$, and $\boldsymbol{Q}$ is a symmetric positive-definite $M$-matrix with all eigenvalues in $[\frac{2\alpha}{1+\alpha}, \frac{2}{1+\alpha}]$. To solve Equ. (3) is equivalent to solving a quadratic problem

$$\boldsymbol{x}^* = \arg\min_{\boldsymbol{x} \in \mathbb{R}^n} \left\{ f(\boldsymbol{x}) \triangleq \frac{1}{2}\boldsymbol{x}^\top \boldsymbol{Q}\boldsymbol{x} - \boldsymbol{x}^\top \boldsymbol{b} \right\}, \quad (4)$$

where $f$ is strongly convex with condition number $1/\alpha$. Indeed, Equ. (3) is a symmetrized version of Equ. (1) and has a unique solution $\boldsymbol{x}^* = \boldsymbol{Q}^{-1}\boldsymbol{b}$. The PPR vector $\boldsymbol{\pi}$ can be recovered from $\boldsymbol{x}^*$ by $\boldsymbol{\pi} = \boldsymbol{D}^{1/2}\boldsymbol{x}^*$. It is convenient to denote estimate of $\boldsymbol{\pi}$ as $\boldsymbol{\pi}^{(t)} \triangleq \boldsymbol{D}^{1/2}\boldsymbol{x}^{(t)}$. Given $\boldsymbol{x}^{(t)}$, we define the residual $\boldsymbol{r}^{(t)} \triangleq \boldsymbol{b} - \boldsymbol{Q}\boldsymbol{x}^{(t)}$. If $\boldsymbol{x}^{(t)}$ is returned by a local solver for solving either Equ. (3) or Equ. (4), we then equivalently require $\|\boldsymbol{D}^{-1/2}(\boldsymbol{x}^{(t)} - \boldsymbol{x}^*)\|_\infty \leq \epsilon$. Hence, it is enough to have a stop condition $\|\boldsymbol{D}^{-1/2}\boldsymbol{r}^{(t)}\|_\infty \leq 2\alpha\epsilon/(1 + \alpha)$ for local solvers of Equ. (3) and Equ. (4).[4]

Fountoulakis et al. [13] demonstrated that APPR is equivalent to a coordinate descent solver for minimizing $f$ in Equ. (4) and introduced an ISTA-style solver by minimizing $f(\boldsymbol{x}) + \epsilon\alpha\|\boldsymbol{D}^{1/2}\boldsymbol{x}\|_1$, which provides a method with runtime bound $\tilde{\mathcal{O}}(1/(\epsilon\alpha))$ for achieving the same estimation guarantee of APPR. On one hand, one may note that the runtime bound $\Theta(1/(\alpha\epsilon))$ provided in Lemma 2.1 becomes less valuable when $\epsilon \leq 1/m$; on the other hand, all previous local variants [6, 9, 13, 37]

---

[4]See a justification in Appendix B.

of APPR are critically based on some monotonicity property. This limitation could impede the development of faster local methods that might violate the monotonicity assumption. The following two sections present the techniques and tools to address these challenges.

## 3 Local Methods via Evolving Set Process

Our investigation begins with the *locally evolving set process*, as inspired by the stochastic counterpart [38]. The process reveals that APPR is essentially a local variant of GS-SOR. We then show how to use this process to build faster local solvers based on GS-SOR. We further develop a *local parallelizable* gradient descent with runtime $\Theta(1/(\alpha\epsilon))$.

### 3.1 Locally evolving set process

Given $\alpha, \epsilon, s$, and $\mathcal{G}$, a local solver for Equ. (3) keeps track of an *active set* $\mathcal{S}_t \subset \mathcal{V}$ at each iteration $t$. That is, only nodes in $\mathcal{S}_t$ are used to update $\boldsymbol{x}$ or $\boldsymbol{r}$. The next set $\mathcal{S}_{t+1}$ is determined by current $\mathcal{S}_t$ and an associated local solver $\mathcal{A}$. We define this process as the following local evolving set system.

**Definition 3.1** (Locally evolving set process). Given a parameter configuration $\theta \triangleq (\alpha, \epsilon, s, \mathcal{G})$, and a local iterative method $\mathcal{A}$, the locally evolving set process generates a sequence of $(\mathcal{S}_t, \boldsymbol{x}^{(t)}, \boldsymbol{r}^{(t)})$ representing as the following dynamic system

$$\left(\mathcal{S}_{t+1}, \boldsymbol{x}^{(t+1)}, \boldsymbol{r}^{(t+1)}\right) = \boldsymbol{\Phi}_\theta\left(\mathcal{S}_t, \boldsymbol{x}^{(t)}, \boldsymbol{r}^{(t)}, \mathcal{A}\right), \quad \forall t \geq 0, \tag{5}$$

where $\mathcal{S}_{t+1} \subseteq \mathcal{S}_t \cup (\cup_{u \in \mathcal{S}_t} \mathcal{N}(u))$ and we denote the active set $\mathcal{S}_t = \{u_1, u_2, \ldots, u_{|\mathcal{S}_t|}\}$. The set $\mathcal{S}_t$ is maintained via a queue data structure. We say this process *converges* when the last set $\mathcal{S}_T = \emptyset$ if there exists such $T$; the generated sequence of active nodes are

$$(\mathcal{S}_0, \boldsymbol{x}^{(0)}, \boldsymbol{r}^{(0)}) \rightarrow (\mathcal{S}_1, \boldsymbol{x}^{(1)}, \boldsymbol{r}^{(1)}) \rightarrow (\mathcal{S}_2, \boldsymbol{x}^{(2)}, \boldsymbol{r}^{(2)}) \rightarrow \ldots \rightarrow (\mathcal{S}_T = \emptyset, \boldsymbol{x}^{(T)}, \boldsymbol{r}^{(T)}).$$

The runtime of the local solver, $\mathcal{A}$ for this whole local process, is then defined as [5]

$$\mathcal{T}_\mathcal{A} \triangleq \sum_{t=0}^{T-1} \mathrm{vol}(\mathcal{S}_t).$$

The framework of this set process provides a new way to design local methods. Furthermore, it helps to analyze the convergence and runtime bound of local solvers by characterizing the sequences $\{\mathrm{vol}(\mathcal{S}_t)\}$, and $\{\|\boldsymbol{r}^{(t)}\|\}$ generated by $\boldsymbol{\Phi}_\theta$. To analyze a new runtime bound, for $T \geq 1$, we define the average of the volume of active node sets $\{\mathrm{vol}(\mathcal{S}_t)\}$ and active ratio sequence $\{\gamma_t\}$ as

$$\overline{\mathrm{vol}}(\mathcal{S}_T) \triangleq \frac{1}{T} \sum_{t=0}^{T-1} \mathrm{vol}(\mathcal{S}_t), \quad \overline{\gamma}_T \triangleq \frac{1}{T} \sum_{t=0}^{T-1} \left\{\gamma_t \triangleq \frac{\sum_{i=1}^{|\mathcal{S}_t|} |\sqrt{d_{u_i}} r_{u_i}^{(t+\Delta_i)}|}{\|\boldsymbol{D}^{1/2}\boldsymbol{r}^{(t)}\|_1}\right\}, \tag{6}$$

where $\Delta_i$ is a smaller time magnitude. We define $\Delta_i = (i-1)/|\mathcal{S}_t|$ for the analysis of APPR and LocSOR while $\Delta_i = 0$ for LocGD in our later analysis. In the rest, we denote $\mathcal{I}_T = \mathrm{supp}(\boldsymbol{r}^{(T)})$.

These two metrics $\overline{\mathrm{vol}}(\mathcal{S}_T)$ and $\overline{\gamma}_T$ characterize the locality of local methods. To demonstrate this local process, Fig. 1 shows $\mathrm{vol}(\mathcal{S}_t)$ of APPR peaks at the early stage, and the active ratio decreases as the active volume

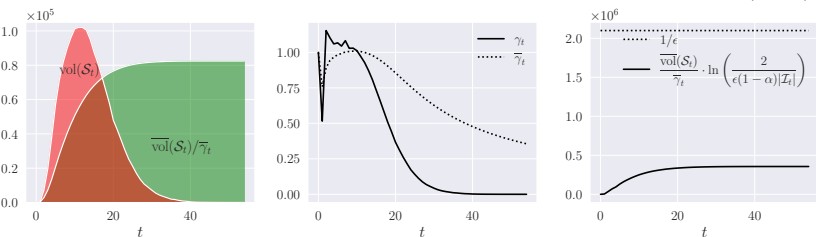

Figure 1: Runtime of APPR in the locally evolving set process on the *com-dblp* graph with $s = 0, \alpha = 0.1$, and $\epsilon = 1/m$. The red region of the left figure is $\mathcal{T}_{\mathrm{APPR}}$. The right two figures show active ratios and $\overline{\mathrm{vol}}(\mathcal{S}_T)/\overline{\gamma}_T \leq 1/\epsilon$.

diminishes. The quantity $\overline{\mathrm{vol}}(\mathcal{S}_T)/\overline{\gamma}_T$ is strictly smaller than $1/\epsilon$, indicating that it could serve as a better factor in the runtime analysis.

[5] In practice, $\mathcal{T}_\mathcal{A} := \sum_{t=0}^{T-1}(\mathrm{vol}(\mathcal{S}_t) + |\mathcal{S}_t|)$ where we ignore $|\mathcal{S}_t|$ for simplicity as $\mathrm{vol}(\mathcal{S}_t)$ dominates $|\mathcal{S}_t|$.

## 3.2 APPR via locally evolving set process

We first demonstrate how this locally evolving set process can represent APPR. For solving Equ. (1), the set $\mathcal{S}_0 = \{s\}$ and the queue-based of APPR (see Algo. 2 in Appendix A) naturally forms a sequence of active sets from $\mathcal{S}_0 = \{s\}$ to $\mathcal{S}_T = \emptyset$, hence converging. Active nodes $u$ in queue satisfy $z_u \geq \epsilon d_u$. To delineate successive iterations $\mathcal{S}_t$ and $\mathcal{S}_{t+1}$, one can insert $*$ at the beginning of $\mathcal{S}_t$. After processing $\mathcal{S}_t$, it serves as an indicator for the next iteration. The star $*$ is reinserted into the queue iteratively until the queue is empty. We use a slightly different notation for presenting tuple $(\mathcal{S}_t, \boldsymbol{p}^{(t)}, \boldsymbol{z}^{(t)})$ to consistent with Sec. 2.1 and write out such evolving process as follows

$$
\boxed{
\begin{aligned}
&\boldsymbol{\Phi}_\theta\left(\mathcal{S}_t, \boldsymbol{p}^{(t)}, \boldsymbol{z}^{(t)}, \mathcal{A} = \text{APPR}\right) : \textbf{ for } u_i \textbf{ in } \mathcal{S}_t := \{u_1, u_2, \ldots, u_{|\mathcal{S}_t|}\} \textbf{ do} \\
&\quad \boldsymbol{p}^{(t+\Delta_{i+1})} \leftarrow \boldsymbol{p}^{(t+\Delta_i)} + \alpha z_{u_i}^{(t+\Delta_i)} \boldsymbol{e}_{u_i}, \quad \Delta_i := (i-1)/|\mathcal{S}_t| \\
&\quad \boldsymbol{z}^{(t+\Delta_{i+1})} \leftarrow \boldsymbol{z}^{(t+\Delta_i)} - \frac{(1+\alpha)}{2} z_{u_i}^{(t+\Delta_i)} \boldsymbol{e}_{u_i} + \frac{(1-\alpha)}{2} z_{u_i}^{(t+\Delta_i)} \boldsymbol{A}\boldsymbol{D}^{-1} \boldsymbol{e}_{u_i}
\end{aligned}
}
\tag{7}
$$

The following lemma establishes the equivalence between APPR and the local variant of GS-SOR method (see Appendix B.1) and provides a new evolving-based bound.

**Lemma 3.2** (New local evolving-based bound for APPR). *Let $\boldsymbol{M} = \alpha^{-1}\left(\boldsymbol{I} - \frac{1-\alpha}{2}\left(\boldsymbol{I} + \boldsymbol{A}\boldsymbol{D}^{-1}\right)\right)$ and $\boldsymbol{s} = \boldsymbol{e}_s$. The linear system $\boldsymbol{M}\boldsymbol{\pi} = \boldsymbol{s}$ is equivalent to Equ. (1). Given $\boldsymbol{p}^{(0)} = \boldsymbol{0}$, $\boldsymbol{z}^{(0)} = \boldsymbol{e}_s$ with $\omega \in (0, 2)$, the local variant of GS-SOR (15) for $\boldsymbol{M}\boldsymbol{\pi} = \boldsymbol{s}$ can be formulated as*

$$
\boldsymbol{p}^{(t+\Delta_{i+1})} \leftarrow \boldsymbol{p}^{(t+\Delta_i)} + \frac{\omega z_{u_i}^{(t+\Delta_i)}}{M_{u_i u_i}} \boldsymbol{e}_{u_i}, \quad \boldsymbol{z}^{(t+\Delta_{i+1})} \leftarrow \boldsymbol{z}^{(t+\Delta_i)} - \frac{\omega z_{u_i}^{(t+\Delta_i)}}{M_{u_i u_i}} \boldsymbol{M} \boldsymbol{e}_{u_i},
$$

*where $u_i$ is an active node in $\mathcal{S}_t$ satisfying $z_{u_i} \geq \epsilon d_{u_i}$ and $\Delta_i = (i-1)/|\mathcal{S}_t|$. Furthermore, when $\omega = \frac{1+\alpha}{2}$, this method reduces to APPR given in (7), and there exists a real implementation (Aglo. 2) of APPR such that the runtime $\mathcal{T}_{\text{APPR}}$ is bounded, that is*

$$
\mathcal{T}_{\text{APPR}} \leq \frac{\overline{\text{vol}}(\mathcal{S}_T)}{\alpha \hat{\gamma}_T} \ln \frac{C_T}{\epsilon}, \text{ where } \frac{\overline{\text{vol}}(\mathcal{S}_T)}{\hat{\gamma}_T} \leq \frac{1}{\epsilon}, \ C_T = \frac{2}{(1-\alpha)|\mathcal{I}_T|}, \hat{\gamma}_T \triangleq \frac{1}{T} \sum_{t=0}^{T-1} \left\{ \frac{\sum_{i=1}^{|\mathcal{S}_t|} |z_{u_i}^{(t+\Delta_i)}|}{\|\boldsymbol{z}^{(t)}\|_1} \right\}.
$$

## 3.3 Faster local variant of GS-SOR

Lemma 3.2 points to the sub-optimality of APPR, as GS-SOR allows for a larger $\omega$. For solving Equ. (3), since APPR essentially serves as a local variant of GS-SOR, we can develop a faster local variant based SOR. To extend this method to solve Equ. (3), we propose a local GS-SOR based on an evolving set process, namely LocSOR, as the following

$$
\boxed{
\begin{aligned}
&\boldsymbol{\Phi}_\theta\left(\mathcal{S}_t, \boldsymbol{x}^{(t)}, \boldsymbol{r}^{(t)}, \mathcal{A} = \text{LocSOR}\right) : \textbf{ for } u_i \textbf{ in } \mathcal{S}_t := \{u_1, \ldots, u_{|\mathcal{S}_t|}\} \textbf{ and } \textbf{do} \\
&\quad \boldsymbol{x}^{(t+\Delta_{i+1})} \leftarrow \boldsymbol{x}^{(t+\Delta_i)} + \omega r_{u_i}^{(t+\Delta_i)} \boldsymbol{e}_{u_i}, \quad \Delta_i = (i-1)/|\mathcal{S}_t| \\
&\quad \boldsymbol{r}^{(t+\Delta_{i+1})} \leftarrow \boldsymbol{r}^{(t+\Delta_i)} - \omega r_{u_i}^{(t+\Delta_i)} \boldsymbol{e}_{u_i} + \frac{(1-\alpha)\omega}{1+\alpha} r_{u_i}^{(t+\Delta_i)} \boldsymbol{D}^{-1/2} \boldsymbol{A}\boldsymbol{D}^{-1/2} \boldsymbol{e}_{u_i}
\end{aligned}
}
\tag{8}
$$

When $\omega \in (0, 1]$, the residual $\boldsymbol{r}$ is still nonnegative and monotonically decreasing, we establish the convergence of LocSOR stated in the following theorem.

**Theorem 3.3** (Runtime bound of LocSOR ($\omega = 1$)). *Given the configuration $\theta = (\alpha, \epsilon, s, \mathcal{G})$ with $\alpha \in (0, 1)$ and $\epsilon \leq 1/d_s$ and let $\boldsymbol{r}^{(T)}$ and $\boldsymbol{x}^{(T)}$ be returned by LocSOR defined in (8) for solving Equ. (3). There exists a real implementation of (8) such that the runtime $\mathcal{T}_{\text{LocSOR}}$ is bounded by*

$$
\frac{1+\alpha}{2} \cdot \frac{\overline{\text{vol}}(\mathcal{S}_T)}{\alpha \overline{\gamma}_T} \left(1 - \frac{\|\boldsymbol{D}^{1/2}\boldsymbol{r}^{(T)}\|_1}{\|\boldsymbol{D}^{1/2}\boldsymbol{r}^{(0)}\|_1}\right) \leq \mathcal{T}_{\text{LocSOR}} \leq \frac{1+\alpha}{2} \cdot \min\left\{ \frac{1}{\alpha\epsilon}, \frac{\overline{\text{vol}}(\mathcal{S}_T)}{\alpha \overline{\gamma}_T} \ln \frac{C}{\epsilon} \right\}
$$

*where $\overline{\text{vol}}(S_T)$ and $\overline{\gamma}_T$ are defined in (6) and $C = \frac{1+\alpha}{(1-\alpha)|\mathcal{I}_T|}$ with $\mathcal{I}_T = \text{supp}(\boldsymbol{r}^{(T)})$. Furthermore, $\overline{\text{vol}}(\mathcal{S}_T)/\overline{\gamma}_T \leq 1/\epsilon$ and the local estimate $\hat{\boldsymbol{\pi}} := \boldsymbol{D}^{1/2}\boldsymbol{x}^{(T)}$ satisfies $\|\boldsymbol{D}^{-1}(\hat{\boldsymbol{\pi}} - \boldsymbol{\pi})\|_\infty \leq \epsilon$.*

Our new evolving bound $\tilde{\mathcal{O}}(\overline{\text{vol}}(\mathcal{S}_T)/(\alpha\overline{\gamma}_T))$ mirroring the actual performance of APPR and empirically much smaller than $\Theta(1/(\alpha\epsilon))$ as illustrated in Fig. 2. Our lower bounds are quite effective when $\epsilon$ is relatively large, while our upper bound is better than Anderson's when $\epsilon$ is small. When $\epsilon \ll \Theta(1/m)$, this new bound is superior to both $\mathcal{O}(1/(\alpha\epsilon))$ and $\tilde{\mathcal{O}}(1/(\sqrt{\alpha}\epsilon))$. This superiority is evident when compared to algorithms like ISTA or FISTA [5] to minimize the $\ell_1$-regularization of $f$ for obtaining an approximate solution of Equ. (3). Additionally, when

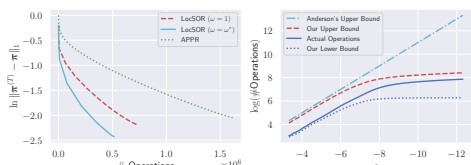

Figure 2: Comparison of runtime between APPR and LOCSOR (left) and runtime bounds (right) as a function of $\epsilon$. We used the same setting as in Fig. 1.

$\omega \in (1, 2)$ and recalling that $\boldsymbol{Q}$ is an $M$-matrix, the standard analysis of SOR shows that the spectral norm of the iteration matrix must be larger than $|\omega - 1|$. Hence, $0 < \omega < 2$ if and only if global SOR converges [55]. When $\omega^*$ is optimal (the point that the spectral radius of the iteration matrix is minimized), we have the following result.

**Corollary 3.4.** *Let* $\omega = \omega^* \triangleq 2/(1 + \sqrt{1 - (1-\alpha)^2/(1+\alpha)^2})$ *and* $\mathcal{S}_t = \mathcal{V}, \forall t \geq 0$ *during the updates, the global version of* LOCSOR *has the following convergence bound*

$$\|\boldsymbol{r}^{(t)}\|_2 \leq \frac{2}{(1+\alpha)\sqrt{d_s}} \left( \frac{1 - \sqrt{\alpha}}{1 + \sqrt{\alpha}} + \epsilon_t \right)^t,$$

*where* $\epsilon_t$ *are small positive numbers with* $\lim_{t\to\infty} \epsilon_t = 0$.

Asymptotically, when $\epsilon_t = o(\sqrt{\alpha})$, then the runtime of global LOCSOR is $\tilde{\mathcal{O}}(m/\sqrt{\alpha})$ where $\tilde{\mathcal{O}}$ hides $\log 1/\epsilon$. The main difficulty of analyzing the *optimal* local LOCSOR is that the nonnegativity and monotonicity of $\boldsymbol{r}^{(t)}$ do not hold. Instead, by using a parameterized second-order difference equation, we develop new techniques based on the Chebyshev method detailed in Sec. 4.

### 3.4 Parallelizable local gradient descent

One disadvantage of LOCSOR is its limited potential for parallelization. The standard GD $\boldsymbol{x}^{(t+1)} = \boldsymbol{x}^{(t)} - \boldsymbol{\nabla} f(\boldsymbol{x}^{(t)})$ (step size = 1), in contrast, is easy to parallelize across the coordinates of the update. Instead of updating $\boldsymbol{r}$ and $\boldsymbol{x}$ synchronously per-coordinate, we propose the following

$$\boxed{\boldsymbol{\Phi}_\theta\left(\mathcal{S}_t, \boldsymbol{x}^{(t)}, \boldsymbol{r}^{(t)}, \mathcal{A} = \text{LOCGD}\right): \boldsymbol{x}^{(t+1)} \leftarrow \boldsymbol{x}^{(t)} + \boldsymbol{r}^{(t)}_{\mathcal{S}_t}, \ \boldsymbol{r}^{(t+1)} \leftarrow \boldsymbol{r}^{(t)} - \boldsymbol{Q}\boldsymbol{r}^{(t)}_{\mathcal{S}_t}} \quad (9)$$

Every coordinate in $\mathcal{S}_t$ is updated in parallel at iteration $t$. Interestingly, LOCGD exhibits nonnegativity and monotonicity properties, and its runtime complexity is similar to that of LOCSOR, as stated in the following theorem (To remind, $\Delta_i = 0$ for $\overline{\gamma}_T$ of LOCGD in Equ. (6) ).

**Theorem 3.5** (Runtime bound of LOCGD). *Given the configuration* $\theta = (\alpha, \epsilon, s, \mathcal{G})$ *with* $\alpha \in (0, 1)$ *and* $\epsilon \leq 1/d_s$ *and let* $\boldsymbol{r}^{(T)}$ *and* $\boldsymbol{x}^{(T)}$ *be returned by* LOCGD *defined in* (9) *for solving Equ.* (4). *There exists a real implementation of* (9) *such that the runtime* $\mathcal{T}_{\text{LOCGD}}$ *is bounded by*

$$\frac{1+\alpha}{2} \cdot \frac{\overline{\text{vol}}(\mathcal{S}_T)}{\alpha\overline{\gamma}_T} \left( 1 - \frac{\|\boldsymbol{D}^{1/2}\boldsymbol{r}^{(T)}\|_1}{\|\boldsymbol{D}^{1/2}\boldsymbol{r}^{(0)}\|_1} \right) \leq \mathcal{T}_{\text{LOCGD}} \leq \frac{1+\alpha}{2} \cdot \min\left\{ \frac{1}{\alpha\epsilon}, \frac{\overline{\text{vol}}(\mathcal{S}_T)}{\alpha\overline{\gamma}_T} \ln\frac{C}{\epsilon} \right\},$$

*where* $C = (1+\alpha)/((1-\alpha)|\mathcal{I}_T|), \mathcal{I}_T = \text{supp}(\boldsymbol{r}^{(T)})$. *Furthermore,* $\overline{\text{vol}}(\mathcal{S}_T)/\overline{\gamma}_T \leq 1/\epsilon$ *and the estimate* $\hat{\boldsymbol{\pi}} := \boldsymbol{D}^{1/2}\boldsymbol{x}^{(T)}$ *satisfies* $\|\boldsymbol{D}^{-1}(\hat{\boldsymbol{\pi}} - \boldsymbol{\pi})\|_\infty \leq \epsilon$.

Note that $\overline{\gamma}_T$ of LOCGD is empirically smaller than that of LOCSOR. Hence, LOCGD is empirically slower than LOCSOR by only a small constant factor (e.g., twice as slow), a finding consistent with observations of their standard counterparts [19]. Nonetheless, LOCGD is much simpler and more amenable to parallelization on platforms such as GPUs compared to APPR.

## 4  Accelerated Local Iterative Methods

This section presents our key contributions where we propose faster local methods based on the Chebyshev method for solving Equ. (3) and the Heavy-Ball (HB) method for Equ. (4).

## 4.1 Local Chebyshev method

Compared with GS and GD, the standard Chebyshev method offers optimal acceleration in solving Equ. (3). Following existing techniques (e.g., see d'Aspremont et al. [11]), we show there exists an upper runtime bound $\tilde{\mathcal{O}}(m/\sqrt{\alpha})$ to meeting the stopping condition where $\tilde{\mathcal{O}}$ hides $\log 1/\epsilon$ (we presented it in Theorem C.6). Hence, the Chebyshev method is one of the optimal first-order linear solvers for solving Equ. (3). However, localizing Chebyshev poses greater challenges due to the additional momentum vector involved in updating $\boldsymbol{x}^{(t)}$. Our key observation is that *if a substantial reduction in the magnitudes of $\boldsymbol{r}^{(t)}$ is required within a subset of $\mathcal{S}_t$, then the corresponding momentum coordinates are likely to possess significant acceleration energy.* Intuitively, a viable strategy involves localizing both the residual and momentum vectors. For $t \geq 1$, denote the "momentum" vector as $\boldsymbol{\Delta}^{(t)} := \boldsymbol{x}^{(t)} - \boldsymbol{x}^{(t-1)}$ and $\delta_{t:t+1} = \delta_t \delta_{t+1}$, we propose the localized Chebyshev as the following

$$
\boxed{
\begin{aligned}
&\boldsymbol{\Phi}_\theta\left(\mathcal{S}_t, \boldsymbol{x}^{(t)}, \boldsymbol{r}^{(t)}, \mathcal{A} = \text{LocCH}\right): \\
&\hat{\boldsymbol{x}}^{(t)} \leftarrow (1 + \delta_{t:t+1})\boldsymbol{r}^{(t)}_{\mathcal{S}_t} + \delta_{t:t+1}\boldsymbol{\Delta}^{(t)}_{\mathcal{S}_t}, \quad \delta_{t+1} = \left(2\frac{1+\alpha}{1-\alpha} - \delta_t\right)^{-1} \\
&\boldsymbol{x}^{(t+1)} \leftarrow \boldsymbol{x}^{(t)} + \hat{\boldsymbol{x}}^{(t)}, \quad \boldsymbol{r}^{(t+1)} \leftarrow \boldsymbol{r}^{(t)} - \hat{\boldsymbol{x}}^{(t)} + \frac{1-\alpha}{1+\alpha}\boldsymbol{W}\hat{\boldsymbol{x}}^{(t)},
\end{aligned}
}
\tag{10}
$$

where $t \geq 1$ with the initials $\boldsymbol{x}^{(0)} = \boldsymbol{0}, \boldsymbol{x}^{(1)} = \boldsymbol{r}^{(0)}$, $\delta_0 = 0, \delta_1 = (1-\alpha)/(1+\alpha)$, and $\boldsymbol{W} = \boldsymbol{D}^{-1/2}\boldsymbol{A}\boldsymbol{D}^{-1/2}$ is normalized adjacency matrix. Our key strategy for analyzing (10) is to rewrite the updates of $\boldsymbol{r}^{(t)}$ as a nonhomogeneous second-order difference equation (see details in Lemma C.8)

$$
\boldsymbol{r}^{(t+1)} - 2\delta_{t+1}\boldsymbol{W}\boldsymbol{r}^{(t)} + \delta_{t:t+1}\boldsymbol{r}^{(t-1)} = \sum_{j=0}^{t}\left((1 + \delta_{j:j+1})\prod_{r=j+1}^{t}\delta_{r:r+1}\boldsymbol{Q}\boldsymbol{r}^{(j)}_{\mathcal{S}_{j,t}}\right), \tag{11}
$$

where we denote $\mathcal{S}_{j,t} = \mathcal{S}_j \cap \cdots \cap \mathcal{S}_{t-1} \cap \overline{\mathcal{S}}_t$ given $t \geq j \geq 0$ where $\overline{\mathcal{S}}_t = \mathcal{V}\backslash\mathcal{S}_t$. In the rest, we define $\tilde{\alpha} = (1 - \sqrt{\alpha})/(1 + \sqrt{\alpha})$ and recall the eigendecomposition of $\boldsymbol{D}^{-1/2}\boldsymbol{A}\boldsymbol{D}^{-1/2} = \boldsymbol{V}\boldsymbol{\Lambda}\boldsymbol{V}^\top$. Based on the above Equ. (11), we have the following key lemma.

**Lemma 4.1.** *Given $t \geq 1, \boldsymbol{x}^{(0)} = \boldsymbol{0}, \boldsymbol{x}^{(1)} = \boldsymbol{r}^{(0)}$. The residual $\boldsymbol{r}^{(t)}$ of LocCH defined in (10) can be expressed as the following*

$$
\boldsymbol{V}^\top\boldsymbol{r}^{(t)} = \delta_{1:t}\boldsymbol{Z}_t\boldsymbol{V}^\top\boldsymbol{r}^{(0)} + \delta_{1:t}t\boldsymbol{u}_{0,t} + 2\sum_{k=1}^{t-1}\delta_{k+1:t}(t-k)\boldsymbol{u}_{k,t},
$$

*where $\boldsymbol{Z}_t$ is a diagonal matrix such that $\|\boldsymbol{Z}_t\|_2 \leq 1$ and*

$$
\boldsymbol{u}_{k,t} = \begin{cases}
\sum_{j=1}^{t-1}\frac{\delta_{2:j}}{t}\boldsymbol{H}_{j,t}\left(\boldsymbol{I} - \frac{1-\alpha}{1+\alpha}\boldsymbol{\Lambda}\right)\boldsymbol{V}^\top\boldsymbol{r}^{(0)}_{\mathcal{S}_{0,j}} & \text{if } k = 0 \\
\sum_{j=k}^{t-1}\frac{\delta_{k+1:j}}{(t-k)}\boldsymbol{H}_{j,t}\left(\frac{1+\alpha}{1-\alpha}\boldsymbol{I} - \boldsymbol{\Lambda}\right)\boldsymbol{V}^\top\boldsymbol{r}^{(k)}_{\mathcal{S}_{k,j}} & \text{if } k \geq 1,
\end{cases}
$$

*where $\boldsymbol{H}_{k,t}$ is a diagonal matrix such that $\|\boldsymbol{H}_{k,t}\|_2 \leq t - k$.*

This key lemma essentially captures the process of residual reduction $\boldsymbol{r}^{(t)}$ of LocCH. Specifically, given current iteration $t$, we define the *running residual reduction rate* for $\boldsymbol{r}^{(k)}$ with $k = 0, 1, 2, \ldots, t-1$ of step $t$ as $\beta_{k,t}$, that is,

$$
\beta_{k,t} \triangleq \frac{\|\boldsymbol{u}_{k,t}\|_2}{\|\boldsymbol{r}^{(k)}\|_2}, \qquad \beta_k \triangleq \max_t \beta_{k,t}. \tag{12}
$$

Note that

$$
\beta_{k,t} \leq \underbrace{\frac{2\|\boldsymbol{r}^{(k)}_{\overline{\mathcal{S}}_k}\|_2}{(1-\alpha)\|\boldsymbol{r}^{(k)}\|_2}}_{\approx \mathcal{O}(\epsilon)} + \underbrace{\sum_{j=k+1}^{t-1}\frac{4\tilde{\alpha}^{j-k}(t-j)}{(1-\alpha)(t-k)}\frac{\|\boldsymbol{r}^{(k)}_{\mathcal{S}_{k,j}}\|_2}{\|\boldsymbol{r}^{(k)}\|_2}}_{\leq \frac{4\tilde{\alpha}}{(1-\alpha)(1-\tilde{\alpha})}},
$$

where whether the last term can be even smaller depends on $\|\boldsymbol{r}^{(k)}_{\mathcal{S}_{k,j}}\|_2$ for $\mathcal{S}_{k,j} = \mathcal{S}_k \cap \cdots \cap \mathcal{S}_{j-1} \cap \overline{\mathcal{S}}_j$. However, we notice that the running geometric mean $\overline{\beta}_t \triangleq (\prod_{j=0}^{t-1}(1 + \beta_j))^{1/t}$ is even smaller in practice. Based on these observations and the assumption on $\overline{\beta}_t$, we establish the following theorem.

**Theorem 4.2** (Runtime bound of LOCCH). *Given the configuration $\theta = (\alpha, \epsilon, s, \mathcal{G})$ with $\alpha \in (0, 1)$ and $\epsilon \leq 1/d_s$ and let $\boldsymbol{r}^{(T)}$ and $\boldsymbol{x}^{(T)}$ be returned by LOCCH defined in (10) for solving Equ. (3). For $t \geq 1$, the residual magnitude $\|\boldsymbol{r}^{(t)}\|_2$ has the following convergence bound*

$$\|\boldsymbol{r}^{(t)}\|_2 \leq \delta_{1:t} \prod_{j=0}^{t-1} (1 + \beta_j) y_t,$$

*where $y_t$ is a sequence of positive numbers solving $y_{t+1} - 2y_t + y_{t-1}/((1 + \beta_{t-1})(1 + \beta_t)) = 0$ with $y_0 = y_1 = \|\boldsymbol{r}^{(0)}\|_2$. Suppose the geometric mean $\overline{\beta}_t \triangleq (\prod_{j=0}^{t-1}(1 + \beta_j))^{1/t}$ of $\beta_t$ be such that $\overline{\beta}_t = 1 + \frac{c\sqrt{\alpha}}{1 - \sqrt{\alpha}}$ where $c \in [0, 2)$. There exists a real implementation of (10) such that the runtime $\mathcal{T}_{\text{LOCCH}}$ is bounded by*

$$\mathcal{T}_{\text{LOCCH}} \leq \Theta\left(\frac{(1 + \sqrt{\alpha})\overline{\text{vol}}(\mathcal{S}_T)}{\sqrt{\alpha}(2 - c)} \ln \frac{2y_T}{\epsilon}\right).$$

Golub & Overton [18] considered the approximate Chebyshev method by assuming that the inexact residual is sufficiently smaller than $\epsilon \|\boldsymbol{r}^{(t)}\|_2$, where $\epsilon$ must be small enough to ensure convergence. However, this assumption is overly stringent for our case. The novelty of our analysis lies in a more elegant treatment of a parameterized second-order difference equation, allowing us to circumvent this assumption. The nested APGD($\hat{\epsilon}$), namely ASPR proposed in Martínez-Rubio et al. [37] has runtime complexity $\tilde{\mathcal{O}}(|\mathcal{S}^*| \widetilde{\text{vol}}(\mathcal{S}^*)/\sqrt{\alpha} + |\mathcal{S}^*| \text{vol}(\mathcal{S}^*))$ where $\mathcal{S}^*$ is

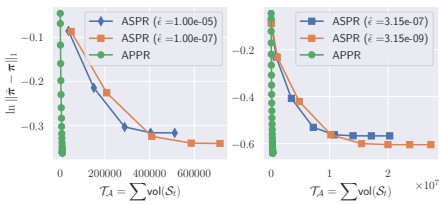

Figure 3: Comparison of runtime between APPR and ASPR. The setting is the same as in Fig. 1. Left $\epsilon = 10^{-4}$ while $\frac{1}{n}$ for right.

the optimal support of $\arg\min_{\boldsymbol{x}} \{f(\boldsymbol{x}) + \epsilon\|\boldsymbol{D}^{1/2}\boldsymbol{x}\|_1\}$ and $\widetilde{\text{vol}}(\mathcal{S}^*) = \text{nnz}(\boldsymbol{Q}_{\mathcal{S}^*, \mathcal{S}^*})$. Although it is difficult to compare our bound to this, one limitation of ASPR is that it assumes to call APGD($\hat{\epsilon}$) $\mathcal{O}(|\mathcal{S}^*|)$ times to finish in the worst case. However, our iteration complexity is $\tilde{\mathcal{O}}(1/(\sqrt{\alpha}(2 - c)))$. Asymptotically, $c = o(\sqrt{\alpha})$ ($\epsilon \to 0$), our complexity is $\tilde{\mathcal{O}}(1/\sqrt{\alpha})$ could be better than $\tilde{\mathcal{O}}(|\mathcal{S}^*|/\sqrt{\alpha})$. Fig. 3 presents a preliminary study on ASPR, indicating that it requires more operations than APPR.

We conclude our analysis by presenting a similar result for the local Heavy Ball (HB). Note the HB method is the one when $\delta_t\delta_{t+1} \to \tilde{\alpha}^2$ where $\tilde{\alpha} = (1 - \sqrt{\alpha})/(1 + \sqrt{\alpha})$. Hence, it has similar convergence analyses as to LOCCH shown in Theorem D.8. The LOCHB has the following updates

$$
\begin{aligned}
&\boldsymbol{\Phi}\left(\mathcal{S}_t; s, \epsilon, \alpha, \mathcal{G}, \mathcal{A} = \text{LOCHB}\right): \\
&\quad \hat{\boldsymbol{x}}^{(t)} \leftarrow (1 + \tilde{\alpha}^2)\boldsymbol{r}_{\mathcal{S}_t}^{(t)} + \tilde{\alpha}^2 \boldsymbol{\Delta}_{\mathcal{S}_t}^{(t)} \\
&\quad \boldsymbol{x}^{(t+1)} \leftarrow \boldsymbol{x}^{(t)} + \hat{\boldsymbol{x}}^{(t)}, \quad \boldsymbol{r}^{(t+1)} \leftarrow \boldsymbol{r}^{(t)} - \hat{\boldsymbol{x}}^{(t)} + \frac{1-\alpha}{1+\alpha}\boldsymbol{W}\hat{\boldsymbol{x}}^{(t)}.
\end{aligned}
\tag{13}
$$

## 5 Generalization and Open Problems

Our framework can be applied to various local methods for large-scale linear systems. Extensions of this framework to other linear systems are detailed in Tab. 2 of Appendix E. More broadly, we consider the feasibility of local methods for solving $\boldsymbol{Qx} = \boldsymbol{b}$, where $\boldsymbol{b}$ is a sparse vector ($|\text{supp}(\boldsymbol{b})| \ll n$) and $\boldsymbol{Q}$ is a positive definite, graph-induced matrix with bounded eigenvalues. This leads us to question whether all standard iterative methods can be effectively localized, raising two key questions

1. Given a graph-induced matrix $\boldsymbol{Q}$ and its spectral radius $\rho(\boldsymbol{Q}) < 1$, a standard solver $\mathcal{A}$, and the corresponding local evolving process $\boldsymbol{\Phi}_\theta(\mathcal{S}_t, \boldsymbol{x}^{(t)}, \boldsymbol{r}^{(t)}, \text{LOC}\mathcal{A})$, does a localized version of $\mathcal{A}$ (over $\mathcal{S}_t$) converge and have local runtime bounds?

2. Based on current analysis, Theorem 4.2 relies on the geometric mean of residual reduction on $\|\boldsymbol{r}^{(k)}\|_2$ being small. How feasible is acceleration within locality constraints? Specifically, a stronger bound could be established for solving Equ. (3) via LOCHB and LOCCH, with a graph-independent bound of

$$\mathcal{T}_{\text{LOC}\mathcal{A}} = \Theta\left(\frac{\overline{\text{vol}}(\mathcal{S}_T)}{\sqrt{\alpha}\overline{\gamma}_T} \ln \frac{C}{\epsilon}\right), \text{ where } C \text{ a graph-independent constant.}$$

Additionally, this work primarily focuses on using first-order neighbors at each iteration. An area for future exploration is generalizing to higher-order neighbors to determine if this leads to faster or more efficient methodologies, which remains an open question.

# 6 Experiments

We conduct experiments over 17 graphs to solve (3) and explore the local clustering task. We address the following questions: 1) Can iterative solvers be effectively localized? 2) How does the performance of accelerated local methods compare to non-accelerated ones? 3) Can our proposed methods reduce the number of operations required for local clustering? [6]

**Baselines.** We consider four baselines: 1) Conjugate Gradient Method (CGM) as a benchmark to compare local and non-local methods; 2) ISTA, the local method proposed by Fountoulakis et al. [13]; 3) FISTA, the momentum-based local algorithm proposed by Hu [22]; and 4) APPR, the classic local method proposed by Andersen et al. [2]. All methods are implemented in Python 3.10 with the numba library [33].

**Efficiency of localized algorithms.** To compare local solvers to their standard counterparts, we set $\alpha = 0.1$, randomly select 50 nodes from each graph to serve as $e_s$ in (3), and run standard GD, SOR, HB, and CH solvers along with their local counterparts: LOCGD, LOCSOR, LOCHB, and LOCCH. We measure the efficiency by the *speedup*, defined as the ratio between the runtime of the standard and local solver. The range of $\epsilon$ is $\epsilon \in [\frac{\alpha}{2(1+\alpha)d_s}, 10^{-4}/n]$. The results, presented in Fig. 4, clearly indicate that our design demonstrates significant speedup, especially around $\epsilon = 1/n$. Remarkably, they still show better performance even when $\epsilon \approx 10^{-4}/n$ (Fig. 5). These results suggest that local solvers are preferred over non-local ones when the precision requirement is in this range.

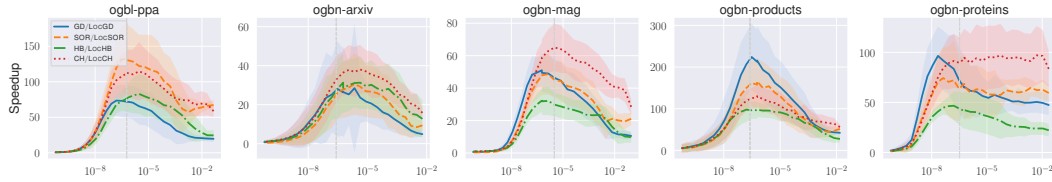

Figure 4: The speedup of local solvers as a function of $\epsilon$. The vertical line is $\epsilon = 1/n$.

**Comparison with local baselines and CGM.** We next compare our three accelerated methods with four baselines. Fig. 5 presents the $\ell_1$-estimation error in terms of the number of operations (quantified as $t \cdot \overline{\mathrm{vol}}(\mathcal{S}_t)$) executed. It is evident that our three solvers use significantly fewer operations compared to CGM and the other three local methods. Again, due to maintaining a nondecreasing set of active nodes, ISTA and FISTA require more operations

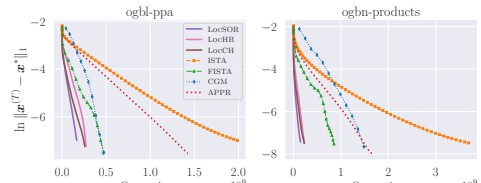

Figure 5: Estimation error as a function of operations required. ($\epsilon = 10^{-4}/n$)

than the locally evolving set process. Ours are more efficient than APPR, where $r^{(0)} = e_s$ is used.

**Efficiency in terms of $\alpha$ and huge-graph tests.** We demonstrate the performance of local solvers in terms of different $\alpha$ ranging from 0.005 to 0.25. Interestingly, in Fig. 13, LOCGD show faster convergence when $\alpha$ is small; this may be because of the advantages of monotonicity properties, which is not present in the accelerated methods. However, in other regions of $\alpha$, accelerated methods are faster. We also tested local solvers on two large-scale graphs where papers100M has 111M nodes and 1.6B edges while com-friendster has 65M nodes with 1.8B edges. Results are shown in Fig. 6; compared with current default local methods, it is several times faster, especially on ogbn-papers100M.

Figure 6: Performance on large-scale graphs.

---

[6]Additional experimental results, setups, and algorithm parameters are provided in Appendix F.

**Case study on local clustering.**
Following the experimental setup in Fountoulakis et al. [13], we consider the task of local clustering on 15 graphs. As partially demonstrated in Tab. 1, compared with APPR and FISTA, LocSOR uses the least operations and is the fastest, demonstrating the advantages of our proposed local solvers.

Table 1: Operations/runtime comparison on local clustering.

| $\mathcal{G}$ | Operations | | | Run Time (Seconds) | | |
|---|---|---|---|---|---|---|
| | APPR | LocSOR | FISTA | APPR | LocSOR | FISTA |
| $\mathcal{G}_1$ | 6.9e+05 | **6.5e+04** | 5.7e+05 | 0.127 | **0.043** | 0.093 |
| $\mathcal{G}_2$ | 6.7e+05 | **8.9e+04** | 4.4e+05 | 0.362 | **0.125** | 0.308 |
| $\mathcal{G}_3$ | 4.3e+05 | **3.5e+04** | 2.9e+05 | 0.069 | **0.014** | 0.042 |
| $\mathcal{G}_4$ | 5.7e+05 | **7.6e+04** | 4.4e+05 | 0.357 | **0.175** | 0.229 |
| $\mathcal{G}_5$ | 5.4e+05 | **9.0e+04** | 5.0e+05 | 0.072 | **0.055** | 0.084 |

## 7 Limitations and Conclusion

Our proposed algorithms may have the following limitations: 1) When $\alpha$ is small, the acceleration effect partially disappears, as observed in Fig. 13. This may be due to the limitations of global counterparts, where the residual may not decrease early; 2) Our new accelerated bound for LocCH depends on an empirically reasonable assumption of residual reduction but lacks theoretical justification.

We propose using a new locally evolving set process framework to characterize algorithm locality and demonstrate that several standard iterative solvers can be effectively localized, significantly speeding up current local solvers. Our local methods could be efficiently implemented into GPU architecture to accelerate the training of GNNs such as APPNP [27] and PPRGo [7]. We also offer open problems in developing faster local methods. It is worth exploring whether subsampling active nodes stochastically or using different queue strategies (priority rather than FIFO) could help speed up the framework further. It also remains interesting to see how to design local algorithms for conjugate direction-based methods such as CGM.

## Acknowledgments and Disclosure of Funding

The authors would like to thank the anonymous reviewers for their helpful comments. The work of Baojian Zhou is sponsored by Shanghai Pujiang Program (No. 22PJ1401300) and the National Natural Science Foundation of China (No. KRH2305047). The work of Deqing Yang is supported by Chinese NSF Major Research Plan No.92270121. The computations in this research were performed using the CFFF platform of Fudan University.

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

# Appendix / supplemental material

# A  Notations and Proof of Lemma 2.1

## A.1  List of Notations

In the rest of the appendix, we use the following notations:

|  | **Description** |
|---|---|
| $\mathcal{G}$ | An undirected connected simple graph with unit weights. |
| $\boldsymbol{A}$ | The adjacency matrix of $\mathcal{G}$. |
| $\boldsymbol{D}$ | The diagonal degree matrix of $\mathcal{G}$. |
| $\boldsymbol{W}$ | The normalized Laplacian matrix $\boldsymbol{W} \triangleq \boldsymbol{D}^{-1/2}\boldsymbol{A}\boldsymbol{D}^{-1/2}$. |
| $\boldsymbol{V}\boldsymbol{\Lambda}\boldsymbol{V}^\top$ | The eigendecomposition of $\boldsymbol{W}$ is given by $\boldsymbol{W} = \boldsymbol{D}^{-1/2}\boldsymbol{A}\boldsymbol{D}^{-1/2} = \boldsymbol{V}\boldsymbol{\Lambda}\boldsymbol{V}^\top$, where $\boldsymbol{\Lambda} = \mathbf{diag}(\lambda_1, \lambda_2, \ldots, \lambda_n)$, with $1 = \lambda_1 \geq \lambda_2 \geq \cdots \geq \lambda_n \geq -1$. When $\lambda_n = -1$, $\mathcal{G}$ is a bipartite graph. |
| $\boldsymbol{Q}$ | The underlying matrix of Equ. (3) is $\boldsymbol{Q} = \boldsymbol{I} - \frac{1-\alpha}{1+\alpha}\boldsymbol{D}^{-1/2}\boldsymbol{A}\boldsymbol{D}^{-1/2}$. |
| $\boldsymbol{r}^{(t)}$ | Given an estimate $\boldsymbol{x}^{(t)}$, the residual is defined as $\boldsymbol{r}^{(t)} \triangleq \boldsymbol{b} - \boldsymbol{Q}\boldsymbol{x}^{(t)}$. |
| $\tilde{\boldsymbol{r}}^{(t)}$ | The $\boldsymbol{D}^{1/2}$-shifted residual is defined as $\tilde{\boldsymbol{r}}^{(t)} = \boldsymbol{D}^{1/2}\boldsymbol{r}^{(t)}$. |
| $\alpha$ | The damping factor $\alpha$, which lies in the interval $(0,1)$. |
| $\tilde{\alpha}$ | A pre-defined constant $\tilde{\alpha} = (1 - \sqrt{\alpha})/(1 + \sqrt{\alpha})$. |
| $\mathcal{T}_\mathcal{A}$ | The total runtime of a local algorithm $\mathcal{A}$. |
| $T_t(x)$ | For $t \geq 1$, $T_t$ denotes the Chebyshev polynomial of the first kind, defined as $T_{t+1}(x) = 2xT_t(x) - T_{t-1}(x)$, with $T_0(x) = 1$, and $T_1(x) = x$. |
| $\delta_t$ | The ratio of $T_{t-1}$ and $T_t$, i.e., $\delta_t = T_{t-1}(\frac{1+\alpha}{1-\alpha})/T_t(\frac{1+\alpha}{1-\alpha})$, and $\delta_{t+1} = (2\frac{1+\alpha}{1-\alpha} - \delta_t)^{-1}$, with $\delta_1 = \frac{1-\alpha}{1+\alpha}$. |
| $\delta_{1:t}$ | The product of all $\delta_t$, i.e., $\delta_{1:t} \triangleq \prod_{i=1}^t \delta_i$. By default, we set $\delta_{0:1} = 0$. |
| $\mathcal{S}_{1:t}$ | The intersection of all $t$-ordered sets, i.e., $\mathcal{S}_{1:t} = \mathcal{S}_1 \cap \mathcal{S}_2 \cap \cdots \cap \mathcal{S}_t$. By default, we set $\mathcal{S}_{t:t-1} = \mathcal{V}$. |
| $\overline{\mathcal{S}}_t$ | The complement of $\mathcal{S}_t$, i.e., $\overline{\mathcal{S}}_t = \mathcal{V}\backslash\mathcal{S}_t$. |
| $\mathcal{S}_{j,t}$ | $\mathcal{S}_{j,t} = \mathcal{S}_j \cap \mathcal{S}_{j+1} \cap \cdots \cap \mathcal{S}_{t-1} \cap \overline{\mathcal{S}}_t = \mathcal{S}_{j:t-1} \cap \overline{\mathcal{S}}_t$. By default, we set $\mathcal{S}_{t,t} = \overline{\mathcal{S}}_t$. |

## A.2  Proof of Lemma 2.1

**Lemma 2.1** (Runtime bound of APPR [2]). *Given $\alpha \in (0,1)$ and the precision $\epsilon \leq 1/d_s$ for node $s \in \mathcal{V}$ with $\boldsymbol{p} \leftarrow \boldsymbol{0}, \boldsymbol{z} \leftarrow \boldsymbol{e}_s$ at the initial, $\mathrm{APPR}(\mathcal{G}, \epsilon, \alpha, s)$ defined in (2) returns an estimate $\boldsymbol{p}$ of $\boldsymbol{\pi}$. There exists a real implementation of (2) (e.g., Algo. 2) such that the runtime $\mathcal{T}_{\mathrm{APPR}}$ satisfies*

$$\mathcal{T}_{\mathrm{APPR}} \leq \Theta\left(1/(\alpha\epsilon)\right).$$

*Furthermore, the estimate $\hat{\boldsymbol{\pi}} := \boldsymbol{p}$ satisfies $\|\boldsymbol{D}^{-1}(\hat{\boldsymbol{\pi}} - \boldsymbol{\pi})\|_\infty \leq \epsilon$ and $\mathrm{vol}(\mathrm{supp}(\hat{\boldsymbol{\pi}})) \leq 2/((1-\alpha)\epsilon)$.*

*Proof.* To find an upper bound of $\mathcal{T}_{\mathrm{APPR}}$, we add a time index for all active nodes $u_1, u_2, \ldots, u_t$ processed in APPR of Algo. 2, a real implementation of (2). The parameter $t$ is the number of active nodes processed, and $u_i$ is the node dequeued at time $i$. So, updates of $\boldsymbol{p}$ and $\boldsymbol{z}$ are from $(\boldsymbol{0}, \boldsymbol{e}_s) = (\boldsymbol{p}^{(0)}, \boldsymbol{z}^{(0)})$ to $(\boldsymbol{p}^{(t)}, \boldsymbol{z}^{(t)})$ as follows:

$$(\boldsymbol{p}^{(0)}, \boldsymbol{z}^{(0)}) \xrightarrow{u_1} (\boldsymbol{p}^{(1)}, \boldsymbol{z}^{(1)}) \xrightarrow{u_2} (\boldsymbol{p}^{(2)}, \boldsymbol{z}^{(2)}) \cdots \xrightarrow{u_t} (\boldsymbol{z}^{(t)}, \boldsymbol{p}^{(t)}).$$

For each active $u_i$, the updates of $(\boldsymbol{p}, \boldsymbol{z})$ by definition can be represented as

$$\boldsymbol{p}^{(i)} = \boldsymbol{p}^{(i-1)} + \alpha z_{u_i}^{(i-1)} \cdot \boldsymbol{e}_{u_i}, \quad \boldsymbol{z}^{(i)} = \boldsymbol{z}^{(i-1)} - \frac{(1+\alpha)z_{u_i}^{(i-1)}}{2}\boldsymbol{e}_{u_i} + \frac{(1-\alpha)z_{u_i}^{(i-1)}}{2}\boldsymbol{A}\boldsymbol{D}^{-1}\boldsymbol{e}_{u_i}.$$

Since $\boldsymbol{z}^{(0)} = \boldsymbol{e}_s \geq \boldsymbol{0}$, then $\boldsymbol{z}^{(i)} \geq 0$ and $\boldsymbol{p}^{(i)} \geq 0$ for all $i$ by induction. Note that $\|\boldsymbol{A}\boldsymbol{D}^{-1}\boldsymbol{e}_{u_i}\|_1 = 1$, we have the following relation from the updates of $\boldsymbol{z}$

$$\|\boldsymbol{z}^{(i)}\|_1 = \|\boldsymbol{z}^{(i-1)}\|_1 - \alpha z_{u_i}^{(i-1)} \iff z_{u_i}^{(i-1)} = \frac{\|\boldsymbol{z}^{(i-1)}\|_1 - \|\boldsymbol{z}^{(i)}\|_1}{\alpha}.$$

Note $\epsilon d_{u_i} \leq z_{u_i}^{(i-1)}$ for each active $u_i$. Summing the above equation over $i = 1, 2, \dots, t$, we have

$$\sum_{i=1}^t \epsilon d_{u_i} \leq \sum_{i=1}^t z_{u_i}^{(i-1)} = \sum_{i=1}^t \left( \frac{\|\boldsymbol{z}^{(i-1)}\|_1 - \|\boldsymbol{z}^{(i)}\|_1}{\alpha} \right) = \frac{\|\boldsymbol{z}^{(0)}\|_1 - \|\boldsymbol{z}^{(t)}\|_1}{\alpha} \leq \frac{\|\boldsymbol{z}^{(0)}\|_1}{\alpha} = \frac{1}{\alpha},$$

where note $\|\boldsymbol{z}^{(0)}\|_1 = \|\boldsymbol{e}_s\|_1 = 1$ by the initial condition. Since $\sum_{i=1}^t d_{u_i}$ exactly captures the number of operations needed, the runtime of Algo. 2 is then bounded as

$$\mathcal{T}_{\text{APPR}} = \Theta \left( \sum_{i=1}^t d_{u_i} \right) \leq \Theta \left( \frac{1}{\alpha\epsilon} \right).$$

To check the quality of estimate $\boldsymbol{p}$, using the updates of $\boldsymbol{p}^{(i)}$ and summing over all $i$, we have

$$\boldsymbol{p}^{(t)} = \alpha \sum_{i=1}^t z_{u_i}^{(i-1)} \boldsymbol{e}_{u_i} = \alpha \underbrace{\left( \frac{(1+\alpha)}{2}\boldsymbol{I} - \frac{(1-\alpha)}{2}\boldsymbol{A}\boldsymbol{D}^{-1} \right)^{-1}}_{\boldsymbol{\Pi}} \sum_{i=1}^t \left( \boldsymbol{z}^{(i-1)} - \boldsymbol{z}^{(i)} \right) = \boldsymbol{\pi} - \boldsymbol{\Pi}\boldsymbol{z}^{(t)},$$

where $\boldsymbol{\Pi}$ is the PPR matrix. The above gives us $\boldsymbol{\pi} - \boldsymbol{p}^{(t)} = \boldsymbol{\Pi} \cdot \boldsymbol{z}^{(t)}$. Since $\mathcal{G}$ is undirected, the $\boldsymbol{\Pi}$ matrix satisfies $\pi_v[u] = (d_u/d_v)\pi_u[v]$ where $\pi_v[u]$ is the $u$-th element of PPR vector of sourcing node $v$. Consider each $u$-th element of $\boldsymbol{\Pi} \cdot \boldsymbol{z}^{(t)}$

$$(\boldsymbol{\Pi} \cdot \boldsymbol{z}^{(t)})_u = \sum_{v \in \mathcal{V}} z_v^{(t)} \cdot \pi_v[u] = \sum_{v \in \mathcal{V}} z_v^{(t)} \cdot \frac{d_u}{d_v} \pi_u[v] \leq \epsilon d_u \sum_{v \in \mathcal{V}} \pi_u[v] = \epsilon d_u,$$

where the last equality is due to $\sum_{v \in \mathcal{V}} \pi_u[v] = 1$. Hence, $(\boldsymbol{\pi} - \boldsymbol{p}^{(t)})_u = (\boldsymbol{\Pi} \cdot \boldsymbol{z}^{(t)})_u \leq \epsilon d_u$, which indicates $\|\boldsymbol{D}^{-1}(\boldsymbol{p}^{(t)} - \boldsymbol{\pi})\|_\infty \leq \epsilon$. To see the bound of $\text{vol}(\text{supp}(\boldsymbol{p}^{(t)}))$, note for any $u \in \text{supp}(\boldsymbol{p}^{(t)})$, it was an active node and there was at least $\tilde{z}_u(1-\alpha)/2$ remain in $u$-th entry of $\boldsymbol{z}^{(t)}$ where we denote $\tilde{z}_u$ as the residual before the last push operation of node $u$; hence

$$\sum_{u \in \text{supp}(\boldsymbol{p})} d_u \leq \sum_{u \in \text{supp}(\boldsymbol{p})} \frac{\tilde{z}_u}{\epsilon} = \sum_{u \in \text{supp}(\boldsymbol{p})} \frac{\tilde{z}_u(1-\alpha)/2}{\epsilon(1-\alpha)/2} \leq \frac{\sum_{u \in \text{supp}(\boldsymbol{p})} z_u^{(t)}}{\epsilon(1-\alpha)/2} \leq \frac{2}{(1-\alpha)\epsilon}.$$

$\square$

---

**Algo. 2** $\text{APPR}(\alpha, \epsilon, s, \mathcal{G})$ via FIFO Queue

1: Initialize: $\boldsymbol{p} \leftarrow \boldsymbol{0}, \boldsymbol{z} \leftarrow \boldsymbol{e}_s, \mathcal{Q} \leftarrow \{*, s\}, t = -1$
2: **while true do**
3:      $u \leftarrow \mathcal{Q}.\text{dequeue}()$
4:      **if** u == * **then**
5:          **if** $\mathcal{Q} = \emptyset$ **then**
6:              **break**
7:          $t \leftarrow t + 1$        // Starting time of $\mathcal{S}_t$
8:          $\mathcal{Q}.\text{enqueue}(*)$     // Marker for next $\mathcal{S}_{t+1}$
9:          **continue**
10:     $\tilde{z} \leftarrow z_u$
11:     $p_u \leftarrow p_u + \alpha \cdot \tilde{z}$
12:     $z_u \leftarrow \tilde{z} \cdot (1 - \alpha)/2$
13:     **for** $v \in \mathcal{N}(u)$ **do**
14:          $z_v \leftarrow z_v + \frac{(1-\alpha)}{2} \cdot \frac{\tilde{z}}{d_u}$
15:          **if** $z_v \geq \epsilon d_v$ **and** $v \notin \mathcal{Q}$ **then**
16:             $\mathcal{Q}.\text{enqueue}(v)$
17:     **if** $z_u \geq \epsilon d_u$ **and** $u \notin \mathcal{Q}$ **then**
18:          $\mathcal{Q}.\text{enqueue}(u)$
19: **return** $\boldsymbol{p}$

Indeed, Lemma 2.1 is a special case of Theorem 1 in [2]. The proof outlined above adheres to the key strategy demonstrated in that theorem, which involves exploring the monotonicity and nonnegativity of $\boldsymbol{z}$. The real implementation of APPR, as shown in Algo. 2, presents a typical queue-based method. It has monotonic properties during the updates of $\boldsymbol{p}$ and $\boldsymbol{z}$ (Lines 10-16 of Algo. 2). It also holds element-wise that $\boldsymbol{p} \geq 0$ and $\boldsymbol{z} \geq 0$. The operations of $\mathcal{Q}.\text{enqueue}(u)$, $\mathcal{Q}.\text{dequeue}()$, and $v \notin \mathcal{Q}$ are all in $\mathcal{O}(1)$. Line 4 to Line 9 is to design the marker for distinguishing between $\mathcal{S}_t$ and $\mathcal{S}_{t+1}$. If all active nodes are processed and no more active nodes are added into $\mathcal{Q}$, then $\mathcal{Q}$ will be empty, and finally, the algorithm returns an estimate $\boldsymbol{p}$ of $\boldsymbol{\pi}$.

# B    Local Iterative Methods via Evolving Set Process

**Justification of an equivalent condition.** We make the justification of an equivalent stop condition for solving (3). Note we require a local solver to return an estimate $\hat{\boldsymbol{\pi}}$ satisfies

$$\|\boldsymbol{D}^{-1}\left(\hat{\boldsymbol{\pi}} - \boldsymbol{\pi}\right)\|_\infty \le \epsilon \tag{14}$$

Since we define $\boldsymbol{Qx} = \boldsymbol{b}$ as $\left(\boldsymbol{I} - \frac{1-\alpha}{1+\alpha}\boldsymbol{D}^{-1/2}\boldsymbol{A}\boldsymbol{D}^{-1/2}\right)\boldsymbol{x} = \frac{2\alpha}{1+\alpha}\boldsymbol{D}^{-1/2}\boldsymbol{e}_s$. With $\boldsymbol{r}^{(t)} = \boldsymbol{b} - \boldsymbol{Qx}^{(t)}$ and the stop condition

$$\|\boldsymbol{D}^{-1/2}\boldsymbol{r}^{(t)}\|_\infty \le \frac{2\alpha\epsilon}{1+\alpha}$$

ensures the estimate $\hat{\boldsymbol{\pi}} = \boldsymbol{D}^{1/2}\boldsymbol{x}^{(t)}$ satisfies (14). To see this, since $\boldsymbol{\pi} = \boldsymbol{D}^{1/2}\boldsymbol{x}^*$, we have

$$\begin{aligned}
\|\boldsymbol{D}^{-1}\left(\hat{\boldsymbol{\pi}} - \boldsymbol{\pi}\right)\|_\infty &= \|\boldsymbol{D}^{-1/2}(\boldsymbol{x}^{(t)} - \boldsymbol{x}^*)\|_\infty \\
&= \|\boldsymbol{D}^{-1/2}(\boldsymbol{Q}^{-1}\boldsymbol{b} - \boldsymbol{Q}^{-1}\boldsymbol{r}^{(t)} - \boldsymbol{Q}^{-1}\boldsymbol{b})\|_\infty \\
&= \|\boldsymbol{D}^{-1/2}\boldsymbol{Q}^{-1}\boldsymbol{D}^{1/2}\boldsymbol{D}^{-1/2}\boldsymbol{r}^{(t)}\|_\infty \\
&\le \|\boldsymbol{D}^{-1/2}\boldsymbol{Q}^{-1}\boldsymbol{D}^{1/2}\|_\infty \cdot \|\boldsymbol{D}^{-1/2}\boldsymbol{r}^{(t)}\|_\infty \\
&\le \|\boldsymbol{D}^{-1/2}\boldsymbol{Q}^{-1}\boldsymbol{D}^{1/2}\|_\infty \cdot \frac{2\alpha\epsilon}{1+\alpha}
\end{aligned}$$

where $\boldsymbol{D}^{-1/2}\boldsymbol{Q}^{-1}\boldsymbol{D}^{1/2} = (\boldsymbol{I} - \frac{1-\alpha}{1+\alpha}\boldsymbol{D}^{-1}\boldsymbol{A})^{-1} = \sum_{i=0}^\infty (\frac{1-\alpha}{1+\alpha}\boldsymbol{D}^{-1}\boldsymbol{A})^i$. This leads to

$$\begin{aligned}
\|\boldsymbol{D}^{-1}\left(\hat{\boldsymbol{\pi}} - \boldsymbol{\pi}\right)\|_\infty &\le \|\sum_{i=0}^\infty (\frac{1-\alpha}{1+\alpha}\boldsymbol{D}^{-1}\boldsymbol{A})^i\|_\infty \cdot \frac{2\alpha\epsilon}{1+\alpha} \\
&\le \frac{1+\alpha}{2\alpha} \cdot \frac{2\alpha\epsilon}{1+\alpha} \\
&= \epsilon.
\end{aligned}$$

## B.1    Local Variant of GS-SOR and Proof of Lemma 3.2

The Gauss-Seidel Successive Over-Relaxation (GS-SOR) solver (see Section 11.2.7 of Golub & Van Loan [19]) for the linear system $\boldsymbol{M}\boldsymbol{\pi} = \boldsymbol{s}$ via the following forward substitution

   **for** $i$ **in** $\mathcal{V} := \{1, 2, \ldots, n\}$ **do** :

$$p_i^{(t+1)} = \omega\left(s_i - \sum_{j=1}^{i-1} M_{ij}p_j^{(t+1)} - \sum_{j=i+1}^n M_{ij}p_j^{(t)}\right)/M_{ii} + (1-\omega)p_i^{(t)},$$

where $\boldsymbol{p}$ is updated from $\boldsymbol{p}^{(t)}$ to $\boldsymbol{p}^{(t+1)}$. When the relaxation parameter $\omega = 1$, GS-SOR reduces to the standard GS method. Equivalently, let $\Delta_i = (i-1)/n$ for $i = 1, 2, \ldots, n$, then GS-SOR updates can be sequentially represented as

   **for** $i$ **in** $\mathcal{V} := \{1, 2, \ldots, n\}$ **do** :

$$\boldsymbol{p}^{(t+\Delta_{i+1})} \leftarrow \boldsymbol{p}^{(t+\Delta_i)} + \frac{\omega}{M_{ii}}\left(s_i - \sum_{j=1}^{i-1} M_{ij}p_j^{(t+\Delta_i)} - \sum_{j=i}^n M_{ij}p_j^{(t+\Delta_i)}\right)\cdot\boldsymbol{e}_i.$$

Therefore, it is natural to define the following local variant of GS-SOR.

**Definition B.1** (Local variant of GS-SOR).  Consider the linear system $\boldsymbol{M}\boldsymbol{\pi} = \boldsymbol{s}$. For $t \ge 0$, we are given an active node set $\mathcal{S}_t = \{u_1, u_2, \ldots, u_{|\mathcal{S}_t|}\}$ and let $\Delta_i = (i-1)/|\mathcal{S}_t|$ for $i = 1, 2, \ldots, |\mathcal{S}_t|$, providing $\omega \in (0, 2)$, it is natural to define the *local variant* of GS-SOR as follows:

   **for** $u_i$ **in** $\mathcal{S}_t := \{u_1, u_2, \ldots, u_{|\mathcal{S}_t|}\}$ **do** :

$$\boldsymbol{p}^{(t+\Delta_{i+1})} \leftarrow \boldsymbol{p}^{(t+\Delta_i)} + \frac{\omega}{M_{u_i u_i}}\left(s_{u_i} - \sum_{j=1}^{i-1} M_{u_i u_j}p_{u_j}^{(t+\Delta_i)} - \sum_{j=i}^n M_{u_i u_j}p_{u_j}^{(t+\Delta_i)}\right)\cdot\boldsymbol{e}_{u_i}, \tag{15}$$

where $\mathcal{S}_t \subseteq \mathcal{V}$. When $\omega = 1$ and $\mathcal{S}_t = \mathcal{V} = \{1, 2, \ldots, n\}$, it reduces to the standard GS.

We use the above definition to show APPR is a local variant of GS-SOR as the following.

**Lemma 3.2** (New local evolving-based bound for APPR). *Let $M = \alpha^{-1}\left(I - \frac{1-\alpha}{2}\left(I + AD^{-1}\right)\right)$ and $s = e_s$. The linear system $M\pi = s$ is equivalent to Equ. (1). Given $p^{(0)} = 0$, $z^{(0)} = e_s$ with $\omega \in (0, 2)$, the local variant of GS-SOR (15) for $M\pi = s$ can be formulated as*

$$p^{(t+\Delta_{i+1})} \leftarrow p^{(t+\Delta_i)} + \frac{\omega z_{u_i}^{(t+\Delta_i)}}{M_{u_i u_i}} e_{u_i}, \quad z^{(t+\Delta_{i+1})} \leftarrow z^{(t+\Delta_i)} - \frac{\omega z_{u_i}^{(t+\Delta_i)}}{M_{u_i u_i}} M e_{u_i},$$

*where $u_i$ is an active node in $\mathcal{S}_t$ satisfying $z_{u_i} \geq \epsilon d_{u_i}$ and $\Delta_i = (i-1)/|\mathcal{S}_t|$. Furthermore, when $\omega = \frac{1+\alpha}{2}$, this method reduces to APPR given in (7), and there exists a real implementation (Aglo. 2) of APPR such that the runtime $\mathcal{T}_{\text{APPR}}$ is bounded by*

$$\mathcal{T}_{\text{APPR}} \leq \frac{\overline{\text{vol}(\mathcal{S}_T)}}{\alpha \hat{\gamma}_T} \ln \frac{C_T}{\epsilon}, \text{ where } \frac{\overline{\text{vol}(\mathcal{S}_T)}}{\hat{\gamma}_T} \leq \frac{1}{\epsilon}, C_T = \frac{2}{(1-\alpha)|\mathcal{I}_T|}, \hat{\gamma}_T \triangleq \frac{1}{T} \sum_{t=0}^{T-1} \left\{ \frac{\sum_{i=1}^{|\mathcal{S}_t|} |z_{u_i}^{(t+\Delta_i)}|}{\|z^{(t)}\|_1} \right\}.$$

*Proof.* Note that $M\pi = s$ is equivalent to Equ. (1). We first rewrite $p^{(t+\Delta_{i+1})}$ in terms of the residual $z$. The residual $z$ at time $t + \Delta_i$ can be written as $z^{(t+\Delta_i)} = s - Mp^{(t+\Delta_i)}$. Note $s_{u_i} - \sum_{j=1}^{i-1} M_{u_i u_j} p_{u_j}^{(t+\Delta_i)} - \sum_{j=i}^{n} M_{u_i u_j} p_{u_j}^{(t+\Delta_i)} = (s - Mp^{(t+\Delta_i)})_{u_i} = z_{u_i}^{(t+\Delta_i)}$. Then, the updates of local GS-SOR defined in (15) can be rewritten as $p^{(t+\Delta_{i+1})} = p^{(t+\Delta_i)} + \frac{\omega}{M_{u_i u_i}} z_{u_i}^{(t+\Delta_i)} \cdot e_{u_i}$. Hence, the updates of $z^{(t+\Delta_i)}$ can be written as

$$z^{(t+\Delta_{i+1})} = s - Mp^{(t+\Delta_{i+1})} = s - M\left(p^{(t+\Delta_i)} + \frac{\omega z_{u_i}^{(t+\Delta_i)}}{M_{u_i u_i}} \cdot e_{u_i}\right) = z^{(t+\Delta_i)} - \frac{\omega z_{u_i}^{(t+\Delta_i)}}{M_{u_i u_i}} M e_{u_i}.$$

Note the diagonal element $M_{u_i u_i} = (1+\alpha)/(2\alpha)$. Hence, when $\omega = \frac{1+\alpha}{2}$, we have

$$p^{(t+\Delta_{i+1})} = p^{(t+\Delta_i)} + \alpha z_{u_i}^{(t+\Delta_i)} \cdot e_{u_i}$$

$$z^{(t+\Delta_{i+1})} = z^{(t+\Delta_i)} - \alpha z_{u_i}^{(t+\Delta_i)} M e_{u_i} = z^{(t+\Delta_i)} - z_{u_i}^{(t+\Delta_i)} \left(\frac{1+\alpha}{2} I - \frac{1-\alpha}{2} AD^{-1}\right) e_{u_i}.$$

The above updates match APPR's evolving set process formulation in (7). The rest is to show a new runtime bound. Adding $\ell_1$-norm on both sides of the above equation, then note $\|z^{(t+\Delta_{i+1})}\|_1 = \|z^{(t+\Delta_i)}\|_1 - \alpha z_{u_i}^{(t+\Delta_i)}$ for $i = 1, 2, \ldots, |\mathcal{S}_t|$. We have

$$\|z^{(t+1)}\|_1 = \left(1 - \frac{\alpha \sum_{i=1}^{|\mathcal{S}_t|} z_{u_i}^{(t+\Delta_i)}}{\|z^{(t)}\|_1}\right) \|z^{(t)}\|_1 = (1 - \alpha \beta_t) \|z^{(t)}\|_1 = \prod_{i=0}^{t} (1 - \alpha \beta_t) \|z^{(0)}\|_1,$$

where we define $\beta_t := \frac{\sum_{i=1}^{|\mathcal{S}_t|} |z_{u_i}^{(t+\Delta_i)}|}{\|z^{(t)}\|_1}$. Let $t = T - 1$, we have

$$\ln \frac{\|z^{(T)}\|_1}{\|z^{(0)}\|_1} = \sum_{t=0}^{T-1} \ln(1 - \alpha \beta_t) \leq -\sum_{t=0}^{T-1} \alpha \beta_t \quad \Rightarrow \quad T \leq \frac{1}{\alpha \hat{\gamma}_T} \ln \frac{\|z^{(0)}\|_1}{\|z^{(T)}\|_1},$$

where the first inequality is due to $\ln(1 + x) \leq x$ for $x > -1$. For each nonzero node $u \in \mathcal{I}_T = \{z_u^{(T)} : z_u^{(T)} \neq 0, u \in \mathcal{V}\}$, consider the last time $t'$ that it was altered. Then, either the alteration came from $u$ being an active node, with $z_u^{(t')} \geq d_u \epsilon$, and after the PUSH operation it became $z_u^{(t'')} \geq \frac{(1-\alpha)d_u}{2}\epsilon$; or the alteration came from a neighboring node $v_u \in \mathcal{N}(u)$ pushing its mass onto $u$, which ensures that $z_u^{(t'')} \geq \frac{(1-\alpha)}{2d_{v_u}} z_{v_u}^{(t')} \geq \frac{(1-\alpha)}{2}\epsilon$. These two cases provide a lower bound of $\frac{1-\alpha}{2}\epsilon$. Hence, $\|z^{(T)}\|_1 \geq \frac{\epsilon(1-\alpha)|\mathcal{I}_T|}{2}$, which leads to the corresponding constant $C_T$.

To see the lower bound of $1/\epsilon$, note $\epsilon d_{u_i} \leq z_{u_i}^{(t+\Delta_i)}$ for all $i = 1, 2, \ldots, |\mathcal{S}_t|$. Then we have

$$\epsilon \, \text{vol}(\mathcal{S}_t) \leq \sum_{i=1}^{|\mathcal{S}_t|} z_{u_i}^{(t+\Delta_i)}$$

$$= \beta_t \|z^{(t)}\|_1$$

$$\leq \beta_t,$$

where we defined $\beta_t := \frac{\sum_{i=1}^{|\mathcal{S}_t|} |z_{u_i}^{(t+\Delta_i)}|}{\|\boldsymbol{z}^{(t)}\|_1}$ and the last inequality is due to the monotonic decreasing of $\|\boldsymbol{z}^{(t)}\|_1$, i.e., $1 \geq \|\boldsymbol{z}^{(0)}\|_1 \geq \cdots \geq \|\boldsymbol{z}^{(T)}\|_1$. Applying the above inequality for all $t = 0, 1, 2 \ldots, T - 1$, it leads to

$$\epsilon \operatorname{vol}(\mathcal{S}_t) \leq \beta_t$$

$$\Rightarrow \quad \epsilon \sum_{t=0}^{T-1} \operatorname{vol}(\mathcal{S}_t) \leq \sum_{t=0}^{T-1} \beta_t$$

$$\Rightarrow \quad \frac{\overline{\operatorname{vol}}(\mathcal{S}_T)}{\hat{\gamma}_T} \leq \frac{1}{\epsilon},$$

where the last derivation is from the fact that $\hat{\gamma}_T = \frac{1}{T}\left\{ \sum_{t=0}^{T-1} \beta_t := \frac{\sum_{i=1}^{|\mathcal{S}_t|} |z_{u_i}^{(t+\Delta_i)}|}{\|\boldsymbol{z}^{(t)}\|_1} \right\}$.  $\square$

*Remark* B.2. The connection between APPR and the Gauss-Seidel is not new [28, 29, 16, 9]. Our work is the first work that has linked APPR and the Gauss-Seidel with a locally evolving set process.

## B.2    LOCSOR and Proof of Theorem 3.3

In this subsection, recall we defined $\tilde{\boldsymbol{r}}^{(t)} = \boldsymbol{D}^{1/2}\boldsymbol{r}^{(t)}$.

**Lemma B.3** (Local iteration complexity of LOCSOR ($\omega \leq 1$)). *Denote $\mathcal{S}_t = \{u_1, u_2, \ldots, u_{|\mathcal{S}_t|}\}$ as the active node set at the $t$-th iteration. When $\omega \in (0,1]$, all vectors $\tilde{\boldsymbol{r}}^{(t)} \geq 0$ are nonnegative and magnitudes are decreasing $\|\tilde{\boldsymbol{r}}^{(t+1)}\|_1 < \|\tilde{\boldsymbol{r}}^{(t)}\|_1$. Let $T$ be the total number of iterations needed. Then, at iteration $T$, we have*

$$T \in \frac{(1+\alpha)}{2\alpha\omega\overline{\gamma}_T}\left[1 - \frac{\|\tilde{\boldsymbol{r}}^{(T)}\|_1}{\|\tilde{\boldsymbol{r}}^{(0)}\|_1}, \ln\frac{\|\tilde{\boldsymbol{r}}^{(0)}\|_1}{\|\tilde{\boldsymbol{r}}^{(T)}\|_1}\right], \quad \overline{\gamma}_T \triangleq \frac{1}{T}\sum_{t=0}^{T-1}\left\{\gamma_t \triangleq \sum_{i=1}^{|\mathcal{S}_t|} \frac{\tilde{r}_{u_i}^{(t+\Delta_i)}}{\|\tilde{\boldsymbol{r}}^{(t)}\|_1}\right\}, \quad (16)$$

*where $\overline{\gamma}_t = t^{-1}\sum_{\tau=0}^{t-1}\gamma_\tau$ is the mean of active ratio factors defined in Equ. (6).*

*Proof.* Recall $u_i \in \mathcal{S}_t = \{u_1, \ldots, u_{|\mathcal{S}_t|}\}$ and $\Delta_i = \frac{i-1}{|\mathcal{S}_t|}$, LOCSOR in Algo. 3 updates

$$\boldsymbol{x}^{(t+\Delta_{i+1})} = \boldsymbol{x}^{(t+\Delta_i)} + \omega r_{u_i}^{(t+\Delta_i)} \cdot \boldsymbol{e}_{u_i}$$

$$\boldsymbol{r}^{(t+\Delta_{i+1})} = \boldsymbol{r}^{(t+\Delta_i)} - \omega r_{u_i}^{(t+\Delta_i)} \cdot \boldsymbol{e}_{u_i} + \frac{(1-\alpha)\omega}{1+\alpha} r_{u_i}^{(t+\Delta_i)} \cdot \boldsymbol{D}^{-1/2}\boldsymbol{A}\boldsymbol{D}^{-1/2}\boldsymbol{e}_{u_i}$$

Note $\boldsymbol{r} \geq \boldsymbol{0}$ during updates when $\omega \in (0,1]$ and recall $\tilde{\boldsymbol{r}}^{(t)} = \boldsymbol{D}^{1/2}\boldsymbol{r}^{(t)}$, we have

$$\|\tilde{\boldsymbol{r}}^{(t+\Delta_{i+1})} + \omega\tilde{r}_{u_i}^{(t+\Delta_i)} \cdot \boldsymbol{e}_u\|_1 = \|\tilde{\boldsymbol{r}}^{(t+\Delta_i)} + \frac{(1-\alpha)\omega}{1+\alpha}\tilde{r}_{u_i}^{(t+\Delta_i)} \cdot \boldsymbol{A}\boldsymbol{D}^{-1}\boldsymbol{e}_{u_i}\|_1$$

$$\|\tilde{\boldsymbol{r}}^{(t+\Delta_{i+1})}\|_1 + \omega\tilde{r}_{u_i}^{(t+\Delta_i)} = \|\tilde{\boldsymbol{r}}^{(t+\Delta_i)}\|_1 + \frac{(1-\alpha)\omega}{1+\alpha}\tilde{r}_{u_i}^{(t+\Delta_i)}.$$

Summing over the above equations over $u_i$, we have

$$\|\tilde{\boldsymbol{r}}^{(t+1)}\|_1 = \|\tilde{\boldsymbol{r}}^{(t)}\|_1 - \frac{2\alpha\omega}{1+\alpha}\sum_{i=1}^{|\mathcal{S}_t|}\tilde{r}_{u_i}^{(t+\Delta_i)} = \left(1 - \frac{2\alpha\omega}{1+\alpha}\underbrace{\sum_{i=1}^{|\mathcal{S}_t|}\frac{\tilde{r}_{u_i}^{(t+\Delta_i)}}{\|\tilde{\boldsymbol{r}}^{(t)}\|_1}}_{\gamma_t}\right)\|\tilde{\boldsymbol{r}}^{(t)}\|_1. \quad (17)$$

Given $\{x_i\}_{i=0}^{T-1}$ and $x_i \in (0,1)$, the Weierstrass product inequality provides $1 - \sum_{i=0}^{T-1} x_i \leq \prod_{i=0}^{T-1}(1 - x_i)$. By using this inequality, we continue to have a lower bound of $T$ as the following

$$1 - \sum_{t=0}^{T-1}\frac{2\alpha\omega\gamma_t}{1+\alpha} \leq \prod_{t=0}^{T-1}\left(1 - \frac{2\alpha\omega\gamma_t}{1+\alpha}\right) = \frac{\|\tilde{\boldsymbol{r}}^{(T)}\|_1}{\|\tilde{\boldsymbol{r}}^{(0)}\|_1} \quad \Rightarrow \quad \frac{(1+\alpha)\left(1 - \|\tilde{\boldsymbol{r}}^{(T)}\|_1/\|\tilde{\boldsymbol{r}}^{(0)}\|_1\right)}{2\alpha\omega\overline{\gamma}_T} \leq T.$$

To get upper bound of $\gamma_t$, note each active residual $\tilde{r}_{u_i}^{(t+\Delta_i)}$ pushes at most $\frac{(1-\alpha)\omega}{(1+\alpha)}$ times magnitude to $\tilde{\boldsymbol{r}}_{u_{i+1}}, \tilde{\boldsymbol{r}}_{u_{i+2}}$, and $\tilde{\boldsymbol{r}}_{u_{|\mathcal{S}_t|}}$; hence, $\sum_{j=i}^{|\mathcal{S}_t|}\tilde{r}_{u_j}^{(t+\Delta_j)}$ will increase by at most $\tilde{r}_{u_i}^{(t+\Delta_i)} \cdot \frac{(1-\alpha)\omega}{(1+\alpha)} \leq \tilde{r}_{u_i}^{(t+\Delta_i)}$ in total. Hence, overall $u_i$, we have

$$\|\tilde{\boldsymbol{r}}_{\mathcal{S}_t}^{(t)}\|_1 = \sum_{i=1}^{|\mathcal{S}_t|}\tilde{r}_{u_i}^{(t)} \leq \sum_{i=1}^{|\mathcal{S}_t|}\tilde{r}_{u_i}^{(t+\Delta_i)} \leq 2\|\tilde{\boldsymbol{r}}_{\mathcal{S}_t}^{(t)}\|_1.$$

We reach the following lower and upper bounds of $\gamma_t$, $\frac{\|\tilde{\boldsymbol{r}}_{\mathcal{S}_t}^{(t)}\|_1}{\|\tilde{\boldsymbol{r}}^{(t)}\|_1} \leq \gamma_t := \sum_{i=1}^{|\mathcal{S}_t|} \frac{\tilde{r}_{u_i}^{(t+\Delta_i)}}{\|\tilde{\boldsymbol{r}}^{(t)}\|_1} \leq \frac{2\|\tilde{\boldsymbol{r}}_{\mathcal{S}_t}^{(t)}\|_1}{\|\tilde{\boldsymbol{r}}^{(t)}\|_1}$. To check the upper bound of $T$, from Equ. (17), $\|\tilde{\boldsymbol{r}}^{(T)}\|_1 = \prod_{t=0}^{T-1}\left(1 - \frac{2\alpha\omega\gamma_t}{1+\alpha}\right)\|\tilde{\boldsymbol{r}}^{(0)}\|_1$ and

$$\ln\frac{\|\tilde{\boldsymbol{r}}^{(T)}\|_1}{\|\tilde{\boldsymbol{r}}^{(0)}\|_1} = \sum_{t=0}^{T-1}\ln\left(1 - \frac{2\alpha\omega\gamma_t}{1+\alpha}\right) \leq -\sum_{t=0}^{T-1}\frac{2\alpha\omega\gamma_t}{1+\alpha} \quad\Rightarrow\quad T \leq \frac{(1+\alpha)}{2\alpha\omega\overline{\gamma}_T}\ln\frac{\|\tilde{\boldsymbol{r}}^{(0)}\|_1}{\|\tilde{\boldsymbol{r}}^{(T)}\|_1},$$

where the first inequality is due to $\ln(1+x) \leq x$ for $x > -1$. $\qquad\square$

**Theorem 3.3** (Runtime bound of LOCSOR ($\omega = 1$)). *Given the configuration $\theta = (\alpha, \epsilon, s, \mathcal{G})$ with $\alpha \in (0,1)$ and $\epsilon \leq 1/d_s$ and let $\boldsymbol{r}^{(T)}$ and $\boldsymbol{x}^{(T)}$ be returned by LOCSOR defined in (8) for solving Equ. (3). There exists a real implementation of (8) such that the runtime $\mathcal{T}_{\text{LocSOR}}$ is bounded by*

$$\frac{1+\alpha}{2} \cdot \frac{\overline{\text{vol}}(\mathcal{S}_T)}{\alpha\overline{\gamma}_T}\left(1 - \frac{\|\boldsymbol{D}^{1/2}\boldsymbol{r}^{(T)}\|_1}{\|\boldsymbol{D}^{1/2}\boldsymbol{r}^{(0)}\|_1}\right) \leq \mathcal{T}_{\text{LocSOR}} \leq \frac{1+\alpha}{2}\cdot\min\left\{\frac{1}{\alpha\epsilon}, \frac{\overline{\text{vol}}(\mathcal{S}_T)}{\alpha\overline{\gamma}_T}\ln\frac{C}{\epsilon}\right\}$$

*where $\overline{\text{vol}}(\mathcal{S}_T)$ and $\overline{\gamma}_T$ are defined in (6) and $C = \frac{1+\alpha}{(1-\alpha)|\mathcal{I}_T|}$ with $\mathcal{I}_T = \text{supp}(\boldsymbol{r}^{(T)})$. Furthermore, $\overline{\text{vol}}(\mathcal{S}_T)/\overline{\gamma}_T \leq 1/\epsilon$ and the local estimate $\hat{\boldsymbol{\pi}} := \boldsymbol{D}^{1/2}\boldsymbol{x}^{(T)}$ satisfies $\|\boldsymbol{D}^{-1}(\hat{\boldsymbol{\pi}} - \boldsymbol{\pi})\|_\infty \leq \epsilon$.*

*Proof.* After the last iteration $T$, for each nonzero residual $\tilde{r}_u^{(T)} \neq 0, u \in \mathcal{I}_T$, there must be at least one update that happened at node $u$: Node $u$ has a neighbor $v_u \in \mathcal{N}(u)$, which was active. This neighbor $v_u$ pushed some residual $\frac{(1-\alpha)\tilde{r}_{v_u}^{(t')}}{(1+\alpha)d_{v_u}}$ to $u$ where $t' < T$. Hence, for all $u \in \mathcal{I}_T$, we have

$$\|\tilde{\boldsymbol{r}}^{(T)}\|_1 = \sum_{u\in\mathcal{I}_T}\tilde{r}_u^{(T)} \geq \sum_{u\in\mathcal{I}_T}\frac{(1-\alpha)\tilde{r}_{v_u}^{(t')}}{(1+\alpha)d_{v_u}} \geq \sum_{u\in\mathcal{I}_T}\frac{(1-\alpha)2\alpha\epsilon d_{v_u}/(1+\alpha)}{(1+\alpha)d_{v_u}} = \epsilon|\mathcal{I}_T|\frac{2\alpha(1-\alpha)}{(1+\alpha)^2},$$

where the second inequality is because $\tilde{r}_{v_u}^{(t')}$ was active before the push operation. Applying the above lower bound of $\|\tilde{\boldsymbol{r}}^{(T)}\|_1$ to Equ. (16) of Lemma B.3 and note $\|\tilde{\boldsymbol{r}}^{(0)}\|_1 = 2\alpha/(1+\alpha)$, we obtain

$$\frac{\|\tilde{\boldsymbol{r}}^{(0)}\|_1}{\|\tilde{\boldsymbol{r}}^{(T)}\|_1} \leq \frac{\|\tilde{\boldsymbol{r}}^{(0)}\|_1}{\epsilon|\mathcal{I}_T|\cdot\frac{2\alpha(1-\alpha)}{(1+\alpha)^2}} = \frac{1+\alpha}{\epsilon(1-\alpha)|\mathcal{I}_T|} := \frac{C_1}{\epsilon}.$$

The rest is to prove an upper bound $1/(\alpha\epsilon)$ of $\mathcal{T}_{\text{LocSOR}}$. Recall that for any active node $u$, we have residual updates from Algo. 3 as the following

$$\boldsymbol{D}^{1/2}\boldsymbol{r}^{(t+1)} = \boldsymbol{D}^{1/2}\boldsymbol{r}^{(t)} - \omega r_u^{(t)}\boldsymbol{D}^{1/2}\boldsymbol{e}_u + \frac{(1-\alpha)\omega r_u^{(t)}}{1+\alpha}\boldsymbol{A}\boldsymbol{D}^{-1}\boldsymbol{D}^{1/2}\boldsymbol{e}_u.$$

Move $-\omega r_u^{(t)}\boldsymbol{D}^{1/2}\boldsymbol{e}_u$ to the left and note $\|\boldsymbol{A}\boldsymbol{D}^{-1}\boldsymbol{D}^{1/2}\boldsymbol{e}_u\|_1 = \sqrt{d_u}$, we then obtain

$$\|\boldsymbol{D}^{1/2}\boldsymbol{r}^{(t+1)}\|_1 + \omega\sqrt{d_u}r_u^{(t)} = \|\boldsymbol{D}^{1/2}\boldsymbol{r}^{(t)}\|_1 + \frac{(1-\alpha)\omega}{1+\alpha}\sqrt{d_u}r_u^{(t)}.$$

Hence, for each active $u$, we have $\frac{2\alpha\omega\sqrt{d_u}r_u^{(t)}}{1+\alpha} = \|\boldsymbol{D}^{1/2}\boldsymbol{r}^{(t)}\|_1 - \|\boldsymbol{D}^{1/2}\boldsymbol{r}^{(t+1)}\|_1$. Summing them over all active nodes $u$ and noticing $r_u^{(t)} \geq 2\alpha\epsilon\sqrt{d_u}/(1+\alpha)$ by the active condition. Note $\omega = 1$ and $\|\boldsymbol{D}^{1/2}\boldsymbol{r}^{(0)}\|_1 = \frac{2\alpha}{1+\alpha}$, we have run time bounded by

$$\mathcal{T}_{\text{LocSOR}} = \sum_u d_u \leq \left(\frac{1+\alpha}{2\alpha}\right)^2\frac{\sum_t(\|\boldsymbol{D}^{1/2}\boldsymbol{r}^{(t)}\|_1 - \|\boldsymbol{D}^{1/2}\boldsymbol{r}^{(t+1)}\|_1)}{\omega\epsilon} \leq \frac{(1+\alpha)}{2\alpha\epsilon}.$$

Combining the above bound and the bound $T$ shown in Lemma B.3, we prove the lower and upper bound of $\mathcal{T}_{LocSOR}$. To check the lower bound of $1/\epsilon$, i.e., $\overline{\text{vol}}(\mathcal{S}_T)/\overline{\gamma}_T \leq 1/\epsilon$, note $\frac{2\alpha\epsilon d_{u_i}}{1+\alpha} \leq \tilde{r}_{u_i}^{(t+\Delta_i)}$ for all $i = 1, 2, \ldots, |\mathcal{S}_t|$. Then we have

$$\frac{2\alpha\epsilon}{1+\alpha}\text{vol}(\mathcal{S}_t) \leq \sum_{i=1}^{|\mathcal{S}_t|}\tilde{r}_{u_i}^{(t+\Delta_i)}$$
$$= \gamma_t\|\boldsymbol{D}^{1/2}\boldsymbol{r}^{(t)}\|_1$$
$$\leq \gamma_t\|\boldsymbol{D}^{1/2}\boldsymbol{r}^{(0)}\|_1 = \frac{2\alpha\gamma_t}{1+\alpha},$$

where the last inequality is due to the monotonic decreasing of $\|\boldsymbol{D}^{1/2}\boldsymbol{r}^{(t)}\|_1$, i.e., $\frac{2\alpha}{1+\alpha} \geq \|\boldsymbol{D}^{1/2}\boldsymbol{r}^{(0)}\|_1 \geq \cdots \geq \|\boldsymbol{r}^{(T)}\|_1$. Applying the above inequality over all $t = 0, 1, 2 \ldots, T-1$, it leads to

$$\epsilon \operatorname{vol}(\mathcal{S}_t) \leq \gamma_t$$
$$\Rightarrow \quad \epsilon \sum_{t=0}^{T-1} \operatorname{vol}(\mathcal{S}_t) \leq \sum_{t=0}^{T-1} \gamma_t$$
$$\Rightarrow \quad \frac{\overline{\operatorname{vol}}(\mathcal{S}_T)}{\overline{\gamma}_T} \leq \frac{1}{\epsilon}.$$

$\square$

---

**Algo. 3** LocSOR$(\alpha, \epsilon, s, \mathcal{G}, \omega)$ via FIFO Queue

---
1: Initialize: $\boldsymbol{r} \leftarrow c\boldsymbol{e}_s,\ \boldsymbol{x} \leftarrow \boldsymbol{0},\ c = \frac{2\alpha}{1+\alpha},\ t = -1$
2: $\mathcal{Q} \leftarrow \{*, s\}$ // As we assume $\epsilon \leq 1/d_s$
3: **while true do**
4:     $u \leftarrow \mathcal{Q}.\text{dequeue}()$
5:     **if** u == * **then**
6:         **if** $\mathcal{Q} = \emptyset$ **then**
7:             **break**
8:         $t \leftarrow t + 1$       // Starting time of $\mathcal{S}_t$
9:         $\mathcal{Q}.\text{enqueue}(*)$    // Marker for next $\mathcal{S}_{t+1}$
10:        **continue**
11:     $\tilde{r} \leftarrow r_u$
12:     **if** $|r_u| < c \cdot \epsilon d_u$ **then**
13:        **continue**
14:     $x_u \leftarrow x_u + \omega \cdot \tilde{r}$
15:     $r_u \leftarrow r_u - \omega \cdot \tilde{r}$
16:     **for** $v \in \mathcal{N}(u)$ **do**
17:        $r_v \leftarrow r_v + \frac{(1-\alpha)\omega}{(1+\alpha)} \cdot \frac{\tilde{r}}{d_u}$
18:        **if** $|r_v| \geq c \cdot \epsilon d_v$ **and** $v \notin \mathcal{Q}$ **then**
19:           $\mathcal{Q}.\text{enqueue}(v)$
20:     **if** $|r_u| \geq c \cdot \epsilon d_u$ **and** $u \notin \mathcal{Q}$ **then**
21:        $\mathcal{Q}.\text{enqueue}(u)$
22: **return** $\boldsymbol{x}$

---

The real queue-based implementation of LocSOR is presented in Algo. 3. It has monotonic and nonnegative properties during the updates of $\boldsymbol{r} \geq \boldsymbol{0}$ and $\boldsymbol{x} \geq \boldsymbol{0}$ when $\omega \in (0, 1]$. Same as APPR, the operations of $\mathcal{Q}.\text{enqueue}(u)$, $\mathcal{Q}.\text{dequeue}()$, and $v \notin \mathcal{Q}$ are all in $\mathcal{O}(1)$. During the updates, one should note that the real vector $\boldsymbol{r}$ presents $\boldsymbol{D}^{1/2}\boldsymbol{r}^{(t)}$ while the vector $\boldsymbol{x}$ is $\boldsymbol{D}^{1/2}\boldsymbol{x}^{(t)}$. In this case, the original active node condition is implicitly shifting from $|r_u| \geq \frac{2\alpha\epsilon\sqrt{d_u}}{1+\alpha}$ to $\sqrt{d_u}|r_u| \geq \frac{2\alpha\epsilon d_u}{1+\alpha}$. We use this shifted active condition in Lines 11 and 13 and inactive condition in Line 5. When $\omega \in (1, 2)$, it is possible $|r_u| < c \cdot \epsilon d_u$ and LocSOR will ignore this inactive node $u$ during the updates. This step makes sure $\mathcal{S}_t = \{u_i : |r_{u_i}^{(t+\Delta_i)}| \geq \frac{2\alpha\epsilon\sqrt{d_u}}{1+\alpha}\}$ during the updates.

## B.3 Optimal GS-SOR and Proof of Corollary 3.4

We introduce the following standard result.

**Lemma B.4** (Young [55], Section 12.2, Theorem 2.1 ). *Given the GS-SOR method for solving $\boldsymbol{Q}\boldsymbol{x} = \boldsymbol{b}$, if the underlying matrix $\boldsymbol{Q}$ is a Stieltjes matrix and set relaxation parameter $\omega$ as*

$$\omega^* = \frac{2}{1 + \sqrt{1 - \rho(\boldsymbol{B})^2}} = 1 + \left(\frac{\rho(\boldsymbol{B})}{1 + \sqrt{1 - \rho(\boldsymbol{B})^2}}\right)^2, \tag{18}$$

*where $\rho(\boldsymbol{B})$ is the largest eigenvalue (in magnitude) of $\boldsymbol{B} = \boldsymbol{I} - \operatorname{diag}(\boldsymbol{Q})^{-1}\boldsymbol{Q}$, then*

$$\omega^* - 1 \leq \rho(\boldsymbol{L}_{\omega^*}) \leq \sqrt{\omega^* - 1}, \tag{19}$$

*where $\boldsymbol{L}_\omega := (\operatorname{diag}(\boldsymbol{Q}) - \omega\boldsymbol{Q}_L)^{-1}(\omega\boldsymbol{Q}_U - (\omega - 1)\operatorname{diag}(\boldsymbol{Q}))$ with $\boldsymbol{Q} = \operatorname{diag}(\boldsymbol{Q}) - \boldsymbol{Q}_U - \boldsymbol{Q}_L$.*

**Corollary 3.4.** *Let $\omega = \omega^* \triangleq 2/(1 + \sqrt{1 - (1-\alpha)^2/(1+\alpha)^2})$ and $\mathcal{S}_t = \mathcal{V}, \forall t \geq 0$, the global version of* LocSOR *has the following convergence bound*

$$\|\boldsymbol{r}^{(t)}\|_2 \leq \frac{2}{(1+\alpha)\sqrt{d_s}} \left(\frac{1-\sqrt{\alpha}}{1+\sqrt{\alpha}} + \epsilon_t\right)^t, \tag{20}$$

*where $\epsilon_t$ are small positive numbers with $\lim_{t\to\infty} \epsilon_t = 0$.*

*Proof.* Recall $\boldsymbol{Q} = \boldsymbol{I} - \frac{1-\alpha}{1+\alpha}\boldsymbol{D}^{-1/2}\boldsymbol{A}\boldsymbol{D}^{-1/2}$ and we consider the underlying graph as simple which means $\boldsymbol{A}$ has 0 diagonal. Hence, $\mathbf{diag}(\boldsymbol{Q}) = \boldsymbol{I}$ and $\boldsymbol{B}$ is defined as

$$\boldsymbol{B} = \frac{1-\alpha}{1+\alpha}\boldsymbol{D}^{-1/2}\boldsymbol{A}\boldsymbol{D}^{-1/2}, \quad \rho(\boldsymbol{B}) = \frac{1-\alpha}{1+\alpha}.$$

Since $\boldsymbol{Q}$ is a Stieltjes matrix, then Lemma B.4 gives a bound on the spectral radius of $\boldsymbol{L}_\omega$ as

$$\rho(\boldsymbol{L}_{\omega^*}) \leq \left(\frac{2}{1+\sqrt{1-\rho(\boldsymbol{B})^2}} - 1\right)^{1/2} = \left(\frac{2(1+\alpha)}{1+\alpha+2\sqrt{\alpha}} - 1\right)^{1/2} = \frac{1-\sqrt{\alpha}}{1+\sqrt{\alpha}}.$$

Recall that Gelfand's formula states [52]: Given spectral radius $\rho(\boldsymbol{L}_{\omega^*}) := \max_{i\in[n]}|\lambda_i(\boldsymbol{L}_{\omega^*})|$, where $\lambda_i(\cdot)$ is the $i$-th eigenvalue, there exists a sequence $\{\epsilon_t\}_{\geq 0}$ such that $\|\boldsymbol{L}_{\omega^*}^t\|_2 = (\rho(\boldsymbol{L}_{\omega^*}) + \epsilon_t)^t$ and $\lim_{t\to\infty}\epsilon_t = 0$. The standard SOR method is defined as the following

$$\boldsymbol{x}^{(t+1)} = (\mathbf{diag}(\boldsymbol{Q}) - \omega^*\boldsymbol{Q}_L)^{-1}\left(\omega^*\boldsymbol{b} + (\omega^*\boldsymbol{Q}_U - (\omega^* - 1)\mathbf{diag}(\boldsymbol{Q}))\boldsymbol{x}^{(t)}\right)$$

$$= \boldsymbol{L}_{\omega^*}\boldsymbol{x}^{(t)} + \omega^*(\mathbf{diag}(\boldsymbol{Q}) - \omega^*\boldsymbol{Q}_L)^{-1}\boldsymbol{b},$$

Note $\boldsymbol{r}^{(t)} = \boldsymbol{Q}\boldsymbol{e}^{(t)} = \boldsymbol{Q}\boldsymbol{L}_{\omega^*}^t\boldsymbol{Q}^{-1}\boldsymbol{Q}\boldsymbol{e}^{(0)} = \boldsymbol{Q}\boldsymbol{L}_{\omega^*}^t\boldsymbol{Q}^{-1}\boldsymbol{r}^{(0)}$. We have

$$\|\boldsymbol{r}^{(t)}\|_2 = \|\boldsymbol{Q}\boldsymbol{L}_{\omega^*}^t\boldsymbol{Q}^{-1}\boldsymbol{r}^{(0)}\|_2 \leq \|\boldsymbol{Q}\|_2\|\boldsymbol{L}_{\omega^*}^t\|_2\|\boldsymbol{Q}^{-1}\|_2\|\boldsymbol{r}^{(0)}\|_2$$

$$\leq \frac{2}{1+\alpha}\cdot\|\boldsymbol{L}_{\omega^*}^t\|_2\cdot\frac{1+\alpha}{2\alpha}\cdot\frac{2\alpha}{(1+\alpha)\sqrt{d_s}} = \frac{2\|\boldsymbol{L}_{\omega^*}^t\|_2}{(1+\alpha)\sqrt{d_s}}.$$

$\square$

To meet the stop condition, we require $|r_u^{(t)}| \leq \frac{2\alpha\epsilon\sqrt{d_u}}{1+\alpha}$. It is enough to make sure $\|\boldsymbol{r}^{(t)}\|_2 \leq \frac{2\alpha\epsilon}{(1+\alpha)\sqrt{d_s}}$. This leads to find $t$ such that

$$\|\boldsymbol{r}^{(t)}\|_2 \leq \frac{2\|\boldsymbol{L}_{\omega^*}^t\|_2}{(1+\alpha)\sqrt{d_s}} \leq \frac{2\alpha\epsilon}{(1+\alpha)\sqrt{d_s}} \Leftrightarrow \left(\frac{1-\sqrt{\alpha}}{1+\sqrt{\alpha}} + \epsilon_t\right)^t \leq \alpha\epsilon.$$

When $\epsilon_t = o(\sqrt{\alpha})$, then the runtime of global LOCSOR is $\tilde{\mathcal{O}}(m/\sqrt{\alpha})$ where $\tilde{\mathcal{O}}$ hides $\log\frac{1}{\epsilon}$.

## B.4 LOCGD and Proof of Theorem 3.5

The *local* gradient descent, namely LOCGD is to use $\boldsymbol{x}^{(t+1)} = \boldsymbol{x}^{(t)} + \boldsymbol{r}_{\mathcal{S}_t}^{(t)}$ and $\boldsymbol{r}^{(t+1)} = \boldsymbol{r}^{(t)} - \boldsymbol{Q}\boldsymbol{r}_{\mathcal{S}_t}^{(t)}$, where $\mathcal{S}_t = \{u_i : |r_{u_i}^{(t+\Delta_i)}| \geq 2\alpha\epsilon\sqrt{d_u}/(1+\alpha)\}$ where $\Delta_i = 0$. Algo. 4 presents our actual implementation of LOCGD via FIFO Queue.

---

**Algo. 4** LOCGD$(\alpha, \epsilon, s, \mathcal{G})$ via FIFO Queue

---

1: Initialize: $\boldsymbol{r} \leftarrow c\boldsymbol{e}_s$, $\boldsymbol{x} \leftarrow \boldsymbol{0}$, $c = \frac{2\alpha}{1+\alpha}$
2: $\mathcal{Q} \leftarrow \{s\}$ // Assume $\epsilon \leq \frac{1}{d_s}$
3: $t = 0$
4: **while** $\mathcal{Q} \neq \emptyset$ **do**
5:    $\mathcal{S}_t \leftarrow []$
6:    **while** $\mathcal{Q} \neq \emptyset$ **do**
7:       $u \leftarrow \mathcal{Q}.\text{dequeue}()$
8:       $\mathcal{S}_t.\text{append}((u, r_u))$
9:       $x_u \leftarrow x_u + r_u$
10:      $r_u \leftarrow 0$
11:    **for** $(u, \tilde{r}) \in \mathcal{S}_t$ **do**
12:      **for** $v \in \mathcal{N}(u)$ **do**
13:         $r_v \leftarrow r_v + \frac{(1-\alpha)\tilde{r}}{(1+\alpha)d_u}$
14:         **if** $|r_v| \geq c \cdot \epsilon d_v$ and $v \notin \mathcal{Q}$ **then**
15:            $\mathcal{Q}.\text{enqueue}(v)$
16:    $t \leftarrow t + 1$
17: **return** $\boldsymbol{x}, \boldsymbol{r}$

---

Algo. 4 presents LOCGD similar to the real queue-based implementation of LOCSOR. It has monotonic and nonnegative properties during the updates of $\boldsymbol{r} \geq \boldsymbol{0}$ and $\boldsymbol{x} \geq \boldsymbol{0}$. Again, the operations of $\mathcal{Q}.\text{enqueue}(u)$, $\mathcal{Q}.\text{dequeue}()$, and $v \notin \mathcal{Q}$ are all in $\mathcal{O}(1)$. During the updates, one should note that $\boldsymbol{r}$ presents $\boldsymbol{D}^{1/2}\boldsymbol{r}^{(t)}$ while $\boldsymbol{x}$ is $\boldsymbol{D}^{1/2}\boldsymbol{x}^{(t)}$. All shifted conditions are the same as of LOCSOR. The key advantage of LOCGD is that it is highly parallelizable, while LOCSOR is truly an online update, so it is hard to parallelize.

**Lemma B.5** (Iterations of LocGD). *With the initial $\boldsymbol{x}^{(0)} = \boldsymbol{0}, \boldsymbol{r}^{(0)} = \boldsymbol{b}, \mathcal{S}_0 = \text{supp}(\boldsymbol{r}^{(0)})$, denote $\tilde{\boldsymbol{r}}^{(t)} = \boldsymbol{D}^{1/2}\boldsymbol{r}^{(t)}$. LocGD defined in (9) has the following properties: 1) $\boldsymbol{x}^{(t)} \geq 0$, $\boldsymbol{r}^{(t)} \geq 0$ and $\|\boldsymbol{r}^{(t)}\| \geq \|\boldsymbol{r}^{(t+1)}\|_1$; 2) The residual and estimation error satisfies*

$$\|\tilde{\boldsymbol{r}}^{(t+1)}\|_1 = \left(1 - \frac{2\alpha\gamma_t}{1+\alpha}\right)\|\tilde{\boldsymbol{r}}^{(t)}\|_1, \quad \gamma_t = \sum_{i=1}^{|\mathcal{S}_t|} \frac{\tilde{r}_{u_i}^{(t+\Delta_i)}}{\|\tilde{\boldsymbol{r}}^{(t)}\|_1}, \text{ where } \Delta_i = 0.$$

*Proof.* We first show $\boldsymbol{x}^{(t)} \geq \boldsymbol{0}, \boldsymbol{r}^{(t)} \geq \boldsymbol{0}$ are all nonnegative vectors during the updates when $\boldsymbol{b} \geq 0$. This can be seen from the induction. At the initial stage, $\boldsymbol{x}^{(0)} \geq \boldsymbol{0}$ and $\boldsymbol{r}^{(0)} = \boldsymbol{b} \geq 0$. Now assume that for any $t \geq 0$, $\boldsymbol{x}^{(t)} \geq 0$ and $\boldsymbol{r}^{(t)} \geq 0$. Then $\boldsymbol{x}^{(t+1)} = \boldsymbol{x}^{(t)} + \boldsymbol{r}_{\mathcal{S}_k}^{(t)} \geq 0$, and $\boldsymbol{r}^{(t+1)} = \boldsymbol{r}_{\overline{\mathcal{S}_t}}^{(t)} + \frac{1-\alpha}{1+\alpha}\boldsymbol{D}^{-1/2}\boldsymbol{A}\boldsymbol{D}^{-1/2}\boldsymbol{r}_{\mathcal{S}_t}^{(t)} \geq 0$. Therefore, $\boldsymbol{x}^{(t)} \geq 0$ and $\boldsymbol{r}^{(t)} \geq 0$ for all $t$. Note $\tilde{\boldsymbol{r}}^{(t+1)} = \tilde{\boldsymbol{r}}_{\overline{\mathcal{S}_t}}^{(t)} + \frac{1-\alpha}{1+\alpha}\boldsymbol{A}\boldsymbol{D}^{-1} \cdot \tilde{\boldsymbol{r}}_{\mathcal{S}_t}^{(t)}$ and since $\|\boldsymbol{A}\boldsymbol{D}^{-1}\tilde{\boldsymbol{r}}_{\mathcal{S}_t}^{(t)}\|_1 = \|\tilde{\boldsymbol{r}}_{\mathcal{S}_t}^{(t)}\|_1$, we will have

$$\|\tilde{\boldsymbol{r}}^{(t+1)}\|_1 = \left(1 - \frac{2\alpha}{1+\alpha}\frac{\|\tilde{\boldsymbol{r}}_{\mathcal{S}_t}^{(t)}\|_1}{\|\tilde{\boldsymbol{r}}^{(t)}\|_1}\right)\|\tilde{\boldsymbol{r}}^{(t)}\|_1, \text{ where } \gamma_t := \|\tilde{\boldsymbol{r}}_{\mathcal{S}_t}^{(t)}\|_1/\|\tilde{\boldsymbol{r}}^{(t)}\|_1.$$

$\square$

Then, we can bound the total residual as the following theorem.

**Theorem 3.5** (Runtime bound of LocGD). *Given the configuration $\theta = (\alpha, \epsilon, s, \mathcal{G})$ with $\alpha \in (0, 1)$ and $\epsilon \leq 1/d_s$ and let $\boldsymbol{r}^{(T)}$ and $\boldsymbol{x}^{(T)}$ be returned by LocGD defined in (9) for solving Equ. (4). There exists a real implementation of (9) such that the runtime $\mathcal{T}_{\text{LocGD}}$ is bounded by*

$$\frac{1+\alpha}{2} \cdot \frac{\overline{\text{vol}}(\mathcal{S}_T)}{\alpha\overline{\gamma}_T}\left(1 - \frac{\|\tilde{\boldsymbol{r}}^{(T)}\|_1}{\|\tilde{\boldsymbol{r}}^{(0)}\|_1}\right) \leq \mathcal{T}_{\text{LocGD}} \leq \frac{1+\alpha}{2} \cdot \min\left\{\frac{1}{\alpha\epsilon}, \frac{\overline{\text{vol}}(\mathcal{S}_T)}{\alpha\overline{\gamma}_T}\ln\frac{C}{\epsilon}\right\},$$

*where $C = (1 + \alpha)/((1 - \alpha)|\mathcal{I}_T|), \mathcal{I}_T = \text{supp}(\boldsymbol{r}^{(T)})$. Furthermore, $\overline{\text{vol}}(\mathcal{S}_T)/\overline{\gamma}_T \leq 1/\epsilon$ and the estimate $\hat{\boldsymbol{\pi}} := \boldsymbol{D}^{1/2}\boldsymbol{x}^{(T)}$ satisfies $\|\boldsymbol{D}^{-1}(\hat{\boldsymbol{\pi}} - \boldsymbol{\pi})\|_\infty \leq \epsilon$.*

*Proof.* We first show bound $1/(\alpha\epsilon)$. We first rearrange $\boldsymbol{r}^{(t+1)} = \boldsymbol{r}^{(t)} - \boldsymbol{Q}\boldsymbol{r}_{\mathcal{S}_t}^{(t)}$ into

$$\boldsymbol{D}^{1/2}\boldsymbol{r}^{(t+1)} + \boldsymbol{D}^{1/2}\boldsymbol{r}_{\mathcal{S}_t}^{(t)} = \boldsymbol{D}^{1/2}\boldsymbol{r}^{(t)} + \frac{1-\alpha}{1+\alpha}\boldsymbol{A}\boldsymbol{D}^{-1}\boldsymbol{D}^{1/2}\boldsymbol{r}_{\mathcal{S}_t}^{(t)}.$$

Note $\boldsymbol{r}^{(t)} \geq \boldsymbol{0}$ and $\|\boldsymbol{A}\boldsymbol{D}^{-1}\boldsymbol{D}^{1/2}\boldsymbol{r}_{\mathcal{S}_t}^{(t)}\|_1 = \|\boldsymbol{D}^{1/2}\boldsymbol{r}_{\mathcal{S}_t}^{(t)}\|_1$. Hence, it leads to

$$\|\boldsymbol{D}^{1/2}\boldsymbol{r}_{\mathcal{S}_t}^{(t)}\|_1 = \frac{1+\alpha}{2\alpha}\left(\|\boldsymbol{D}^{1/2}\boldsymbol{r}^{(t)}\|_1 - \|\boldsymbol{D}^{1/2}\boldsymbol{r}^{(t+1)}\|_1\right).$$

At each local iterative $t$, by the active node condition $2\alpha\epsilon\sqrt{d_u}/(1+\alpha) \leq r_u^{(t)}$, we have

$$\epsilon\,\text{vol}(\mathcal{S}_t) = \sum_{u \in \mathcal{S}_t} \epsilon d_u \leq \sum_{u \in \mathcal{S}_t} \frac{(1+\alpha)\sqrt{d_u}r_u^{(t)}}{2\alpha} = \frac{1+\alpha}{2\alpha}\left\|\boldsymbol{D}^{1/2}\boldsymbol{r}_{\mathcal{S}_t}^{(t)}\right\|_1.$$

Then the total run time of LocGD presented in Algo. 4 is

$$\sum_{t=0}^{T-1} \text{vol}(\mathcal{S}_t) \leq \frac{1}{\epsilon}\left(\frac{1+\alpha}{2\alpha}\right)^2\left(\|\boldsymbol{D}^{1/2}\boldsymbol{r}^{(0)}\|_1 - \|\boldsymbol{D}^{1/2}\boldsymbol{r}^{(T)}\|_1\right) \leq \frac{1+\alpha}{2\alpha\epsilon}.$$

Therefore, the total run time is at most $\mathcal{T}_{\text{LocGD}} := \sum_{t=0}^{T-1} \text{vol}(\mathcal{S}_t) \leq \frac{1+\alpha}{2\alpha\epsilon}$. For estimating the bounds of $T$, by the Weierstrass product inequality [26] and Lemma B.5, we use the similar argument made in Lemma B.3 and continue to have

$$\frac{(1+\alpha)}{2\alpha\overline{\gamma}_T}\left(1 - \frac{\|\tilde{\boldsymbol{r}}^{(T)}\|_1}{\|\tilde{\boldsymbol{r}}^{(0)}\|_1}\right) \leq T \leq \frac{(1+\alpha)}{2\alpha\overline{\gamma}_T}\ln\frac{\|\tilde{\boldsymbol{r}}^{(0)}\|_1}{\|\tilde{\boldsymbol{r}}^{(T)}\|_1},$$

Note that each nonzero $\tilde{r}_u^{(T)}$ has at least part of the magnitude from the push operation of an active node, say $v_u$ at time $t' < T$. This means each nonzero of $\boldsymbol{D}^{1/2}\boldsymbol{r}^{(T)}$ satisfies

$$\tilde{r}_u^{(T)} \geq \frac{(1-\alpha)\tilde{r}_{v_u}^{(t')}}{(1+\alpha)d_{v_u}} \geq \frac{(1-\alpha)\cdot 2\alpha\epsilon d_{v_u}/(1+\alpha)}{(1+\alpha)d_{v_u}} = \frac{2\alpha(1-\alpha)\epsilon}{(1+\alpha)^2}, \text{ for } u \in \mathcal{I}_T.$$

Hence, we have $\|\tilde{\boldsymbol{r}}^{(T)}\|_1 \geq \frac{2\alpha(1-\alpha)\epsilon|\mathcal{I}_T|}{(1+\alpha)^2}$ and $T$ is further bounded as

$$T \leq \frac{(1+\alpha)}{2\alpha\overline{\gamma}_T} \ln \frac{\|\tilde{\boldsymbol{r}}^{(0)}\|_1}{\frac{2\alpha(1-\alpha)\epsilon|\mathcal{I}_T|}{(1+\alpha)^2}} := \frac{(1+\alpha)}{2\alpha\overline{\gamma}_T} \ln \frac{C_T}{\epsilon}, \text{ where } C_T = \frac{(1+\alpha)}{(1-\alpha)|\mathcal{I}_T|}.$$

The lower bound of $1/\epsilon$, i.e., $\overline{\text{vol}}(\mathcal{S}_t)/\overline{\gamma}_T \leq 1/\epsilon$, directly follows a similar strategy of previous proof by noticing that $\|\boldsymbol{D}^{1/2}\boldsymbol{r}^{(t)}\|_1$ is monotonically decreasing. $\qquad\square$

*Remark* B.6. One may consider designing local methods based on Jacobi and Richardson's iterations. Indeed, these two methods have the same updates as standard GD. Recall the standard GD method to solve (4) is $\boldsymbol{x}^{(t+1)} = \boldsymbol{x}^{(t)} + \boldsymbol{r}^{(t)}, \boldsymbol{r}^{(t+1)} = \boldsymbol{r}^{(t)} - \boldsymbol{Q}\boldsymbol{r}^{(t)}$. The Richardson's iteration is $\boldsymbol{x}^{(t+1)} = (\boldsymbol{I} - \omega\boldsymbol{W})\boldsymbol{x}^{(t)} + \omega\boldsymbol{b}$, i.e., $\boldsymbol{x}^{(t+1)} = \boldsymbol{x}^{(t)} - \omega(\boldsymbol{W}\boldsymbol{x}^{(t)} - \boldsymbol{b})$. The optimal $\omega^* = 2/(\lambda_{\min} + \lambda_{\max})$ where $\lambda_{\min} = 2\alpha/(1+\alpha)$ and $\lambda_{\max} \leq 2/(1+\alpha)$. Hence one can choose $\omega = 1 \leq \omega^*$ [19]. It leads to $\boldsymbol{x}^{(t+1)} = \boldsymbol{x}^{(t)} + \boldsymbol{r}^{(t)}$. One can get the same result for the Jacobi method.

# C   Local Chebyshev Method - LOCCH

## C.1   Nonhomogeneous of Second-order Difference Equation

We begin by providing the solutions of the second-order nonhomogeneous equation as the following

**Lemma C.1** (Stević [47])**.** *The solution of the second-order nonhomogeneous difference equation*

$$x_{t+1} + px_t + qx_{t-1} = f_t, \quad t = 1, 2, \ldots \tag{21}$$

*is characterized by the following two cases*

$$x_t = \begin{cases} \frac{1}{\hat{\lambda}_2 - \hat{\lambda}_1}\left(\hat{\lambda}_1^t\left(\hat{\lambda}_2 x_0 - x_1 - \sum_{k=1}^t \frac{f_k}{\hat{\lambda}_1^k}\right) + \hat{\lambda}_2^t\left(x_1 - \hat{\lambda}_1 x_0 + \sum_{k=1}^t \frac{f_k}{\hat{\lambda}_2^k}\right)\right) & p^2 \neq 4q \\ \left(-\frac{p}{2}\right)^t\left(x_0 - \sum_{k=1}^t \frac{kf_k}{(-p/2)^{k+1}}\right) + t\left(-\frac{p}{2}\right)^{t-1}\left(x_1 - \left(-\frac{p}{2}\right)x_0 + \sum_{k=1}^t \frac{f_k}{(-p/2)^k}\right) & p^2 = 4q \end{cases},$$

*where $\hat{\lambda}_1, \hat{\lambda}_2$ are two roots of $\lambda^2 + p\lambda + q = 0$, and the summation follows convention $\sum_{k=1}^0 \cdot = 0$.*

Based on the above lemma, we have the following corollary

**Corollary C.2** (Second-order nonhomogeneous equation)**.** *Given $|a| \leq 1$, the second-order nonhomogeneous equation*

$$x_{t+1} - 2ax_t + x_{t-1} = f_t$$

*has the following solution*

$$x_t = \begin{cases} x_0 + t(x_1 - x_0) + \sum_{k=1}^t (t-k)f_k & \text{if } a = 1 \\ \frac{\sin(\theta t)x_1 - \sin(\theta(t-1))x_0}{\sin\theta} + \frac{\sum_{k=1}^t \sin(\theta(t-k))f_k}{\sin\theta} & \text{if } |a| < 1 \text{ where } \theta = \arccos(a) \\ (-1)^t(x_0 - t(x_0 + x_1)) + (-1)^t\left(\sum_{k=1}^t (-1)^{-k-1}(t-k)f_k\right) & \text{if } a = -1. \end{cases} \tag{22}$$

*Proof.* The first and last two cases can be directly followed from Lemma C.1. Since $\hat{\lambda}_1$ and $\hat{\lambda}_2$ are the two complex roots of $\lambda^2 - 2a\lambda + 1 = 0$, we write $\hat{\lambda}_1 = re^{i\theta} = r(\cos\theta + i\sin\theta)$ and $\hat{\lambda}_2 = re^{-i\theta} = r(\cos\theta - i\sin\theta)$. It indicates

$$\lambda^2 - 2a\lambda + 1 = \left(\lambda - re^{i\theta}\right)\left(\lambda - re^{-i\theta}\right) = \lambda^2 - r(e^{i\theta} + e^{-i\theta})\lambda + r^2 = 0.$$

Since $1 > a^2$, $\hat{\lambda}_1 = a - i\sqrt{1 - a^2}$, and $\mathbf{Re}(\hat{\lambda}_1)^2 + \mathbf{Im}(\hat{\lambda}_1)^2 = a^2 + (1 - a^2) = 1$, then $r = 1$. Then $\theta = \arccos(\mathbf{Re}(\hat{\lambda}_1)) = \arccos(a)$, and $\sin(\theta) = \mathbf{Im}(\hat{\lambda}_1) = \sqrt{1 - a^2}$. Finally, $\hat{\lambda}_1^t = e^{it\theta} = \cos(t\theta) + i\sin(t\theta)$, and

$$\hat{\lambda}_1 = \cos\theta + i\sin\theta, \quad \hat{\lambda}_2 = \cos\theta - i\sin\theta$$

$$\hat{\lambda}_1\hat{\lambda}_2 = e^{it\theta}e^{-it\theta} = 1$$

$$\hat{\lambda}_2 - \hat{\lambda}_1 = -2i\sin\theta$$

$$\hat{\lambda}_1^t = \cos(\theta t) + i\sin(\theta t), \hat{\lambda}_2^t = \cos(\theta t) - i\sin(\theta t)$$

$$\hat{\lambda}_2^t - \hat{\lambda}_1^t = e^{-it\theta} - e^{it\theta} = -2i\sin(\theta t).$$

$$\hat{\lambda}_1^{t-1} - \hat{\lambda}_2^{t-1} = 2i\sin(\theta(t - 1)) \overset{\hat{\lambda}_1\hat{\lambda}_2=1}{=} \hat{\lambda}_1^t\hat{\lambda}_2 - \hat{\lambda}_2^t\hat{\lambda}_1$$

Based on these, we have

$$x_t = \frac{1}{\hat{\lambda}_2 - \hat{\lambda}_1}\left(\hat{\lambda}_1^t\left(\hat{\lambda}_2 x_0 - x_1 - \sum_{k=1}^{t}\frac{f_k}{\hat{\lambda}_1^k}\right) + \hat{\lambda}_2^t\left(x_1 - \hat{\lambda}_1 x_0 + \sum_{k=1}^{t}\frac{f_k}{\hat{\lambda}_2^k}\right)\right)$$

$$= \frac{\hat{\lambda}_1^t(\hat{\lambda}_2 x_0 - x_1) + \hat{\lambda}_2^t(x_1 - \hat{\lambda}_1 x_0)}{\hat{\lambda}_2 - \hat{\lambda}_1} + \frac{1}{\hat{\lambda}_2 - \hat{\lambda}_1}\left(-\hat{\lambda}_1^t\sum_{k=1}^{t}\frac{f_k}{\hat{\lambda}_1^k} + \hat{\lambda}_2^t\sum_{k=1}^{t}\frac{f_k}{\hat{\lambda}_2^k}\right)$$

$$= \frac{(2i\sin(\theta(t - 1))x_0 - 2i\sin(\theta t)x_1)}{-2i\sin\theta} + \frac{1}{-2i\sin\theta}\left(-\hat{\lambda}_1^t\sum_{k=1}^{t}\frac{f_k}{\hat{\lambda}_1^k} + \hat{\lambda}_2^t\sum_{k=1}^{t}\frac{f_k}{\hat{\lambda}_2^k}\right)$$

$$= \frac{\sin(\theta t)x_1 - \sin(\theta(t - 1))x_0}{\sin\theta} + \frac{1}{-2i\sin\theta}\sum_{k=1}^{t}\left(\hat{\lambda}_2^{t-k} - \hat{\lambda}_1^{t-k}\right)f_k$$

$$= \frac{\sin(\theta t)x_1 - \sin(\theta(t - 1))x_0}{\sin\theta} + \frac{1}{-2i\sin\theta}\sum_{k=1}^{t} -2i\sin(\theta(t - k))f_k$$

$$= \frac{\sin(\theta t)x_1 - \sin(\theta(t - 1))x_0}{\sin\theta} + \frac{\sum_{k=1}^{t}\sin(\theta(t - k))f_k}{\sin\theta}.$$

$\square$

**Lemma C.3.** *For $|\lambda_i| \leq 1, i = 1, 2, \ldots, n$, equations $y_i^{(t+1)} - 2\lambda_i y_i^{(t)} + y_i^{(t-1)} = 0$ have the following solutions*

$$y_i^{(t)} = \begin{cases} y_i^{(0)} + t(y_i^{(1)} - y_i^{(0)}) & \text{if} \quad \lambda_i = 1 \\ \frac{\sin(\theta_i t)y_i^{(1)} - \sin(\theta_i(t-1))y_i^{(0)}}{\sin(\theta_i)} & \text{if} \quad |\lambda_i| < 1 \text{ where } \theta_i = \arccos(\lambda_i) \\ (-1)^t(z_{i,0} - t(y_i^{(0)} + y_i^{(1)})) & \text{if} \quad \lambda_i = -1. \end{cases} \tag{23}$$

*Furthermore, when $y_i^{(1)} = \lambda_i y_i^{(0)}$ for $i = 1, 2, \ldots, n$, then solutions can be simplified as*

$$y_i^{(t)} = \begin{cases} y_i^{(0)} & \text{if} \quad \lambda_i = 1 \\ y_i^{(0)}\cos(\theta_i t) & \text{if} \quad |\lambda_i| < 1 \text{ where } \theta_i = \arccos(\lambda_i) \\ z_{i,0}(-1)^t & \text{if} \quad \lambda_i = -1. \end{cases} \tag{24}$$

*Proof.* The first part is a consequence of Corollary C.2 by letting $f_t = 0$. To see the second identity of this lemma, note that

$$\frac{(\lambda_i \sin(\theta_i t) - \sin(\theta_i(t - 1)))}{\sin\theta_i} = \cos(\theta_i t).$$

Indeed, by expanding $\sin(\theta_i(t-1))$, we have
$$\sin(\theta_i(t-1)) = \sin(\theta_i t - \theta_i)$$
$$= \sin(\theta_i t)\cos(-\theta_i) + \cos(\theta_i t)\sin(-\theta_i)$$
$$= \sin(\theta_i t)\cos(\theta_i) - \cos(\theta_i t)\sin(\theta_i)$$
$$= \lambda_i \sin(\theta_i t) - \cos(\theta_i t)\sin(\theta_i)$$

Hence, when $y_i^{(1)} = \lambda_i y_i^{(0)}$, we have
$$\frac{\sin(\theta_i t)y_i^{(1)} - \sin(\theta_i(t-1))y_i^{(0)}}{\sin(\theta_i)} = \frac{(\lambda_i \sin(\theta_i t) - \sin(\theta_i(t-1)))\,y_i^{(0)}}{\sin\theta_i} = \cos(\theta_i t)y_i^{(0)}.$$

$\square$

**Lemma C.4.** *Given $t \geq 1, |\lambda_i| \leq 1$, the $n$ second-order difference equations*
$$y_i^{(t+1)} - 2\lambda_i y_i^{(t)} + y_i^{(t-1)} = h_{i,t}, \qquad i = 1, 2, \dots, n.$$
*have the following solutions*
$$y_i^{(t)} = \begin{cases} y_i^{(0)} + t(y_i^{(1)} - y_i^{(0)}) + \sum_{k=1}^{t-1}(t-k)h_{i,k} & \text{if} \quad \lambda_i = 1 \\ \frac{\sin(\theta_i t)y_i^{(1)} - \sin(\theta_i(t-1))y_i^{(0)}}{\sin(\theta_i)} + \sum_{k=1}^{t-1}\frac{\sin(\theta_i(t-k))}{\sin\theta_i}h_{i,k} & \text{if} \quad |\lambda_i| < 1 \text{ where } \theta_i = \arccos(\lambda_i) \\ (-1)^t(z_{i,0} - t(y_i^{(0)} + y_i^{(1)})) + \sum_{k=1}^{t-1}(-1)^{t-k-1}(t-k)h_{i,k} & \text{if} \quad \lambda_i = -1. \end{cases}$$
(25)

*Furthermore, with initial conditions $y_i^{(1)} = \lambda_i y_i^{(0)}$, $y_i^{(t)}$ can be simplified as*
$$y_i^{(t)} = \begin{cases} y_i^{(0)} + \sum_{k=1}^{t-1}(t-k)h_{i,k} & \text{if} \quad \lambda_i = 1 \\ \cos(\theta_i t)y_i^{(0)} + \sum_{k=1}^{t-1}\frac{\sin(\theta_i(t-k))}{\sin\theta_i}h_{i,k} & \text{if} \quad |\lambda_i| < 1 \text{ where } \theta_i = \arccos(\lambda_i) \\ (-1)^t y_i^{(0)} + \sum_{k=1}^{t-1}(-1)^{t-k-1}(t-k)h_{i,k} & \text{if} \quad \lambda_i = -1. \end{cases}$$
(26)

*Proof.* The first part is a consequence of Corollary C.2 by letting $f_t = h_{i,t}$. We prove the second part by considering three cases of $\lambda_i$ as

- **Case 1.** When $\lambda_i = 1$, we have $y_i^{(t+1)} - 2y_i^{(t)} + y_i^{(t-1)} = h_{i,t}$. For $t \geq 1$, the solution the above is
$$y_i^{(t)} = y_i^{(0)} + t(y_i^{(1)} - y_i^{(0)}) + \sum_{k=1}^{t-1}(t-k)h_{i,k} = y_i^{(0)} + \sum_{k=1}^{t-1}(t-k)h_{i,k},$$
where the second equation is due to $y_i^{(1)} = \lambda_1 y_i^{(0)} = y_i^{(0)}$.

- **Case 2.** When $\lambda_n = -1$ ($\mathcal{G}$ is a bipartite graph), we have $y_i^{(t+1)} + 2y_i^{(t)} + y_i^{(t-1)} = h_{i,t}$. For $t \geq 1$, the solution is
$$y_i^{(t)} = (-1)^t y_i^{(0)} + (-1)^t \sum_{k=1}^{t-1}\frac{(t-k)h_{i,k}}{(-1)^{k+1}} = (-1)^t y_i^{(0)} + \sum_{k=1}^{t-1}(-1)^{t-k-1}(t-k)h_{i,k}.$$

- **Case 3.** When $|\lambda_i| < 1$, and define $\theta_i = \arccos(\lambda_i)$. We use a similar argument in Lemma C.3 and continue to have
$$y_i^{(t)} = \frac{\sin(\theta_i t)y_i^{(1)} - \sin(\theta_i(t-1))y_i^{(0)}}{\sin\theta_i} + \frac{\sum_{k=1}^{t-1}\sin(\theta_i(t-k))h_{i,k}}{\sin\theta_i}$$
$$= \frac{(\lambda_i \sin(\theta_i t) - \sin(\theta_i(t-1)))\,y_i^{(0)}}{\sin\theta_i} + \frac{\sum_{k=1}^{t-1}\sin(\theta_i(t-k))h_{i,k}}{\sin\theta_i}$$
$$= \cos(\theta_i t)y_i^{(0)} + \sum_{k=1}^{t-1}\frac{\sin(\theta_i(t-k))}{\sin\theta_i}h_{i,k}.$$

$\square$

## C.2 Properties on Ratio of Chebyshev Polynomials

The next lemma presents the properties of Chebyshev polynomials.

**Lemma C.5** (Chebyshev polynomial bound). *For $t \geq 1$, the Chebyshev polynomial of the first kind is defined recursively as*

$$T_{t+1}(x) = 2xT_t(x) - T_{t-1}(x) \quad \text{with} \quad T_0(x) = 1, \quad T_1(x) = x.$$

*For $t \geq 1$, define $\delta_t = T_{t-1}(\frac{1+\alpha}{1-\alpha})/T_t(\frac{1+\alpha}{1-\alpha})$, then*

1. *$T_t(x = \frac{1+\alpha}{1-\alpha})$ and $\delta_t$ defines the following sequence*

$$\delta_{t+1} = \left( 2\frac{1+\alpha}{1-\alpha} - \delta_t \right)^{-1}, \text{ where } \delta_1 = \frac{1-\alpha}{1+\alpha}.$$

2. *The closed-form $\delta_{1:t}$ can be upper bounded as*

$$\delta_{1:t} = \frac{1}{T_t(\frac{1+\alpha}{1-\alpha})} = \frac{2}{\tilde{\alpha}^t + \tilde{\alpha}^{-t}} \leq 2\left( \frac{1-\sqrt{\alpha}}{1+\sqrt{\alpha}} \right)^t.$$

3. *Note $\delta_1 = T_0/T_1 = 1/x = \frac{1-\alpha}{1+\alpha}$, the sequence $\{\delta_t\}$ satisfies $\delta_t < 1, \forall t \geq 1$ and*

$$1 = 2\delta_{t+1}x - \delta_t\delta_{t+1}, \quad t = 1, 2, \ldots.$$

4. *Denote $\delta_{j:t} = \prod_{i=j}^t \delta_i$ for $t \geq j \geq 0$ and set the default value $\delta_{j:j-1} = 1$ for $j \geq 0$, then*

$$\delta_{1:t}/(\delta_{1:k}) = \delta_{k+1:t} \leq 2\tilde{\alpha}^{t-k}, \text{ for } t \geq k \geq 0.$$

*Proof.* For the first item, let $x = \frac{1+\alpha}{1-\alpha}$, use the Chebyshev equation, we have

$$1 = 2\left( \frac{1+\alpha}{1-\alpha} \right)\frac{T_t}{T_{t+1}} - \frac{T_{t-1}}{T_{t+1}} = 2\left( \frac{1+\alpha}{1-\alpha} \right)\delta_{t+1} - \delta_t\delta_{t+1} \quad \Rightarrow \quad \delta_{t+1}^{-1} = 2\left( \frac{1+\alpha}{1-\alpha} \right) - \delta_t.$$

For the second item, for all $t \geq 0$, if $\xi = \frac{x+x^{-1}}{2} \neq 0$, it is well known that the $T_t$ can be rewritten as

$$T_t\left( \xi = \frac{x+x^{-1}}{2} \right) = \frac{x^t + x^{-t}}{2}.$$

For our problem, recall we defined $\tilde{\alpha} = \frac{1-\sqrt{\alpha}}{1+\sqrt{\alpha}}$ and using $x = \frac{1+\alpha}{1-\alpha} \neq 0$, one can verify that

$$x = \frac{1+\alpha}{1-\alpha} = \frac{\tilde{\alpha} + \tilde{\alpha}^{-1}}{2} \iff T_t\left( x = \frac{1+\alpha}{1-\alpha} \right) = T_t\left( \frac{\tilde{\alpha} + \tilde{\alpha}^{-1}}{2} \right) = \frac{\tilde{\alpha}^t + \tilde{\alpha}^{-t}}{2}$$

and

$$\prod_{j=1}^t \delta_j = \frac{T_0}{T_1} \cdot \frac{T_1}{T_2} \cdot \frac{T_2}{T_3} \cdots \frac{T_{t-1}}{T_t} = \frac{1}{T_t} = \frac{2}{\tilde{\alpha}^t + \tilde{\alpha}^{-t}} = \frac{2\tilde{\alpha}^t}{\tilde{\alpha}^{2t} + 1} \leq 2\tilde{\alpha}^t = 2\left( \frac{1-\sqrt{\alpha}}{1+\sqrt{\alpha}} \right)^t.$$

For the third item, it is sufficient to show that $\delta_t \leq \frac{1}{x}$ for all $t \geq 1$, for $x = \frac{1+\alpha}{1-\alpha}$. This can be done recursively, since $\delta_1 = x$ and

$$\delta_{t+1} = \frac{1}{2x - \delta_t} \leq \frac{1}{2x - \frac{1}{x}} \overset{x \geq \frac{1}{x}}{\leq} \frac{1}{x}.$$

For the last item, note when $t \geq 1$ and $k \geq 1$, we have the following inequalities

$$\prod_{j=1}^t \delta_j \cdot \prod_{j=1}^k \delta_j^{-1} = \frac{2}{\tilde{\alpha}^t + \tilde{\alpha}^{-t}} \cdot \frac{\tilde{\alpha}^k + \tilde{\alpha}^{-k}}{2} = \frac{\tilde{\alpha}^k + \tilde{\alpha}^{-k}}{\tilde{\alpha}^t + \tilde{\alpha}^{-t}} = \tilde{\alpha}^{t-k}\frac{\tilde{\alpha}^{2k} + 1}{\tilde{\alpha}^{2t} + 1} \leq 2\tilde{\alpha}^{t-k},$$

where note $\frac{\tilde{\alpha}^{2k}+1}{\tilde{\alpha}^{2t}+1} \in [1, 2]$ for $t \geq k$. $\qquad\square$

## C.3 Standard Chebyshev (CH) Method and Proof of Theorem C.6

This subsection introduces the standard Chebyshev algorithm. Our following theorem is to prove the runtime complexity of the Chebyshev polynomial iteration for solving Equ. (3).

**Theorem C.6** (Standard CH). *For $t \geq 1$, consider the Chebyshev polynomials to solve Equ. (3) as*

$$\boldsymbol{x}^{(t+1)} = \boldsymbol{x}^{(t)} + (1 + \delta_{t:t+1})\boldsymbol{r}^{(t)} + \delta_{t:t+1}\big(\boldsymbol{x}^{(t)} - \boldsymbol{x}^{(t-1)}\big), \ \ \boldsymbol{r}^{(t+1)} = 2\delta_{t+1}\boldsymbol{W}\boldsymbol{r}^{(t)} - \delta_{t:t+1}\boldsymbol{r}^{(t-1)},$$

*where $\boldsymbol{x}^{(0)} = \boldsymbol{0}, \boldsymbol{x}^{(1)} = \boldsymbol{x}^{(0)} + \boldsymbol{r}^{(0)}$ and $\delta_{t+1} = \big(2\frac{1+\alpha}{1-\alpha} - \delta_t\big)^{-1}$ with $\delta_1 = \frac{1-\alpha}{1+\alpha}$. Assume $\epsilon < 1/d_s$, then the residual has the following convergence bound*

$$\left\|\boldsymbol{r}^{(t)}\right\|_2 \leq 2\left(\frac{1 - \sqrt{\alpha}}{1 + \sqrt{\alpha}}\right)^t \|\boldsymbol{b}\|_2.$$

*Let the estimate be $\hat{\boldsymbol{\pi}} = \boldsymbol{D}^{1/2}\boldsymbol{x}^{(t)}$, the the runtime of CH for reaching $\|\boldsymbol{D}^{-1/2}\boldsymbol{r}^{(t)}\|_\infty \leq \frac{2\alpha\epsilon}{1+\alpha}$ with $\|\boldsymbol{D}^{-1}(\hat{\boldsymbol{\pi}} - \boldsymbol{\pi})\|_\infty \leq \epsilon$ guarantee is at most*

$$\mathcal{T}_{\text{CH}} \leq \Theta\left(m\left\lceil\frac{1 + \sqrt{\alpha}}{2\sqrt{\alpha}}\ln\frac{2}{\epsilon}\right\rceil\right) = \tilde{\Theta}\left(\frac{m}{\sqrt{\alpha}}\right).$$

*Proof.* Recall eigendecomposition of $\boldsymbol{W} = \boldsymbol{V}\boldsymbol{\Lambda}\boldsymbol{V}^\top$ where $\boldsymbol{V} = [\boldsymbol{v}_1, \boldsymbol{v}_2, \ldots, \boldsymbol{v}_n]$ and each $\boldsymbol{v}_i$ is the eigenvector. For $t \geq 1$, the residual $\boldsymbol{r}^{(t)}$ can be written as $n$ second-order difference equations as

$$\boldsymbol{V}^\top\boldsymbol{r}^{(t+1)} - 2\delta_{t+1}\boldsymbol{\Lambda}\boldsymbol{V}^\top\boldsymbol{r}^{(t)} + \delta_{t:t+1}\boldsymbol{V}^\top\boldsymbol{r}^{(t-1)} = \boldsymbol{0},$$

where each $i$-th element-wise equation of the above can be written as the following

$$\boldsymbol{v}_i^\top\boldsymbol{r}^{(t+1)} - 2\delta_{t+1}\lambda_i\boldsymbol{v}_i^\top\boldsymbol{r}^{(t)} + \delta_{t+1}\delta_t\boldsymbol{v}_i^\top\boldsymbol{r}^{(t-1)} = 0, \qquad i = 1, 2, \ldots, n.$$

Define $\boldsymbol{V}^\top\boldsymbol{r}^{(t)} = \delta_{1:t}\boldsymbol{y}^{(t)}$. Each component $\boldsymbol{v}_i^\top\boldsymbol{r}^{(t)}$ is $\boldsymbol{v}_i^\top\boldsymbol{r}^{(t)} = \delta_{1:t}y_i^{(t)}$ where $\boldsymbol{v}_i^\top\boldsymbol{r}^{(0)} := y_i^{(0)}$ by default. The above can be rewritten

$$\delta_{1:t+1}y_i^{(t+1)} - 2\delta_{t+1}\delta_{1:t}\lambda_i y_i^{(t)} + \delta_{t+1}\delta_t\delta_{1:t-1}y_i^{(t-1)} = 0$$
$$\iff y_i^{(t+1)} - 2\lambda_i y_i^{(t)} + y_i^{(t-1)} = 0, \tag{27}$$

where $\iff$ follows from $\delta_{1:t+1} \neq 0$. Note $\boldsymbol{V}^\top\boldsymbol{r}^{(1)} = \frac{1-\alpha}{1+\alpha}\boldsymbol{V}^\top\boldsymbol{V}\boldsymbol{\Lambda}\boldsymbol{V}^\top\boldsymbol{r}^{(0)} = \delta_1\boldsymbol{\Lambda}\boldsymbol{V}^\top\boldsymbol{r}^{(0)}$, we have

$$\boldsymbol{V}^\top\boldsymbol{r}^{(0)} = \boldsymbol{y}^{(0)} \qquad \Rightarrow \qquad \boldsymbol{V}^\top\boldsymbol{r}^{(1)} = \delta_1\boldsymbol{y}^{(1)} = \delta_1\boldsymbol{\Lambda}\boldsymbol{V}^\top\boldsymbol{r}^{(0)} = \delta_1\boldsymbol{\Lambda}\boldsymbol{y}^{(0)},$$

where it follows from $\delta_1 \neq 0$. As $\boldsymbol{y}^{(1)} = \boldsymbol{\Lambda}\boldsymbol{y}^{(0)}$, follow Equ. (24) of Lemma C.3, $y_i^{(t)}$ has the solution

$$y_i^{(t)} = \begin{cases} y_i^{(0)}\cos(\theta_i t) & |\lambda_i| < 1 \\ y_i^{(0)}\lambda_i^t & |\lambda_i| = 1 \end{cases} \leq \begin{cases} |y_i^{(0)}| & |\lambda_i| < 1 \\ |y_i^{(0)}| & |\lambda_i| = 1 \end{cases},$$

where $\theta_i = \arccos(\lambda_i)$. We can write down $\boldsymbol{r}^{(t)}$ in terms of $\boldsymbol{y}^{(t)}$

$$\boldsymbol{V}^\top\boldsymbol{r}^{(t)} = \delta_{1:t}\boldsymbol{y}^{(t)} = \delta_{1:t}\boldsymbol{Z}_t\boldsymbol{y}^{(0)},$$

where $\boldsymbol{Z}_t$ has two possible forms

$$\boldsymbol{Z}_t = \begin{cases} \mathbf{diag}\,(1, \ldots, \cos(\theta_i t), \ldots, (-1)^t) & \text{for bipartite graphs} \\ \mathbf{diag}\,(1, \ldots, \cos(\theta_i t), \ldots, \cos(\theta_n t)) & \text{for non-bipartite graphs.} \end{cases}$$

Hence, $\|\boldsymbol{Z}_t\|_2 \leq 1$ and $\|\boldsymbol{Z}_t\boldsymbol{y}^{(0)}\|_2 \leq \|\boldsymbol{y}^{(0)}\|_2$. We have

$$\|\boldsymbol{r}^{(t)}\|_2 = \|\boldsymbol{V}^\top\boldsymbol{r}^{(t)}\|_2 \leq \delta_{1:t}\|\boldsymbol{y}^{(0)}\|_2 \leq 2\left(\frac{1 - \sqrt{\alpha}}{1 + \sqrt{\alpha}}\right)^t \|\boldsymbol{b}\|_2,$$

where the last inequality follows Lemma C.5 and note $\boldsymbol{z}^{(0)} = \boldsymbol{V}^\top \boldsymbol{r}^{(0)}$. To meet the stop condition of $\left\{ |r_u^{(t)}| < 2\alpha\epsilon\sqrt{d_u}/(1+\alpha), u \in \mathcal{V} \right\} = \emptyset$, it is sufficient to choose a minimal integer $t$ such that

$$2 \left( \frac{1 - \sqrt{\alpha}}{1 + \sqrt{\alpha}} \right)^t \|\boldsymbol{b}\|_2 < \frac{2\alpha\epsilon}{(1+\alpha)\sqrt{d_s}}.$$

To see this, if the above inequality is satisfied, then note for any node $u$, we have

$$|r_u^{(t)}| \leq \|\boldsymbol{r}^{(t)}\|_\infty \leq \|\boldsymbol{r}^{(t)}\|_2 \leq 2 \left( \frac{1 - \sqrt{\alpha}}{1 + \sqrt{\alpha}} \right)^t \|\boldsymbol{b}\|_2 < \frac{2\alpha\epsilon}{(1+\alpha)\sqrt{d_s}} \leq \frac{2\alpha\epsilon\sqrt{d_u}}{(1+\alpha)},$$

where all nodes are inactive. So, it gives $t \ln \left( \frac{1-\sqrt{\alpha}}{1+\sqrt{\alpha}} \right) \leq \ln \frac{\epsilon}{2}$ by noticing $\|\boldsymbol{b}\|_2 = 2\alpha/((1+\alpha)\sqrt{d_s})$.

It indicates $t \geq \left\lceil \ln \frac{2}{\epsilon} \middle/ \ln \left( \frac{1+\sqrt{\alpha}}{1-\sqrt{\alpha}} \right) \right\rceil$. As $\frac{1}{\ln(1+x)} \leq \frac{1+x}{x}$ for $x > 0$, it is sufficient to choose $t$

$$t = \left\lceil \frac{1 + \sqrt{\alpha}}{2\sqrt{\alpha}} \ln \frac{2}{\epsilon} \right\rceil.$$

$\square$

## C.4   Residual Updates of LOCCH and Proof of Lemma 4.1

We propose the following local Chebyshev iteration procedure

$$\boldsymbol{x}^{(t+1)} = \boldsymbol{x}^{(t)} + (1 + \delta_{t:t+1})\boldsymbol{r}_{\mathcal{S}_t}^{(t)} + \delta_{t:t+1}\left(\boldsymbol{x}^{(t)} - \boldsymbol{x}^{(t-1)}\right)_{\mathcal{S}_t}.$$

Our next Lemma is to expanding $\left(\boldsymbol{x}^{(t)} - \boldsymbol{x}^{(t-1)}\right)_{\mathcal{S}_t}$

**Lemma C.7.** *For $t \geq 1$ with initials $\boldsymbol{x}^{(0)} = \boldsymbol{0}$ and $\boldsymbol{x}^{(1)} = \boldsymbol{x}^{(0)} + \boldsymbol{r}_{\mathcal{S}_0}^{(0)}$, the local Chebyshev iterative is the following*

$$\boldsymbol{x}^{(t+1)} = \boldsymbol{x}^{(t)} + (1 + \delta_{t:t+1})\boldsymbol{r}_{\mathcal{S}_t}^{(t)} + \delta_{t:t+1}\left(\boldsymbol{x}^{(t)} - \boldsymbol{x}^{(t-1)}\right)_{\mathcal{S}_t}.$$

*Denote $\boldsymbol{\Delta}^{(t)} = \boldsymbol{x}^{(t)} - \boldsymbol{x}^{(t-1)}$, then $\boldsymbol{\Delta}^{(t)} = \sum_{j=0}^{t-1} \left( (1 + \delta_{j:j+1}) \prod_{r=j+1}^{t-1} \delta_{r:r+1} \boldsymbol{r}_{\mathcal{S}_{j:t-1}}^{(j)} \right)$, where $\delta_{0:1} = 0$, $\mathcal{S}_{0:t} = \mathcal{S}_0 \cap \mathcal{S}_1 \cap \cdots \cap \mathcal{S}_t$ and $\delta_{j:j+1} = \delta_j \delta_{j+1}$. We have the following*

$$(1 + \delta_{t:t+1})\boldsymbol{r}_{\overline{\mathcal{S}}_t}^{(t)} + \delta_{t:t+1}\left(\boldsymbol{x}^{(t)} - \boldsymbol{x}^{(t-1)}\right)_{\overline{\mathcal{S}}_t} = \sum_{j=0}^{t} \left( (1 + \delta_{j:j+1}) \prod_{r=j+1}^{t} \delta_{r:r+1} \boldsymbol{r}_{\mathcal{S}_{j,t}}^{(j)} \right),$$

*where $\mathcal{S}_{j,t} \triangleq \mathcal{S}_{j:t-1} \cap \overline{\mathcal{S}}_t$.*

*Proof.* Recall we defined $\delta_0 = 0$ so that $\delta_{0:1} = 0$. We prove this lemma by induction.

For $t = 1$, note $\delta_{0:1} = 0$ and the support of $\boldsymbol{r}^{(0)}$ is $\mathcal{S}_0 = \operatorname{supp}(\boldsymbol{r}^{(0)})$, then

$$\boldsymbol{x}^{(1)} - \boldsymbol{x}^{(0)} = (1 + \delta_{0:1})\boldsymbol{r}^{(0)}.$$

For $t = 2$, note $\mathcal{S}_{1:1} = \mathcal{S}_1$ and $\mathcal{S}_{0:1} = \mathcal{S}_0 \cap \mathcal{S}_1$ by our notation, then

$$\begin{aligned}
\boldsymbol{x}^{(2)} - \boldsymbol{x}^{(1)} &= (1 + \delta_{1:2})\boldsymbol{r}_{\mathcal{S}_{1:1}}^{(1)} + \delta_{1:2}(\boldsymbol{x}^{(1)} - \boldsymbol{x}^{(0)})_{\mathcal{S}_1} \\
&= (1 + \delta_{1:2})\boldsymbol{r}_{\mathcal{S}_{1:1}}^{(1)} + (1 + \delta_{0:1})\delta_{1:2}\boldsymbol{r}_{\mathcal{S}_{0:1}}^{(0)}.
\end{aligned}$$

For $t = 3$, one can build $\boldsymbol{x}^{(3)} - \boldsymbol{x}^{(2)}$ based on $\boldsymbol{x}^{(2)} - \boldsymbol{x}^{(1)}$ and recall $\mathcal{S}_{2:2} = \mathcal{S}_2$, $\mathcal{S}_{1:2} = \mathcal{S}_1 \cap \mathcal{S}_2$

$$\begin{aligned}
\boldsymbol{x}^{(3)} - \boldsymbol{x}^{(2)} &= (1 + \delta_{2:3})\boldsymbol{r}_{\mathcal{S}_{2:2}}^{(2)} + \delta_{2:3}(\boldsymbol{x}^{(2)} - \boldsymbol{x}^{(1)})_{\mathcal{S}_2} \\
&= (1 + \delta_{2:3})\boldsymbol{r}_{\mathcal{S}_{2:2}}^{(2)} + \delta_{2:3}((1 + \delta_{1:2})\boldsymbol{r}_{\mathcal{S}_{1:1}}^{(1)} + (1 + \delta_{0:1})\delta_{1:2}\boldsymbol{r}_{\mathcal{S}_{0:1}}^{(0)})_{\mathcal{S}_2} \\
&= (1 + \delta_{2:3})\boldsymbol{r}_{\mathcal{S}_{2:2}}^{(2)} + (1 + \delta_{1:2})\delta_{2:3}\boldsymbol{r}_{\mathcal{S}_{1:2}}^{(1)} + (1 + \delta_{0:1})\delta_{1:2}\delta_{2:3}\boldsymbol{r}_{\mathcal{S}_{0:2}}^{(0)}.
\end{aligned}$$

For $t = 4$, we continue to have

$$
\begin{aligned}
\boldsymbol{x}^{(4)} - \boldsymbol{x}^{(3)} &= (1 + \delta_{3:4})\boldsymbol{r}^{(3)}_{\mathcal{S}_{3:3}} + \delta_{3:4}(\boldsymbol{x}^{(3)} - \boldsymbol{x}^{(2)})_{\mathcal{S}_{3:3}} \\
&= (1 + \delta_{3:4})\boldsymbol{r}^{(3)}_{\mathcal{S}_{3:3}} + \delta_{3:4}((1 + \delta_{2:3})\boldsymbol{r}^{(2)}_{\mathcal{S}_{2:2}} + (1 + \delta_{1:2})\delta_{2:3}\boldsymbol{r}^{(1)}_{\mathcal{S}_{1:2}} \\
&\quad + (1 + \delta_{0:1})\delta_{1:2}\delta_{2:3}\boldsymbol{r}^{(0)}_{\mathcal{S}_{0:2}})_{\mathcal{S}_{3:3}} \\
&= (1 + \delta_{3:4})\boldsymbol{r}^{(3)}_{\mathcal{S}_{3:3}} + (1 + \delta_{2:3})\delta_{3:4}\boldsymbol{r}^{(2)}_{\mathcal{S}_{2:3}} + (1 + \delta_{1:2})\delta_{3:4}\delta_{2:3}\boldsymbol{r}^{(1)}_{\mathcal{S}_{1:3}} \\
&\quad + (1 + \delta_{0:1})\delta_{1:2}\delta_{2:3}\delta_{3:4}\boldsymbol{r}^{(0)}_{\mathcal{S}_{0:3}} \\
(\boldsymbol{x}^{(4)} - \boldsymbol{x}^{(3)})_{\overline{\mathcal{S}}_4} &= \left((1 + \delta_{3:4})\boldsymbol{r}^{(3)}_{\mathcal{S}_{3:3}} + \delta_{3:4}(\boldsymbol{x}^{(3)} - \boldsymbol{x}^{(2)})_{\mathcal{S}_{3:3}}\right)_{\overline{\mathcal{S}}_4}.
\end{aligned}
$$

By induction $t \geq 1$,

$$
\boldsymbol{x}^{(t)} - \boldsymbol{x}^{(t-1)} = \sum_{j=0}^{t-1}\left((1 + \delta_{j:j+1}) \prod_{r=j+1}^{t-1} \delta_{r:r+1}\boldsymbol{r}^{(j)}_{\mathcal{S}_{j:t-1}}\right),
$$

where the convention notation $\sum_{i=1}^{0} \cdot = 0$ and $\prod_{j=i+1}^{i} \cdot = 1$. To verify the inductive step, consider for $t + 1$, we have

$$
\begin{aligned}
\boldsymbol{x}^{(t+1)} - \boldsymbol{x}^{(t)} &= (1 + \delta_{t:t+1})\boldsymbol{r}^{(t)}_{\mathcal{S}_t} + \delta_{t:t+1}(\boldsymbol{x}^{(t)} - \boldsymbol{x}^{(t-1)})_{\mathcal{S}_t} \\
&= (1 + \delta_{t:t+1})\boldsymbol{r}^{(t)}_{\mathcal{S}_t} + \delta_{t:t+1}(\boldsymbol{x}^{(t)} - \boldsymbol{x}^{(t-1)})_{\mathcal{S}_t} \\
&= (1 + \delta_{t:t+1})\boldsymbol{r}^{(t)}_{\mathcal{S}_t} + \delta_{t:t+1}\left(\sum_{j=0}^{t-1}\left((1 + \delta_{j:j+1}) \prod_{r=j+1}^{t-1} \delta_{r:r+1}\boldsymbol{r}^{(j)}_{\mathcal{S}_{j:t-1}}\right)\right)_{\mathcal{S}_t} \\
&= \sum_{j=0}^{t}\left((1 + \delta_{j:j+1}) \prod_{r=j+1}^{t} \delta_{r:r+1}\boldsymbol{r}^{(j)}_{\mathcal{S}_{j:t}}\right).
\end{aligned}
$$

To see the second equation, note

$$
\begin{aligned}
&(1 + \delta_{t:t+1})\boldsymbol{r}^{(t)}_{\overline{\mathcal{S}}_t} + \delta_{t:t+1}\left(\boldsymbol{x}^{(t)} - \boldsymbol{x}^{(t-1)}\right)_{\overline{\mathcal{S}}_t} \\
&= (1 + \delta_{t:t+1})\boldsymbol{r}^{(t)}_{\overline{\mathcal{S}}_t} + \sum_{j=0}^{t-1}\left((1 + \delta_{j:j+1}) \prod_{r=j+1}^{t} \delta_{r:r+1}\boldsymbol{r}^{(j)}_{\mathcal{S}_{j:t-1} \cap \overline{\mathcal{S}}_t}\right) \\
&= \sum_{j=0}^{t}\left((1 + \delta_{j:j+1}) \prod_{r=j+1}^{t} \delta_{r:r+1}\boldsymbol{r}^{(j)}_{\mathcal{S}_{j,t}}\right),
\end{aligned}
$$

where recall we denote $\mathcal{S}_{j,t} \triangleq \mathcal{S}_{j:t-1} \cap \overline{\mathcal{S}}_t$. $\qquad\square$

**Lemma C.8** (Local Chebyshev updates). *Given the updates of $\boldsymbol{x}^{(t+1)}$ as defined by LOCCH in* (10), *we have the following local updates*

$$
\begin{cases}
\boldsymbol{x}^{(t+1)} = \boldsymbol{x}^{(t)} + (1 + \delta_{t:t+1})\boldsymbol{r}^{(t)}_{\mathcal{S}_t} + \delta_{t:t+1}\left(\boldsymbol{x}^{(t)} - \boldsymbol{x}^{(t-1)}\right)_{\mathcal{S}_t} \\
\boldsymbol{r}^{(t+1)} - 2\delta_{t+1}\boldsymbol{W}\boldsymbol{r}^{(t)} + \delta_{t:t+1}\boldsymbol{r}^{(t-1)} = \sum_{j=0}^{t}\left((1 + \delta_{j:j+1}) \prod_{r=j+1}^{t} \delta_{r:r+1}\boldsymbol{Q}\boldsymbol{r}^{(j)}_{\mathcal{S}_{j,t}}\right)
\end{cases} \tag{28}
$$

*Proof.* We only need to show the second equation of (28). The residual of LOCCH updates is

$$\boldsymbol{r}^{(t+1)} = \boldsymbol{b} - \boldsymbol{Q}\boldsymbol{x}^{(t+1)}$$

$$\boldsymbol{r}^{(t+1)} = \boldsymbol{b} - \boldsymbol{Q}\big(\boldsymbol{x}^{(t)} + (1+\delta_{t:t+1})\boldsymbol{r}^{(t)}_{\mathcal{S}_t} + \delta_{t:t+1}(\boldsymbol{x}^{(t)} - \boldsymbol{x}^{(t-1)})_{\mathcal{S}_t}\big)$$

$$= \boldsymbol{r}^{(t)} - (1+\delta_{t:t+1})\boldsymbol{Q}\boldsymbol{r}^{(t)}_{\mathcal{S}_t} - \delta_{t:t+1}\boldsymbol{Q}(\boldsymbol{x}^{(t)} - \boldsymbol{x}^{(t-1)})_{\mathcal{S}_t}$$

$$\underbrace{\boldsymbol{r}^{(t+1)} + (1+\delta_{t:t+1})\boldsymbol{Q}\boldsymbol{r}^{(t)} + \delta_{t:t+1}\boldsymbol{Q}(\boldsymbol{x}^{(t)} - \boldsymbol{x}^{(t-1)}) - \boldsymbol{r}^{(t)}}_{\boldsymbol{u}}$$

$$= \underbrace{(1+\delta_{t:t+1})\boldsymbol{Q}\boldsymbol{r}^{(t)}_{\overline{\mathcal{S}}_t} + \delta_{t:t+1}\boldsymbol{Q}\big(\boldsymbol{x}^{(t)} - \boldsymbol{x}^{(t-1)}\big)_{\overline{\mathcal{S}}_t}}_{\text{small noisy part}}.$$

Note $\boldsymbol{Q}(\boldsymbol{x}^{(t)} - \boldsymbol{x}^{(t-1)}) = \boldsymbol{b} - \boldsymbol{Q}\boldsymbol{x}^{(t-1)} - (\boldsymbol{b} - \boldsymbol{Q}(\boldsymbol{x}^{(t)}) = \boldsymbol{r}^{(t-1)} - \boldsymbol{r}^{(t)}$ and then $\boldsymbol{u}$ becomes

$$\boldsymbol{u} = \boldsymbol{r}^{(t+1)} + (1+\delta_{t:t+1})\boldsymbol{Q}\boldsymbol{r}^{(t)} + \delta_{t:t+1}(\boldsymbol{r}^{(t-1)} - \boldsymbol{r}^{(t)}) - \boldsymbol{r}^{(t)}$$

$$= \boldsymbol{r}^{(t+1)} - 2\delta_{t+1}\boldsymbol{W}\boldsymbol{r}^{(t)} + \delta_{t:t+1}\boldsymbol{r}^{(t-1)},$$

where the last equality is due to $(1+\delta_{t:t+1})\boldsymbol{Q}\boldsymbol{r}^{(t)} = (1+\delta_{t:t+1})\boldsymbol{r}^{(t)} - 2\delta_{t+1}\boldsymbol{W}\boldsymbol{r}^{(t)}$ by noticing $(1+\delta_t\delta_{t+1})\frac{1-\alpha}{1+\alpha} = 2\delta_{t+1}$ in Lemma C.5. Hence, we have the second equation. To see the noisy part, note by Lemma C.7

$$(1+\delta_{t:t+1})\boldsymbol{r}^{(t)}_{\overline{\mathcal{S}}_t} + \delta_{t:t+1}\big(\boldsymbol{x}^{(t)} - \boldsymbol{x}^{(t-1)}\big)_{\overline{\mathcal{S}}_t} = (1+\delta_{t:t+1})\boldsymbol{r}^{(t)}_{\overline{\mathcal{S}}_t} + \sum_{j=0}^{t-1}\Big((1+\delta_{j:j+1})\prod_{r=j+1}^{t}\delta_{r:r+1}\boldsymbol{r}^{(j)}_{\mathcal{S}_{j,t}}\Big)$$

$$(1+\delta_{t:t+1})\boldsymbol{Q}\boldsymbol{r}^{(t)}_{\overline{\mathcal{S}}_t} + \delta_{t:t+1}\boldsymbol{Q}\big(\boldsymbol{x}^{(t)} - \boldsymbol{x}^{(t-1)}\big)_{\overline{\mathcal{S}}_t} = \sum_{j=0}^{t}\Big((1+\delta_{j:j+1})\prod_{r=j+1}^{t}\delta_{r:r+1}\boldsymbol{Q}\boldsymbol{r}^{(j)}_{\mathcal{S}_{j,t}}\Big).$$

$\square$

**Lemma 4.1** (Residual updates of LOCCH). *Given* $t \geq 1, \boldsymbol{x}^{(0)} = \boldsymbol{0}, \boldsymbol{x}^{(1)} = \boldsymbol{x}^0 + \boldsymbol{r}^{(0)}_{\mathcal{S}_0}$. *The residual* $\boldsymbol{r}^{(t)}$ *of* LOCCH *defined in Equ.* (28) *satisfies*

$$\boldsymbol{V}^\top\boldsymbol{r}^{(t)} = \delta_{1:t}\boldsymbol{Z}_t\boldsymbol{V}^\top\boldsymbol{r}^{(0)} + \delta_{1:t}t\boldsymbol{u}_{0,t} + 2\sum_{k=1}^{t-1}\delta_{k+1:t}(t-k)\boldsymbol{u}_{k,t}, \tag{29}$$

*where*

$$\boldsymbol{u}_{k,t} = \begin{cases} \sum_{j=1}^{t-1}\delta_{2:j}\boldsymbol{H}_{j,t}\left(\boldsymbol{I} - \frac{1-\alpha}{1+\alpha}\boldsymbol{\Lambda}\right)\boldsymbol{V}^\top\boldsymbol{r}^{(0)}_{\mathcal{S}_{0,j}}/t & \text{if } k = 0 \\ \sum_{j=k}^{t-1}\left(\delta_{k+1:j}\boldsymbol{H}_{j,t}\left(\frac{1+\alpha}{1-\alpha}\boldsymbol{I} - \boldsymbol{\Lambda}\right)\boldsymbol{V}^\top\boldsymbol{r}^{(k)}_{\mathcal{S}_{k,j}}\right)/(t-k) & \text{if } k \geq 1, \end{cases}$$

$$\boldsymbol{Z}_t = \begin{cases} \mathbf{diag}\left(1, \ldots, \cos(\theta_i t), \ldots, (-1)^t\right) & \text{for bipartite graphs} \\ \mathbf{diag}\left(1, \ldots, \cos(\theta_i t), \ldots, \cos(\theta_n t)\right) & \text{for non-bipartite graphs}, \end{cases}$$

$$\boldsymbol{H}_{k,t} = \begin{cases} \mathbf{diag}\left(t-k, \ldots, \frac{\sin(\theta_i(t-k))}{\sin\theta_i}, \ldots, (-1)^{t-k-1}(t-k)\right) & \text{for bipartite graphs} \\ \mathbf{diag}\left(t-k, \ldots, \frac{\sin(\theta_i(t-k))}{\sin\theta_i}, \ldots, \frac{\sin(\theta_n(t-k))}{\sin\theta_n}\right) & \text{for non-bipartite graphs}. \end{cases}$$

*Proof.* We first decompose the residual equation in (28) as

$$\boldsymbol{V}^\top\boldsymbol{r}^{(t+1)} - 2\delta_{t+1}\boldsymbol{\Lambda}\boldsymbol{V}^\top\boldsymbol{r}^{(t)} + \delta_{t:t+1}\boldsymbol{V}^\top\boldsymbol{r}^{(t-1)}$$

$$= \underbrace{\sum_{j=0}^{t}\left((1+\delta_{j:j+1})\prod_{r=j+1}^{t}(\delta_{r:r+1})\left(\boldsymbol{I} - \frac{1-\alpha}{1+\alpha}\boldsymbol{\Lambda}\right)\boldsymbol{V}^\top\boldsymbol{r}^{(j)}_{\mathcal{S}_{j,t}}\right)}_{\boldsymbol{f}^{(t)}}.$$

Define $\boldsymbol{V}^\top \boldsymbol{r}^{(t)} = \delta_{1:t}\boldsymbol{y}^{(t)}$ and $\boldsymbol{V}^\top \boldsymbol{r}^{(0)} = \delta_{1:0}\boldsymbol{y}^{(0)} = \boldsymbol{y}^{(0)}$ by default. Then we have

$$\delta_{1:t+1}\boldsymbol{y}^{(t+1)} - 2\delta_{1:t+1}\boldsymbol{\Lambda}\boldsymbol{y}^{(t)} + \delta_{1:t+1}\boldsymbol{y}^{(t-1)} = \boldsymbol{f}^{(t)} \;\Rightarrow\; \boldsymbol{y}^{(t+1)} - 2\boldsymbol{\Lambda}\boldsymbol{y}^{(t)} + \boldsymbol{y}^{(t-1)} = \frac{\boldsymbol{f}^{(t)}}{\delta_{1:t+1}}.$$

Note $\boldsymbol{V}^\top \boldsymbol{r}^{(1)} = \delta_1 \boldsymbol{y}^{(1)} = \boldsymbol{V}^\top \delta_1 \boldsymbol{W}\boldsymbol{r}^{(0)} = \delta_1 \boldsymbol{\Lambda}\boldsymbol{y}^{(0)}$, which indicates $\boldsymbol{y}^{(1)} = \boldsymbol{\Lambda}\boldsymbol{y}^{(0)}$. Then, by the Lemma C.4, each $y_i^{(t)}$ has the solution

$$y_i^{(t)} = \begin{cases} y_i^{(0)} + \sum_{k=1}^{t-1}(t-k)f_i^{(k)}/(\delta_{1:k+1}) & \text{if} \quad \lambda_i = 1 \\ \cos(\theta_i t)y_i^{(0)} + \sum_{k=1}^{t-1}\frac{\sin(\theta_i(t-k))}{\sin\theta_i}f_i^{(k)}/(\delta_{1:k+1}) & \text{if} \quad |\lambda_i| < 1 \\ (-1)^t y_i^{(0)} + \sum_{k=1}^{t-1}(-1)^{t-k-1}(t-k)f_i^{(k)}/(\delta_{1:k+1}) & \text{if} \quad \lambda_i = -1. \end{cases}$$

Use $\boldsymbol{Z}_t$ and $\boldsymbol{H}_{k,t}$, we write the solution of the second-order difference equation as

$$\boldsymbol{y}^{(t)} = \boldsymbol{Z}_t \boldsymbol{y}^{(0)} + \sum_{k=1}^{t-1} \boldsymbol{H}_{k,t}\boldsymbol{f}^{(k)}/(\delta_{1:k+1})$$

$$\boldsymbol{V}^\top\boldsymbol{r}^{(t)} = \delta_{1:t}\boldsymbol{y}^{(t)} = \delta_{1:t}\boldsymbol{Z}_t\boldsymbol{V}^\top\boldsymbol{r}^{(0)} + \delta_{1:t}\sum_{k=1}^{t-1}\boldsymbol{H}_{k,t}\boldsymbol{f}^{(k)}/(\delta_{1:k+1})$$

$$\boldsymbol{V}^\top\boldsymbol{r}^{(t)} = \delta_{1:t}\boldsymbol{Z}_t\boldsymbol{V}^\top\boldsymbol{r}^{(0)} + \sum_{k=1}^{t-1}\delta_{k+2:t}\boldsymbol{H}_{k,t}\sum_{j=0}^{k}\left((1+\delta_{j:j+1})\prod_{r=j+1}^{k}(\delta_{r:r+1})\left(\boldsymbol{I}-\frac{1-\alpha}{1+\alpha}\boldsymbol{\Lambda}\right)\boldsymbol{V}^\top\boldsymbol{r}_{\mathcal{S}_{j,k}}^{(j)}\right).$$

Note $1 + \delta_{j:j+1} = 2\delta_{j+1}\frac{1+\alpha}{1-\alpha}, j \geq 1$, then $(1+\delta_{j:j+1})\prod_{r=j+1}^{k}(\delta_{r:r+1}) = 2\frac{1+\alpha}{1-\alpha}\delta_{j+1:k}\delta_{j+1:k+1}$. Then, we have

$$\boldsymbol{V}^\top\boldsymbol{r}^{(t)} = \delta_{1:t}\boldsymbol{Z}_t\boldsymbol{V}^\top\boldsymbol{r}^{(0)} + \sum_{k=1}^{t-1}\delta_{k+2:t}\boldsymbol{H}_{k,t}\left(\delta_{2:k}\delta_{1:k+1}\left(\boldsymbol{I}-\frac{1-\alpha}{1+\alpha}\boldsymbol{\Lambda}\right)\boldsymbol{V}^\top\boldsymbol{r}_{\mathcal{S}_{0,k}}^{(0)}\right)$$

$$+ 2\sum_{k=1}^{t-1}\delta_{k+2:t}\boldsymbol{H}_{k,t}\sum_{j=1}^{k}\left(\delta_{j+1:k}\delta_{j+1:k+1}\left(\frac{1+\alpha}{1-\alpha}\boldsymbol{I}-\boldsymbol{\Lambda}\right)\boldsymbol{V}^\top\boldsymbol{r}_{\mathcal{S}_{j,k}}^{(j)}\right)$$

$$= \delta_{1:t}\boldsymbol{Z}_t\boldsymbol{V}^\top\boldsymbol{r}^{(0)} + \delta_{1:t}\sum_{k=1}^{t-1}\delta_{2:k}\boldsymbol{H}_{k,t}\left(\boldsymbol{I}-\frac{1-\alpha}{1+\alpha}\boldsymbol{\Lambda}\right)\boldsymbol{V}^\top\boldsymbol{r}_{\mathcal{S}_{0,k}}^{(0)}$$

$$+ 2\sum_{k=1}^{t-1}\delta_{k+2:t}\boldsymbol{H}_{k,t}\sum_{j=1}^{k}\left(\delta_{j+1:k}\delta_{j+1:k+1}\left(\frac{1+\alpha}{1-\alpha}\boldsymbol{I}-\boldsymbol{\Lambda}\right)\boldsymbol{V}^\top\boldsymbol{r}_{\mathcal{S}_{j,k}}^{(j)}\right),$$

where the last term can be expanded as

$$\sum_{k=1}^{t-1}\delta_{k+2:t}\boldsymbol{H}_{k,t}\sum_{j=1}^{k}\left(\delta_{j+1:k}\delta_{j+1:k+1}\underbrace{\left(\frac{1+\alpha}{1-\alpha}\boldsymbol{I}-\boldsymbol{\Lambda}\right)\boldsymbol{V}^\top\boldsymbol{r}_{\mathcal{S}_{j,k}}^{(j)}}_{\boldsymbol{w}_{\mathcal{S}_{j,k}}^{(j)}}\right) =$$

$$\delta_{3:t}\boldsymbol{H}_{1,t}\delta_{2:1}\delta_{2:2}\boldsymbol{w}_{\mathcal{S}_{1,1}}^{(1)} +$$

$$\delta_{4:t}\boldsymbol{H}_{2,t}\delta_{2:2}\delta_{2:3}\boldsymbol{w}_{\mathcal{S}_{1,2}}^{(1)} + \delta_{4:t}\boldsymbol{H}_{2,t}\delta_{3:2}\delta_{3:3}\boldsymbol{u}_{\mathcal{S}_{2,2}}^{(2)} +$$

$$\delta_{5:t}\boldsymbol{H}_{3,t}\delta_{2:3}\delta_{2:4}\boldsymbol{w}_{\mathcal{S}_{1,3}}^{(1)} + \delta_{5:t}\boldsymbol{H}_{3,t}\delta_{3:3}\delta_{3:4}\boldsymbol{w}_{\mathcal{S}_{2,3}}^{(2)} + \delta_{5:t}\boldsymbol{H}_{3,t}\delta_{4:3}\delta_{4:4}\boldsymbol{w}_{\mathcal{S}_{3,3}}^{(3)} +$$

$$\delta_{6:t}\boldsymbol{H}_{4,t}\delta_{2:4}\delta_{2:5}\boldsymbol{w}_{\mathcal{S}_{1,4}}^{(1)} + \delta_{6:t}\boldsymbol{H}_{4,t}\delta_{3:4}\delta_{3:5}\boldsymbol{w}_{\mathcal{S}_{2,4}}^{(2)} + \delta_{6:t}\boldsymbol{H}_{4,t}\delta_{4:4}\delta_{4:5}\boldsymbol{w}_{\mathcal{S}_{3,4}}^{(3)} + \delta_{6:t}\boldsymbol{H}_{4,t}\delta_{5:4}\delta_{5:5}\boldsymbol{w}_{\mathcal{S}_{4,4}}^{(4)} +$$

$$\delta_{7:t}\boldsymbol{H}_{5,t}\delta_{2:5}\delta_{2:6}\boldsymbol{w}_{\mathcal{S}_{1,5}}^{(1)} + \delta_{7:t}\boldsymbol{H}_{5,t}\delta_{3:5}\delta_{3:6}\boldsymbol{w}_{\mathcal{S}_{2,5}}^{(2)} + \delta_{7:t}\boldsymbol{H}_{5,t}\delta_{4:5}\delta_{4:6}\boldsymbol{w}_{\mathcal{S}_{3,5}}^{(3)} + \delta_{7:t}\boldsymbol{H}_{5,t}\delta_{5:5}\delta_{5:6}\boldsymbol{w}_{\mathcal{S}_{4,5}}^{(4)}$$

$$+ \delta_{7:t}\boldsymbol{H}_{5,t}\delta_{6:5}\delta_{6:6}\boldsymbol{w}_{\mathcal{S}_{5,5}}^{(5)} +$$

$$\vdots$$

$$= \sum_{k=1}^{t-1}\delta_{k+1:t}\sum_{j=k}^{t-1}\left(\delta_{k+1:j}\boldsymbol{H}_{j,t}\left(\frac{1+\alpha}{1-\alpha}\boldsymbol{I}-\boldsymbol{\Lambda}\right)\boldsymbol{V}^\top\boldsymbol{r}_{\mathcal{S}_{k,j}}^{(k)}\right).$$

Here, we denote $\delta_{k+1:k} = 1$. The final iterative update is

$$\boldsymbol{V}^\top \boldsymbol{r}^{(t)} = \delta_{1:t} \boldsymbol{Z}_t \boldsymbol{V}^\top \boldsymbol{r}^{(0)} + \delta_{1:t} t \underbrace{\sum_{j=1}^{t-1} \delta_{2:j} \boldsymbol{H}_{j,t} \left( \boldsymbol{I} - \frac{1-\alpha}{1+\alpha} \boldsymbol{\Lambda} \right) \boldsymbol{V}^\top \boldsymbol{r}^{(0)}_{\mathcal{S}_{0,j}} \Big/ t}_{\boldsymbol{u}_{0,t}}$$

$$+ 2 \sum_{k=1}^{t-1} \delta_{k+1:t}(t-k) \underbrace{\sum_{j=k}^{t-1} \left( \delta_{k+1:j} \boldsymbol{H}_{j,t} \left( \frac{1+\alpha}{1-\alpha} \boldsymbol{I} - \boldsymbol{\Lambda} \right) \boldsymbol{V}^\top \boldsymbol{r}^{(k)}_{\mathcal{S}_{k,j}} \right) \Big/ (t-k)}_{\boldsymbol{u}_{k,t}}.$$

$\square$

### C.5 Convergence of LOCCH and Proof of Theorem 4.2

**Corollary C.9.** *Let $\beta_k$ be lower bound of residual reduction satisfies $\|\boldsymbol{u}_{k,t}\|_2 \le \beta_k \|\boldsymbol{r}^{(k)}\|_2$, then the upper bound of $\|\boldsymbol{r}^{(t)}\|_2$ can be characterized as*

$$\|\boldsymbol{r}^{(t)}\|_2 \le \delta_{1:t} \prod_{j=0}^{t-1} (1+\beta_j) y_t, \text{ where } y_{t+1} - 2y_t + \frac{y_{t-1}}{(1+\beta_{t-1})(1+\beta_t)} = 0, \qquad (30)$$

*where $y_0 = y_1 = \|\boldsymbol{r}^0\|_2$.*

*Proof.* Since $\|\boldsymbol{u}_{k,t}\|_2 \le \beta_k \|\boldsymbol{r}^{(k)}\|_2$, the final iterative updates (29) can be bounded as

$$\boldsymbol{V}^\top \boldsymbol{r}^{(t)} = \delta_{1:t} \boldsymbol{Z}_t \boldsymbol{V}^\top \boldsymbol{r}^{(0)} + \delta_{1:t} t \boldsymbol{u}_{0,t} + 2 \sum_{k=1}^{t-1} \delta_{k+1:t}(t-k) \boldsymbol{u}_{k,t}$$

$$\|\boldsymbol{r}^{(t)}\|_2 \le \delta_{1:t} \|\boldsymbol{r}^{(0)}\|_2 + \delta_{1:t} t \beta_0 \|\boldsymbol{r}^{(0)}\|_2 + 2 \sum_{k=1}^{t-1} \delta_{k+1:t}(t-k) \beta_k \|\boldsymbol{r}^{(k)}\|_2$$

$$\|\boldsymbol{r}^{(t)}\|_2 - 2 \sum_{k=1}^{t-1} \delta_{k+1:t}(t-k) \beta_k \|\boldsymbol{r}^{(k)}\|_2 \le \delta_{1:t}(1 + t\beta_0) \|\boldsymbol{r}^{(0)}\|_2, \qquad (31)$$

where $t = 0, 1, \ldots, T$. These $T+1$ (including a trivial one where $\|\boldsymbol{r}^{(0)}\|_2 \le \|\boldsymbol{r}^{(0)}\|_2$) inequalities shown in Equ. (31) form a system of linear inequality matrix as

$$\underbrace{\begin{pmatrix} 1 & 0 & 0 & \cdots & 0 \\ -z_{21} & 1 & 0 & \cdots & 0 \\ -z_{31} & -z_{32} & 1 & \cdots & 0 \\ \vdots & \vdots & \vdots & \ddots & \vdots \\ -z_{T1} & -z_{T2} & -z_{T3} & \cdots & 1 \end{pmatrix}}_{\boldsymbol{I} - \boldsymbol{Z}_L} \begin{pmatrix} \|\boldsymbol{r}^{(1)}\|_2 \\ \|\boldsymbol{r}^{(2)}\|_2 \\ \|\boldsymbol{r}^{(3)}\|_2 \\ \vdots \\ \|\boldsymbol{r}^{(T-1)}\|_2 \end{pmatrix} \le \begin{pmatrix} \delta_{1:1}(1+1\beta_0)\|\boldsymbol{r}^{(0)}\|_2 \\ \delta_{1:2}(1+2\beta_0)\|\boldsymbol{r}^{(0)}\|_2 \\ \delta_{1:3}(1+3\beta_0)\|\boldsymbol{r}^{(0)}\|_2 \\ \vdots \\ \delta_{1:T}(1+T\beta_0 t)\|\boldsymbol{r}^{(0)}\|_2 \end{pmatrix} := \boldsymbol{c},$$

where $(\boldsymbol{Z}_L)_{tk} = 2\delta_{k+1:t}(t-k)\beta_k$ for $t = 2, 3, \ldots, T$ and $k = 1, 2, \ldots, t-1$. Assume that $\boldsymbol{N} \in \mathbb{R}^{T \times T}$ is a strictly lower triangular matrix, then we know the established formula $(\boldsymbol{I} + \boldsymbol{N})^{-1} = \boldsymbol{I} + \sum_{k=1}^{T-1} (-1)^k \boldsymbol{N}^k$. Hence, we have the following

$$(\boldsymbol{I} - \boldsymbol{Z}_L)^{-1} = \boldsymbol{I} + \sum_{k=1}^{T-1} \boldsymbol{Z}_L^k.$$

Given that $(\boldsymbol{I} - \boldsymbol{Z}_L)^{-1} \geq \boldsymbol{0}$, then we obtain an upper-bound of $\begin{pmatrix} \|\boldsymbol{r}^{(1)}\|_2 \\ \|\boldsymbol{r}^{(2)}\|_2 \\ \vdots \\ \|\boldsymbol{r}^{(T)}\|_2 \end{pmatrix} \leq \boldsymbol{z} \triangleq (\boldsymbol{I} - \boldsymbol{Z}_L)^{-1} \boldsymbol{c}.$

It leads to the following new second-order difference equation

$$z_t - 2 \sum_{k=1}^{t-1} \delta_{k+1:t}(t-k)\beta_k z_k = \delta_{1:t}(1 + t\beta_0)z_0, \qquad \text{for } t = 1, 2, \ldots, T,$$

where the initial value of $z_0 = \|\boldsymbol{r}^{(0)}\|_2$. Following the argument in Theorem 1 of Golub & Overton [18], we construct a second-order homogeneous equation for $z_t$ as

$$z_t = \delta_{1:t}(1 + t\beta_0)z_0 + 2 \sum_{k=1}^{t-1} \delta_{k+1:t}(t-k)\beta_k z_k$$

$$z_{t+1} = \delta_{1:t+1}(1 + (t+1)\beta_0)z_0 + 2 \sum_{k=1}^{t} \delta_{k+1:t+1}(t+1-k)\beta_k z_k$$

$$\delta_{t+1}z_t = \delta_{1:t+1}(1 + t\beta_0)z_0 + 2 \sum_{k=1}^{t} \delta_{k+1:t+1}(t-k)\beta_k z_k$$

$$z_{t+1} - \delta_{t+1}z_t = \delta_{1:t+1}\beta_0 z_0 + 2 \sum_{k=1}^{t} \delta_{k+1:t+1}\beta_k z_k, \tag{32}$$

where Equ. (32) is obtained by the difference between the second equation and the third equation. Similarly,

$$z_{t-1} = \delta_{1:t-1}(1 + (t-1)\beta_0)z_0 + 2 \sum_{k=1}^{t-2} \delta_{k+1:t-1}(t-k-1)\beta_k z_k$$

$$\delta_{t:t+1}z_{t-1} = \delta_{1:t+1}(1 + (t-1)\beta_0)z_0 + 2 \sum_{k=1}^{t-2} \delta_{k+1:t+1}(t-k-1)\beta_k z_k$$

$$\delta_{t+1}z_t - \delta_{t:t+1}z_{t-1} = \delta_{1:t+1}\beta_0 z_0 + 2 \sum_{k=1}^{t-1} \delta_{k+1:t+1}\beta_k z_k, \tag{33}$$

where Equ. (33) is obtained by the difference of the first two. Hence, Equ. (32) − (33) gives us

$$z_{t+1} - 2(1 + \beta_t)\delta_{t+1}z_t + \delta_{t:t+1}z_{t-1} = 0, \qquad \text{for } t = 1, 2, \ldots, T,$$

where two initials are $z_0 = \|\boldsymbol{r}^{(0)}\|_0$ and $z_1 = \delta_1(1 + \beta_0)\|\boldsymbol{r}^{(0)}\|_2$. Let $z_t = \delta_{1:t}\hat{z}_t$, then

$$\hat{z}_{t+1} - 2(1 + \beta_t)\hat{z}_t + \hat{z}_{t-1} = 0,$$

where two initials are $\hat{z}_0 = z_0 = \|\boldsymbol{r}^{(0)}\|_0$ and $\hat{z}_1 = (1 + \beta_0)\|\boldsymbol{r}^{(0)}\|_2$. We finish the proof by setting $\prod_{j=0}^{t-1}(1 + \beta_j)y_t = \hat{z}_t$. $\qquad\qquad\square$

*Remark* C.10. Key points of the above proof strategy largely follow from Golub & Overton [18]. However, different from the original technique, we generalize the strategy to a parameterized version.

**Lemma C.11.** *Given $\beta_j \geq 0$, the following second-order difference equation*

$$x_{t+1} - 2(1 + \beta_t)x_t + x_{t-1} = 0.$$

*has the following solution*

$$x_t = \prod_{j=0}^{t-1}(1 + \beta_j)y_t,$$

*where $y_{t+1} - 2y_t + \frac{y_{t-1}}{(1+\beta_{t-1})(1+\beta_t)} = 0$ with $y_0 = x_0$ and $y_1 = x_1/(1 + \beta_0)$.*

*Proof.* Assume $x_t = \left(-\frac{1}{2}\right)^t \prod_{j=0}^{t-1}\left(-2(1+\beta_j)\right) y_t$. Then, following the equation, we have

$$\left(-\frac{1}{2}\right)^{t+1} \prod_{j=0}^{t} \left(-2(1+\beta_j)\right) y_{t+1}$$

$$- 2(1+\beta_t)\left(-\frac{1}{2}\right)^t \prod_{j=0}^{t-1} \left(-2(1+\beta_j)\right) y_t + \left(-\frac{1}{2}\right)^{t-1} \prod_{j=0}^{t-2} \left(-2(1+\beta_j)\right) y_{t-1} = 0.$$

Since $\beta_j \geq 0$, the term $\prod_{j=0}^{t}\left(-2(1+\beta_j)\right) \neq 0$, we divide it on both sides to have

$$\left(-\frac{1}{2}\right)^{t+1} y_{t+1} + \left(-\frac{1}{2}\right)^t y_t + \left(-\frac{1}{2}\right)^{t-1} \frac{1}{4(1+\beta_{t-1})(1+\beta_t)} y_{t-1} = 0.$$

Hence, it is simplified into $y_{t+1} - 2y_t + \frac{y_{t-1}}{(1+\beta_{t-1})(1+\beta_t)} = 0$. To make a simplification on $x_t$, we prove the lemma. $\square$

**Theorem 4.2** (Runtime bound of LocCH). *Given the configuration $\theta = (\alpha, \epsilon, s, \mathcal{G})$ with $\alpha \in (0,1)$ and $\epsilon \leq 1/d_s$ and let $\boldsymbol{r}^{(T)}$ and $\boldsymbol{x}^{(T)}$ be returned by LocCH defined in (10) for solving Equ. (3). For $t \geq 1$, the residual magnitude $\|\boldsymbol{r}^{(t)}\|_2$ has the following convergence bound*

$$\|\boldsymbol{r}^{(t)}\|_2 \leq \delta_{1:t} \prod_{j=0}^{t-1}(1+\beta_j)y_t,$$

*where $y_t$ is a sequence of positive numbers solving $y_{t+1} - 2y_t + y_{t-1}/((1+\beta_{t-1})(1+\beta_t)) = 0$ with $y_0 = y_1 = \|\boldsymbol{r}^{(0)}\|_2$. Suppose the geometric mean $\overline{\beta}_t \triangleq (\prod_{j=0}^{t-1}(1+\beta_j))^{1/t}$ of $\beta_t$ be such that $\overline{\beta}_t = 1 + \frac{c\sqrt{\alpha}}{1-\sqrt{\alpha}}$ where $c \in [0,2)$. There exists a real implementation of (9) such that the runtime $\mathcal{T}_{\text{LocCH}}$ is bounded by*

$$\mathcal{T}_{\text{LocCH}} \leq \Theta\left(\frac{(1+\sqrt{\alpha})\overline{\text{vol}}(\mathcal{S}_T)}{\sqrt{\alpha}(2-c)} \ln \frac{2y_T}{\epsilon}\right).$$

*Proof.* The convergence bound of $\boldsymbol{r}^{(t)}$ directly follows from Corollary C.9. Since we assume that there exists $c \in [0,2)$ such that $\prod_{j=0}^{t-1}(1+\beta_j) \leq \left(1 + \frac{c\sqrt{\alpha}}{1-\sqrt{\alpha}}\right)^t$. Then multiplying both sides by $\tilde{\alpha}^t$, we have

$$\tilde{\alpha}^t \prod_{j=0}^{t-1}(1+\beta_j) \leq \left(1 - \frac{(2-c)\sqrt{\alpha}}{1+\sqrt{\alpha}}\right)^t.$$

Then we have

$$\|\boldsymbol{r}^{(t)}\|_2 \leq \delta_{1:t}\prod_{j=0}^{t-1}(1+\beta_j)y_t \overset{\delta_{1:t}\leq 2\tilde{\alpha}^t}{\leq} 2\tilde{\alpha}^t \overline{\beta}_t^t y_t \leq \epsilon$$

$$t \ln\left(\frac{1-\sqrt{\alpha}}{1+\sqrt{\alpha}}\left(\prod_{j=0}^{t-1}(1+\beta_j)\right)^{1/t}\right) \leq \ln\left(\frac{\epsilon}{2y_t}\right)$$

$$t \geq \left\lceil \ln\left(\frac{2y_t}{\epsilon}\right) \bigg/ \ln\left(\frac{1+\sqrt{\alpha}}{(1-\sqrt{\alpha})\overline{\beta}_t}\right)\right\rceil$$

Since $\overline{\beta}_t = \left(\prod_{j=0}^{t-1}(1+\beta_j)\right)^{1/t}$, and by using $\frac{1+x}{x} \geq \frac{1}{\ln(1+x)}$ and letting $x = \frac{1+\sqrt{\alpha}}{(1-\sqrt{\alpha})\overline{\beta}_t} - 1 > 0$, then $t$ can be lower bounded further by

$$t \geq \left\lceil \frac{1+\sqrt{\alpha}}{1+\sqrt{\alpha}-(1-\sqrt{\alpha})\overline{\beta}_t} \ln\left(\frac{2y_t}{\epsilon}\right)\right\rceil \geq \left\lceil \ln\left(\frac{2y_t}{\epsilon}\right) \bigg/ \ln\left(\frac{1+\sqrt{\alpha}}{(1-\sqrt{\alpha})\overline{\beta}_t}\right)\right\rceil.$$

Since we assumed $\beta_t = (1 + \frac{c\sqrt{\alpha}}{1-\sqrt{\alpha}})$, which means $1 \leq \bar{\beta}_t = (1 + \frac{c\sqrt{\alpha}}{1-\sqrt{\alpha}})$, so $\bar{\beta}_t \in \left[1, \frac{1+\sqrt{\alpha}}{1-\sqrt{\alpha}}\right]$. Then, we find such an upper bound of $t$ so that LocCH converges.

$$t = \left\lceil \frac{1+\sqrt{\alpha}}{1+\sqrt{\alpha}-(1-\sqrt{\alpha})\bar{\beta}_t} \ln\left(\frac{2y_t}{\epsilon}\right) \right\rceil = \left\lceil \frac{1+\sqrt{\alpha}}{(2-c)\sqrt{\alpha}} \ln\left(\frac{2y_t}{\epsilon}\right) \right\rceil.$$

$\square$

## C.6 Implementation of LocCH

We present the implementation of LocCH as follows: Recall the sequence $\delta_{t+1} = \left(2\frac{1+\alpha}{1-\alpha} - \delta_t\right)^{-1}, t = 1, 2, \ldots$ with $\delta_1 = \frac{1-\alpha}{1+\alpha}$. Denote $\tilde{x}^{(t)} \triangleq x^{(t)} - x^{(t-1)}, \Delta^{(t)} := (1 + \delta_{t:t+1})r^{(t)} + \delta_{t:t+1}\tilde{x}^{(t)}$, we have

$$x^{(t+1)} = x^{(t)} + (1 + \delta_{t:t+1})r_{\mathcal{S}_t}^{(t)} + \delta_{t:t+1}\tilde{x}_{\mathcal{S}_t}^{(t)} = x^{(t)} + \Delta_{\mathcal{S}_t}^{(t)}$$

$$r^{(t+1)} = b - Q\left(x^{(t)} + (1 + \delta_{t:t+1})r_{\mathcal{S}_t}^{(t)} + \delta_{t:t+1}\tilde{x}_{\mathcal{S}_t}^{(t)}\right) = r^{(t)} - Q\Delta_{\mathcal{S}_t}^{(t)}$$

$$\tilde{x}^{(t+1)} = \tilde{x}^{(t)} + \Delta_{\mathcal{S}_t}^{(t)} - \Delta_{\mathcal{S}_{t-1}}^{(t-1)}.$$

- When $t = 0$, we have $x^{(0)} = 0$, $r^{(0)} = b$, $\tilde{x}^{(0)} = 0$, $\Delta^{(0)} = r^{(0)}$.
- When $t = 1$, we have $x^{(1)} = r_{\mathcal{S}_0}^{(0)}$, $r^{(1)} = \frac{1-\alpha}{1+\alpha}W\Delta_{\mathcal{S}_0}^{(0)}$, $\tilde{x}^{(1)} = r_{\mathcal{S}_0}^{(0)}$, $\Delta^{(1)} = (1 + \delta_{1:2})r^{(1)} + \delta_{1:2}\tilde{x}^{(1)}$.
- When $t \geq 1$, we can recursively calculate the following vectors

$$x^{(t+1)} = x^{(t)} + \Delta_{\mathcal{S}_t}^{(t)}$$

$$r^{(t+1)} = r^{(t)} - \Delta_{\mathcal{S}_t}^{(t)} + \frac{1-\alpha}{1+\alpha}W\Delta_{\mathcal{S}_t}^{(t)}$$

$$\tilde{x}^{(t+1)} = \tilde{x}^{(t)} + \Delta_{\mathcal{S}_t}^{(t)} - \Delta_{\mathcal{S}_{t-1}}^{(t-1)}.$$

Therefore, at per-iteration, we only need to save sub-vectors $\Delta_{\mathcal{S}_t}$ and $\Delta_{\mathcal{S}_{t-1}}$ and update $x$ locally.

# D  Local Heavy-Ball Method - LocHB

## D.1  Standard HB and Proof Theorem D.2

**Lemma D.1** (The standard HB updates). *The updates $x^{(t)}$ and $r^{(t)}$ of the HB method for solving Equ. (4) can be written as*

$$x^{(t+1)} = x^{(t)} + (1 + \tilde{\alpha}^2)r^{(t)} + \tilde{\alpha}^2(x^{(t)} - x^{(t-1)})$$

$$r^{(t+1)} = 2\tilde{\alpha}Wr^{(t)} - \tilde{\alpha}^2 r^{(t-1)}.$$

*The residual updates can be rewritten as a second-order homogeneous equation*

$$y^{(t+1)} - 2\Lambda y^{(t)} + y^{(t-1)} = 0, \quad \forall t = 1, 2, 3, \ldots$$

*where $y^{(t)}$ is such that $r^{(t)} = \tilde{\alpha}^t V y^{(t)}, t \geq 0$ with $y^{(0)} = V^\top r^{(0)} = V^\top b$.*

*Proof.* We follow the standard Polyak's heavy-ball method [40] as

$$x^{(t+1)} = x^{(t)} - \eta_\alpha \nabla f(x^{(t)}) + \eta_\beta(x^{(t)} - x^{(t-1)}),$$

where $\nabla f(x^{(t)}) = Qx^{(t)} - b$ and $\eta_\alpha = 4/(\sqrt{2/(1+\alpha)} + \sqrt{2\alpha/(1+\alpha)})^2 = 2(1+\alpha)/(1+\sqrt{\alpha})^2 = 1 + \tilde{\alpha}^2$ and $\eta_\beta = (\sqrt{2/(1+\alpha)} - \sqrt{2\alpha/(1+\alpha)})^2/(\sqrt{2/(1+\alpha)} + \sqrt{2\alpha/(1+\alpha)})^2 = \tilde{\alpha}^2$. Hence, it leads to the following updates

$$x^{(t+1)} = x^{(t)} + (1 + \tilde{\alpha}^2)r^{(t)} + \tilde{\alpha}^2(x^{(t)} - x^{(t-1)}).$$

Inserting

$$r^{(t)} = Q(x^* - x^{(t)}) = b - \left(I - \frac{1-\alpha}{1+\alpha}W\right)x^{(t)} = b - \left(I - \frac{2\tilde{\alpha}}{1+\tilde{\alpha}^2}W\right)x^{(t)}$$

then $x^{(t+1)} = 2\tilde{\alpha}Wx^{(t)} - \tilde{\alpha}^2 x^{(t-1)} + (1+\tilde{\alpha}^2)b$ and since

$$QW = \left(I - \frac{1-\alpha}{1+\alpha}W\right)W = W\left(I - \frac{1-\alpha}{1+\alpha}W\right) = WQ$$

So

$$
\begin{aligned}
r^{(t+1)} &= -Q(x^{(t+1)} - x^*) \\
&= -2\tilde{\alpha}QWx^{(t)} + \tilde{\alpha}^2 Qx^{(t-1)} + (I - (1+\tilde{\alpha}^2)Q)b \\
&= -2\tilde{\alpha}QWx^{(t)} + \tilde{\alpha}^2 Qx^{(t-1)} + \left(-\tilde{\alpha}^2 + 2\tilde{\alpha}W\right)b \\
&= 2\tilde{\alpha}Wr^{(t)} - \tilde{\alpha}^2 r^{(t-1)}
\end{aligned}
$$

Using $r^{(t)} = \tilde{\alpha}^t V y^{(t)}, t \geq 0$

$$\tilde{\alpha}^{t+1}Vy^{(t+1)} = 2\tilde{\alpha}W\tilde{\alpha}^t Vy^{(t)} - \tilde{\alpha}^2\tilde{\alpha}^{t-1}Vy^{(t-1)} \quad \Rightarrow \quad Vy^{(t+1)} = 2WVy^{(t)} - Vy^{(t-1)}.$$

As $W = V\Lambda V^\top$ and $V^\top = V^{-1}$ is orthogonal matrix, we continue to have

$$V^\top Vy^{(t+1)} - 2V^\top V\Lambda V^\top Vy^{(t)} + V^\top Vy^{(t-1)} = 0 \quad \Rightarrow \quad y^{(t+1)} - 2\Lambda y^{(t)} + y^{(t-1)} = 0.$$

$\square$

**Theorem D.2** (Convergence analysis of Heavy-Ball (HB)). *To solve the minimization problem in Equ. (4), we propose the following standard HB updates as*

$$x^{(t+1)} = x^{(t)} + (1+\tilde{\alpha}^2)r^{(t)} + \tilde{\alpha}^2\left(x^{(t)} - x^{(t-1)}\right), \qquad r^{(t+1)} = 2\tilde{\alpha}Wr^{(t)} - \tilde{\alpha}^2 r^{(t-1)},$$

*where the initial condition is $x^{(0)} = 0, r^{(0)} = b, x^{(1)} = x^{(0)} + \Gamma r^{(0)}, r^{(1)} = b - Qx^{(1)}$.
Then there exists a constant $\tau$ such that the total iteration complexity to reach the stop condition
$\{u : |r_u| \leq \epsilon d_u, u \in \mathcal{V}\} = \emptyset$ is*

$$t = \left\lceil \frac{1+\sqrt{\alpha}}{2\sqrt{\alpha}} \ln \frac{C_t \|r^{(0)}\|_2}{\epsilon} \right\rceil,$$

*where $C_t = 1$ if $\Gamma = Q^{-1}(I - \tilde{\alpha}W)$ (ideal case); $C_t = \max\left\{\frac{1+\tilde{\alpha}^{-1}}{\sqrt{1-\lambda_2^2}}, 1 + (1+\tilde{\alpha}^{-1})t\right\}$ if $\Gamma = 0$
(practical case); and $C_t = \frac{2}{\sqrt{1-\lambda_2^2}}$ if $\Gamma = \frac{(1-\tilde{\alpha})(1+\alpha)}{2}I$ ($\mathcal{G}$ is not bi-partite graph).*

*Proof.* Recall $W = V\Lambda V^\top$ and then $V^\top r^{(t+1)} = 2\tilde{\alpha}\Lambda V^\top r^{(t)} - \tilde{\alpha}^2 V^\top r^{(t-1)}$. By Lemma D.1, we have

$$y^{(t+1)} - 2\Lambda y^{(t)} + y^{(t-1)} = 0,$$

where we obtained $n$ second-order difference equations

$$y_i^{(t+1)} - 2\lambda_i y_i^{(t)} + y_i^{(t-1)} = 0, \quad \forall i = 1, 2, \ldots, n.$$

Follow the Lemma C.3, Equ. (23) has the solution

$$y_i^{(t)} = \begin{cases} \frac{\sin(\theta_i t)y_i^{(1)} - \sin(\theta_i(t-1))y_i^{(0)}}{\sin(\theta_i)} & |\lambda_i| < 1 \text{ where } \theta_i = \arccos(\lambda_i) \\ (y_i^{(0)} + (y_i^{(1)} - \lambda_i y_i^{(0)})t)\lambda_i^t & |\lambda_i| = 1, \end{cases} \tag{34}$$

where in the case of $|\lambda_i| < 1$. We consider the three cases of $\Gamma$

- **Ideal Case:** We can eliminate $t$ in (34), when $\boldsymbol{y}^{(1)} = \boldsymbol{\Lambda}\boldsymbol{y}^{(0)}$, we get $y_i^{(1)} = \lambda_i y_i^{(0)}$, and then $y_i^{(t)}$ can be simplified into

$$y_i^{(t)} = \begin{cases} \frac{(\lambda_i \sin(\theta_i t) - \sin(\theta_i(t-1)))y_i^{(0)}}{\sin\theta_i} & |\lambda_i| < 1 \\ y_i^{(0)}\lambda_i^t & |\lambda_i| = 1 \end{cases} = \begin{cases} y_i^{(0)}\cos(\theta_i t) & |\lambda_i| < 1 \\ y_i^{(0)}\lambda_i^t & |\lambda_i| = 1 \end{cases} \leq \begin{cases} |y_i^{(0)}| & |\lambda_i| < 1 \\ |y_i^{(0)}| & |\lambda_i| = 1 \end{cases}.$$

In this case, $\boldsymbol{\Gamma}$ needs to be $\boldsymbol{\Gamma} = \boldsymbol{Q}^{-1}(\boldsymbol{I} - \tilde{\alpha}\boldsymbol{W})$. Therefore, we have

$$\|\boldsymbol{V}^\top \boldsymbol{r}^{(t)}\|_2 = \|\boldsymbol{r}^{(t)}\|_2 = \tilde{\alpha}^t \|\boldsymbol{y}^{(t)}\|_2 \leq \tilde{\alpha}^t \|\boldsymbol{y}^{(0)}\|_2 = \tilde{\alpha}^t \|\boldsymbol{V}^\top \boldsymbol{r}^{(0)}\|_2 = \tilde{\alpha}^t \|\boldsymbol{r}^{(0)}\|_2.$$

- **Practical Case:** Just letting $\boldsymbol{x}^{(1)} = \boldsymbol{x}^{(0)} = \boldsymbol{0}$, we have $\tilde{\alpha}\boldsymbol{y}^{(1)} = \boldsymbol{y}^{(0)}$, then

$$y_i^{(t)} = \begin{cases} \frac{\tilde{\alpha}^{-1}\sin(\theta_i t) - \sin(\theta_i(t-1))}{\sin(\theta_i)}y_i^{(0)} & |\lambda_i| < 1 \\ (1 + (\tilde{\alpha}^{-1} - \lambda_i)t)y_i^{(0)}\lambda_i^t & |\lambda_i| = 1 \end{cases} \leq \max\left\{\frac{1 + \tilde{\alpha}^{-1}}{\sqrt{1 - \lambda_2^2}}, 1 + (1 + \tilde{\alpha}^{-1})t\right\}|y_i^{(0)}|,$$

where $\theta_i = \arccos(\lambda_i)$.

- **Non-bipartite graph Case**: When the graph is non-bipartite, we can eliminate $t$, as the following: We choose $\boldsymbol{\Gamma} = \tau\boldsymbol{I}$, we have

$$\boldsymbol{x}^{(1)} = \boldsymbol{x}^{(0)} + \boldsymbol{\Gamma}\boldsymbol{r}^{(0)} = \boldsymbol{\Gamma}\boldsymbol{r}^{(0)}, \qquad \boldsymbol{r}^{(1)} = \boldsymbol{b} - \boldsymbol{Q}\boldsymbol{x}^{(1)} = \boldsymbol{r}^{(0)} - \tau\boldsymbol{Q}\boldsymbol{r}^{(0)}$$

$$\boldsymbol{V}^\top\boldsymbol{r}^{(1)} = (1 - \tau)\boldsymbol{V}^\top\boldsymbol{r}^{(0)} + \frac{(1-\alpha)\tau}{1+\alpha}\boldsymbol{\Lambda}\boldsymbol{V}^\top\boldsymbol{r}^{(0)}, \qquad \boldsymbol{v}_i^\top\boldsymbol{r}^{(1)} = (1 - \tau + \frac{(1-\alpha)\tau}{1+\alpha}\lambda_i)\boldsymbol{v}_i^\top\boldsymbol{r}^{(0)}$$

We have the following relations

$$\boldsymbol{v}_i^\top\boldsymbol{r}^{(0)} = y_i^{(0)}$$

$$\boldsymbol{v}_i^\top\boldsymbol{r}^{(1)} = \tilde{\alpha}y_i^{(1)} = (1 - \tau + \frac{(1-\alpha)\tau}{1+\alpha}\lambda_i)\boldsymbol{v}_i^\top\boldsymbol{r}^{(0)} = (1 - \tau + \frac{(1-\alpha)\tau\lambda_i}{1+\alpha})y_i^{(0)},$$

To make $t$ disappear when $\lambda_i = 1$, we need $y_i^{(0)} = y_i^{(1)}$, or

$$1 - \tau + \frac{(1-\alpha)\tau}{1+\alpha} = \frac{1 - \sqrt{\alpha}}{1 + \sqrt{\alpha}} \iff \tau = \frac{1+\alpha}{\alpha + \sqrt{\alpha}}.$$

In this case, we have

$$|y_i^{(t)}| \leq \frac{2}{\sqrt{1 - \lambda_2^2}}|y_i^{(0)}|$$

To make sure the algorithm stops when the stop condition is met, it is enough for

$$\|\boldsymbol{r}^{(t)}\|_2 = \tilde{\alpha}^t\|\boldsymbol{y}^{(t)}\|_2 \leq \tilde{\alpha}^t C_t\|\boldsymbol{y}^{(0)}\|_2 = \tilde{\alpha}^t C_t\|\boldsymbol{V}^\top\boldsymbol{r}^{(0)}\|_2 = \tilde{\alpha}^t C_t\|\boldsymbol{r}^{(0)}\|_2 \leq \epsilon.$$

This means $C_t\|\boldsymbol{r}^{(0)}\|_2\tilde{\alpha}^t \leq \epsilon$, which leads to the following

$$t = \left\lceil \frac{1 + \sqrt{\alpha}}{2\sqrt{\alpha}}\ln\frac{C_t\|\boldsymbol{r}^{(0)}\|_2}{\epsilon} \right\rceil.$$

$\square$

*Remark* D.3. The constant that appears in the bound involves the second largest eigenvalue $\lambda_2$ of $\boldsymbol{A}\boldsymbol{D}^{-1}$. It is deeply related to the mixing time of random walk [8] where the second largest eigenvalue determines the mixing time of the walk. A smaller absolute value of the second largest eigenvalue indicates that a random walk on the graph will mix (i.e., approach its steady-state distribution) more quickly. Our proof is partially inspired by d'Aspremont et al. [11] where we directly bound $\boldsymbol{r}^{(t)}$ instead of providing bound for $\boldsymbol{e}^{(t)}$.

## D.2 Residual Updates of LOCHB and Proof of Theorem D.6

**Lemma D.4.** *Let the local heavy-ball method be defined as*

$$\boldsymbol{x}^{(t+1)} = \boldsymbol{x}^{(t)} + \boldsymbol{\Delta}_{\mathcal{S}_t}^{(t)}, \quad \boldsymbol{r}^{(t+1)} = \boldsymbol{r}^{(t)} - \boldsymbol{Q}\boldsymbol{\Delta}_{\mathcal{S}_t}^{(t)}, \quad \boldsymbol{\Delta}^{(t)} = (1 + \tilde{\alpha}^2)\boldsymbol{r}^{(t)} + \tilde{\alpha}^2 \big(\boldsymbol{x}^{(t)} - \boldsymbol{x}^{(t-1)}\big),$$

*where $\boldsymbol{x}^{(0)} = \boldsymbol{0}, \boldsymbol{x}^{(1)} = \boldsymbol{\Gamma}\boldsymbol{r}^{(0)}$ and $\boldsymbol{\Gamma} = \mathbf{diag}(\Gamma_1, \Gamma_2, \ldots, \Gamma_n)$ is initial step size matrix. We have the following expanding sequence*

$$\tilde{\alpha}^2 (\boldsymbol{x}^{(t)} - \boldsymbol{x}^{(t-1)})_{\overline{\mathcal{S}}_t} = (1 + \tilde{\alpha}^2) \sum_{i=1}^{t-1} \tilde{\alpha}^{2(t-i)} \boldsymbol{r}_{\mathcal{S}_{i:t-1} \cap \overline{\mathcal{S}}_t}^{(i)} + \tilde{\alpha}^{2t} \boldsymbol{\Gamma} \boldsymbol{r}_{\mathcal{S}_{0:t-1} \cap \overline{\mathcal{S}}_t}^{(0)}, \qquad \forall t \geq 1$$

$$\boldsymbol{\Delta}_{\overline{\mathcal{S}}_t}^{(t)} = (1 + \tilde{\alpha}^2) \sum_{i=1}^{t} \tilde{\alpha}^{2(t-i)} \boldsymbol{r}_{\mathcal{S}_{i:t-1} \cap \overline{\mathcal{S}}_t}^{(i)} + \tilde{\alpha}^{2t} \boldsymbol{\Gamma} \boldsymbol{r}_{\mathcal{S}_{0:t-1} \cap \overline{\mathcal{S}}_t}^{(0)}, \qquad \forall t \geq 1.$$

*Furthermore, we have the following sequence*

$$\tilde{\alpha}^2 \Big(\boldsymbol{V}^\top - \frac{1-\alpha}{1+\alpha} \boldsymbol{\Lambda} \boldsymbol{V}^\top\Big)\big(\boldsymbol{x}^{(t)} - \boldsymbol{x}^{(t-1)}\big)_{\overline{\mathcal{S}}_t} =$$

$$(1 + \tilde{\alpha}^2) \sum_{i=1}^{t-1} \tilde{\alpha}^{2(t-i)} \Big(\boldsymbol{V}^\top - \frac{1-\alpha}{1+\alpha} \boldsymbol{\Lambda} \boldsymbol{V}^\top\Big) \boldsymbol{r}_{\overline{\mathcal{S}}_{i,t}}^{(i)} + \boldsymbol{\Gamma} \tilde{\alpha}^{2t} \Big(\boldsymbol{V}^\top - \frac{1-\alpha}{1+\alpha} \boldsymbol{\Lambda} \boldsymbol{V}^\top\Big) \boldsymbol{r}_{\overline{\mathcal{S}}_{0,t}}^{(0)}.$$

*where we denote $\overline{\mathcal{S}}_{i,t} \triangleq \mathcal{S}_{i:t-1} \cap \overline{\mathcal{S}}_t$.*

*Proof.* We assume all nonzeros in $\boldsymbol{b}$ are active nodes at time $t = 0$ and $t = 1$, i.e., $\mathcal{S}_0 = \boldsymbol{r}^{(0)} = \mathrm{supp}(\boldsymbol{b})$. The local updates can be expressed as

$$\boldsymbol{x}^{(t+1)} = \boldsymbol{x}^{(t)} + (1 + \tilde{\alpha}^2) \boldsymbol{r}_{\mathcal{S}_t}^{(t)} + \tilde{\alpha}^2 \big(\boldsymbol{x}^{(t)} - \boldsymbol{x}^{(t-1)}\big)_{\mathcal{S}_t}$$

$$\boldsymbol{r}^{(t+1)} = \boldsymbol{b} - \boldsymbol{Q}\boldsymbol{x}^{(t+1)}$$

$$= \underbrace{\boldsymbol{r}^{(t)} - (1 + \tilde{\alpha}^2) \boldsymbol{Q}\boldsymbol{r}^{(t)} - \tilde{\alpha}^2 \boldsymbol{Q}\big(\boldsymbol{x}^{(t)} - \boldsymbol{x}^{(t-1)}\big)}_{\text{original updates}}$$

$$+ \underbrace{(1 + \tilde{\alpha}^2) \boldsymbol{Q}\boldsymbol{r}_{\overline{\mathcal{S}}_t}^{(t)} + \tilde{\alpha}^2 \boldsymbol{Q}\big(\boldsymbol{x}^{(t)} - \boldsymbol{x}^{(t-1)}\big)_{\overline{\mathcal{S}}_t}}_{\text{noisy with small magnitudes}}$$

$$= 2\tilde{\alpha}\boldsymbol{W}\boldsymbol{r}^{(t)} - \tilde{\alpha}^2 \boldsymbol{r}^{(t-1)} + \underbrace{(1 + \tilde{\alpha}^2) \boldsymbol{Q}\boldsymbol{r}_{\overline{\mathcal{S}}_t}^{(t)} + \tilde{\alpha}^2 \boldsymbol{Q}\big(\boldsymbol{x}^{(t)} - \boldsymbol{x}^{(t-1)}\big)_{\overline{\mathcal{S}}_t}}_{\text{noisy with small magnitudes}}$$

$$= 2\tilde{\alpha}\boldsymbol{W}\boldsymbol{r}^{(t)} - \tilde{\alpha}^2 \boldsymbol{r}^{(t-1)} + (1 + \tilde{\alpha}^2)\Big(\boldsymbol{I} - \frac{1-\alpha}{1+\alpha}\boldsymbol{W}\Big)\boldsymbol{r}_{\overline{\mathcal{S}}_t}^{(t)} + \tilde{\alpha}^2 \boldsymbol{Q}\big(\boldsymbol{x}^{(t)} - \boldsymbol{x}^{(t-1)}\big)_{\overline{\mathcal{S}}_t}$$

$$\boldsymbol{r}^{(t+1)} - 2\tilde{\alpha}\boldsymbol{W}\boldsymbol{r}^{(t)} + \tilde{\alpha}^2 \boldsymbol{r}^{(t-1)} = (1 + \tilde{\alpha}^2)\boldsymbol{r}_{\overline{\mathcal{S}}_t}^{(t)} - 2\tilde{\alpha}\boldsymbol{W}\boldsymbol{r}_{\overline{\mathcal{S}}_t}^{(t)} + \tilde{\alpha}^2 \boldsymbol{Q}\big(\boldsymbol{x}^{(t)} - \boldsymbol{x}^{(t-1)}\big)_{\overline{\mathcal{S}}_t}$$

For $t \geq 1$, we can expand $\boldsymbol{x}^{(t+1)} - \boldsymbol{x}^{(t)}$ as the following

$$\boldsymbol{x}^{(t+1)} = \boldsymbol{x}^{(t)} + (1+\tilde{\alpha}^2)\boldsymbol{r}_{\mathcal{S}_t}^{(t)} + \tilde{\alpha}^2\big(\boldsymbol{x}^{(t)} - \boldsymbol{x}^{(t-1)}\big)_{\mathcal{S}_t}$$

$$\begin{aligned}
\boldsymbol{\Delta}^{(t)} = \boldsymbol{x}^{(t+1)} - \boldsymbol{x}^{(t)} &= (1+\tilde{\alpha}^2)\boldsymbol{r}_{\mathcal{S}_t}^{(t)} + \tilde{\alpha}^2\big((1+\tilde{\alpha}^2)\boldsymbol{r}_{\mathcal{S}_{t-1}}^{(t-1)} + \tilde{\alpha}^2\big(\boldsymbol{x}^{(t-1)} - \boldsymbol{x}^{(t-2)}\big)_{\mathcal{S}_{t-1}}\big)_{\mathcal{S}_t} \\
&= (1+\tilde{\alpha}^2)\boldsymbol{r}_{\mathcal{S}_t}^{(t)} + \tilde{\alpha}^2(1+\tilde{\alpha}^2)\boldsymbol{r}_{\mathcal{S}_{t-1:t}}^{(t-1)} + \tilde{\alpha}^4\big(\boldsymbol{x}^{(t-1)} - \boldsymbol{x}^{(t-2)}\big)_{\mathcal{S}_{t-1:t}} \\
&= (1+\tilde{\alpha}^2)\boldsymbol{r}_{\mathcal{S}_t}^{(t)} + \tilde{\alpha}^2(1+\tilde{\alpha}^2)\boldsymbol{r}_{\mathcal{S}_{t-1:t}}^{(t-1)} + \tilde{\alpha}^4\big((1+\tilde{\alpha}^2)\boldsymbol{r}_{\mathcal{S}_{t-2}}^{(t-2)} \\
&\quad + \tilde{\alpha}^2\big(\boldsymbol{x}^{(t-2)} - \boldsymbol{x}^{(t-3)}\big)_{\mathcal{S}_{t-2}}\big)_{\mathcal{S}_{t-1:t}} \\
&= (1+\tilde{\alpha}^2)\boldsymbol{r}_{\mathcal{S}_t}^{(t)} + \tilde{\alpha}^2(1+\tilde{\alpha}^2)\boldsymbol{r}_{\mathcal{S}_{t-1:t}}^{(t-1)} + \tilde{\alpha}^4(1+\tilde{\alpha}^2)\boldsymbol{r}_{\mathcal{S}_{t-2:t}}^{(t-2)} \\
&\quad + \tilde{\alpha}^6\big(\boldsymbol{x}^{(t-2)} - \boldsymbol{x}^{(t-3)}\big)_{\mathcal{S}_{t-2:t}} \\
&= (1+\tilde{\alpha}^2)\sum_{i=t-2}^{t}\tilde{\alpha}^{2(t-i)}\boldsymbol{r}_{\mathcal{S}_{i:t}}^{(i)} + \tilde{\alpha}^6\big(\boldsymbol{x}^{(t-2)} - \boldsymbol{x}^{(t-3)}\big)_{\mathcal{S}_{t-2:t}} \\
&= (1+\tilde{\alpha}^2)\sum_{i=1}^{t}\tilde{\alpha}^{2(t-i)}\boldsymbol{r}_{\mathcal{S}_{i:t}}^{(i)} + \tilde{\alpha}^{2t}\big(\boldsymbol{x}^{(1)} - \boldsymbol{x}^{(0)}\big)_{\mathcal{S}_{1:t}} \\
&= (1+\tilde{\alpha}^2)\sum_{i=1}^{t}\tilde{\alpha}^{2(t-i)}\boldsymbol{r}_{\mathcal{S}_{i:t}}^{(i)} + \tilde{\alpha}^{2t}\boldsymbol{\Gamma}\boldsymbol{r}_{\mathcal{S}_{1:t}}^{(0)}.
\end{aligned}$$

Note $\mathcal{S}_0 = \mathrm{supp}(\boldsymbol{r}^{(0)})$, then $\boldsymbol{r}_{\mathcal{S}_{1:t}}^{(0)} = \boldsymbol{r}_{\mathcal{S}_{0:t}}^{(0)}$, we continue to have

$$\boldsymbol{x}^{(t)} - \boldsymbol{x}^{(t-1)} = (1+\tilde{\alpha}^2)\sum_{i=1}^{t-1}\tilde{\alpha}^{2(t-i-1)}\boldsymbol{r}_{\mathcal{S}_{i:t-1}}^{(i)} + \boldsymbol{\Gamma}\tilde{\alpha}^{2(t-1)}\boldsymbol{r}_{\mathcal{S}_{0:t-1}}^{(0)}, \qquad \forall t \geq 1$$

$$\tilde{\alpha}^2(\boldsymbol{x}^{(t)} - \boldsymbol{x}^{(t-1)})_{\overline{\mathcal{S}}_t} = (1+\tilde{\alpha}^2)\sum_{i=1}^{t-1}\tilde{\alpha}^{2(t-i)}\boldsymbol{r}_{\mathcal{S}_{i:t-1}\cap\overline{\mathcal{S}}_t}^{(i)} + \boldsymbol{\Gamma}\tilde{\alpha}^{2t}\boldsymbol{r}_{\mathcal{S}_{0:t-1}\cap\overline{\mathcal{S}}_t}^{(0)}, \qquad \forall t \geq 1$$

The rest follows readily. $\qquad\qquad\square$

**Lemma D.5** (The nonhomogeneous difference equation). *Given $\boldsymbol{y}^{(1)} = \boldsymbol{\Lambda}\boldsymbol{y}^{(0)}$, equations*

$$\boldsymbol{y}^{(t+1)} - 2\boldsymbol{\Lambda}\boldsymbol{y}^{(t)} + \boldsymbol{y}^{(t-1)} := \boldsymbol{f}^{(t)}.$$

*have the following solutions*

$$\boldsymbol{y}^{(t)} = \boldsymbol{Z}_t\boldsymbol{y}^{(0)} + \sum_{k=1}^{t-1}\boldsymbol{H}_{k,t}\boldsymbol{f}^{(k)},$$

*where*

$$\boldsymbol{Z}_t = \begin{cases} \mathbf{diag}\left(1, \ldots, \cos(\theta_i t), \ldots, (-1)^t\right) & \textit{for bipartite graphs} \\ \mathbf{diag}\left(1, \ldots, \cos(\theta_i t), \ldots, \cos(\theta_n t)\right) & \textit{for non-bipartite graphs,} \end{cases}$$

$$\boldsymbol{H}_{k,t} = \begin{cases} \mathbf{diag}\left(t-k, \ldots, \frac{\sin(\theta_i(t-k))}{\sin\theta_i}, \ldots, (-1)^{t-k-1}(t-k)\right) & \textit{for bipartite graphs} \\ \mathbf{diag}\left(t-k, \ldots, \frac{\sin(\theta_i(t-k))}{\sin\theta_i}, \ldots, \frac{\sin(\theta_n(t-k))}{\sin\theta_n}\right) & \textit{for non-bipartite graphs.} \end{cases}$$

*Proof.* This directly follows from Lemma C.4. $\qquad\qquad\square$

**Theorem D.6** (Representation of $\boldsymbol{r}^{(t)}$ for LOCHB). *Given $t \geq 1$, $\boldsymbol{x}^{(0)} = \boldsymbol{0}$ and $\boldsymbol{x}^{(1)} = \boldsymbol{\Gamma} \boldsymbol{r}_{\mathcal{S}_0}^{(0)}$. The residual of $\boldsymbol{r}^{(t)}$ of LOCHB satisfies*

$$\boldsymbol{V}^\top \boldsymbol{r}^{(t)} = \tilde{\alpha}^t \boldsymbol{Z}_t \boldsymbol{V}^\top \boldsymbol{r}^{(0)} + \underbrace{\tilde{\alpha}^t t \sum_{k=1}^{t-1} \tilde{\alpha}^{k-1} \boldsymbol{H}_{k,t} \boldsymbol{V}^\top \boldsymbol{Q} \boldsymbol{\Gamma} \boldsymbol{r}_{\mathcal{S}_{0,k}}^{(0)} \Big/ t}_{\boldsymbol{u}_{0,t}}$$

$$+ 2 \sum_{k=1}^{t-1} \tilde{\alpha}^{t-k} (t-k) \underbrace{\sum_{j=k}^{t-1} \tilde{\alpha}^{j-k} \boldsymbol{H}_{j,t} \left( \frac{1+\alpha}{1-\alpha} - \boldsymbol{\Lambda} \right) \boldsymbol{V}^\top \boldsymbol{r}_{\mathcal{S}_{k,j}}^{(k)} \Big/ (t-k)}_{\boldsymbol{u}_{k,t}} .$$

*Proof.* Follow Lemma D.4, we have

$$
\begin{aligned}
\boldsymbol{r}^{(t+1)} - \boldsymbol{Q} \boldsymbol{\Delta}_{\bar{\mathcal{S}}_t}^{(t)} &= \boldsymbol{r}^{(t)} - \boldsymbol{Q} \boldsymbol{\Delta}^{(t)} \\
&= \boldsymbol{r}^{(t)} - (1+\tilde{\alpha}^2) \boldsymbol{Q} \boldsymbol{r}^{(t)} - \tilde{\alpha}^2 \boldsymbol{Q} (\boldsymbol{x}^{(t)} - \boldsymbol{x}^{(t-1)}) \\
&= -\tilde{\alpha}^2 \boldsymbol{r}^{(t)} + 2\tilde{\alpha} \boldsymbol{W} \boldsymbol{r}^{(t)} - \tilde{\alpha}^2 \boldsymbol{Q} (\boldsymbol{x}^{(t)} - \boldsymbol{x}^{(t-1)}) \\
&= 2\tilde{\alpha} \boldsymbol{W} \boldsymbol{r}^{(t)} - \tilde{\alpha}^2 \boldsymbol{r}^{(t-1)}
\end{aligned}
$$

So we can write the updates of LOCHB as

$$
\begin{aligned}
\boldsymbol{r}^{(t+1)} - 2\tilde{\alpha} \boldsymbol{W} \boldsymbol{r}^{(t)} + \tilde{\alpha}^2 \boldsymbol{r}^{(t-1)} &= \boldsymbol{Q} \boldsymbol{\Delta}_{\bar{\mathcal{S}}_t}^{(t)} \\
&= (1+\tilde{\alpha}^2) \sum_{i=1}^{t} \tilde{\alpha}^{2(t-i)} \boldsymbol{Q} \boldsymbol{r}_{\bar{\mathcal{S}}_{i,t}}^{(i)} + \tilde{\alpha}^{2t} \boldsymbol{Q} \boldsymbol{\Gamma} \boldsymbol{r}_{\bar{\mathcal{S}}_{0,t}}^{(0)} .
\end{aligned}
$$

Write $\boldsymbol{V}^\top \boldsymbol{r}^{(t)} = \tilde{\alpha}^t \boldsymbol{y}^{(t)}$, and note

$$\boldsymbol{r}^{(t)} = \tilde{\alpha}^t \boldsymbol{V} \boldsymbol{y}^{(t)}, \quad \boldsymbol{r}_{\bar{\mathcal{S}}_t}^{(t)} = \tilde{\alpha}^t \left( \boldsymbol{V} \boldsymbol{y}^{(t)} \right)_{\bar{\mathcal{S}}_t} \quad \Rightarrow \quad \boldsymbol{V}^\top \boldsymbol{r}_{\bar{\mathcal{S}}_t}^{(t)} = \tilde{\alpha}^t \boldsymbol{V}^\top \left( \boldsymbol{V} \boldsymbol{y}^{(t)} \right)_{\bar{\mathcal{S}}_t} .$$

Hence,

$$\boldsymbol{V}^\top \boldsymbol{r}^{(t+1)} - 2\tilde{\alpha} \boldsymbol{V}^\top \boldsymbol{W} \boldsymbol{r}^{(t)} + \tilde{\alpha}^2 \boldsymbol{V}^\top \boldsymbol{r}^{(t-1)}$$

$$= (1+\tilde{\alpha}^2) \sum_{i=1}^{t} \tilde{\alpha}^{2(t-i)} \boldsymbol{V}^\top \boldsymbol{Q} \boldsymbol{r}_{\bar{\mathcal{S}}_{i,t}}^{(i)} + \tilde{\alpha}^{2t} \boldsymbol{V}^\top \boldsymbol{Q} \boldsymbol{\Gamma} \boldsymbol{r}_{\bar{\mathcal{S}}_{0,t}}^{(0)}$$

$$\Longleftrightarrow \tilde{\alpha}^{t+1} \boldsymbol{y}^{(t+1)} - 2\tilde{\alpha} \tilde{\alpha}^t \boldsymbol{W} \boldsymbol{y}^{(t)} + \tilde{\alpha}^2 \tilde{\alpha}^{t-1} \boldsymbol{y}^{(t-1)}$$

$$= (1+\tilde{\alpha}^2) \sum_{i=1}^{t} \tilde{\alpha}^{2(t-i)} \tilde{\alpha}^i \boldsymbol{Q} \boldsymbol{V}^\top (\boldsymbol{V}^\top \boldsymbol{y}^{(i)})_{\bar{\mathcal{S}}_{i,t}} + \tilde{\alpha}^{2t} \boldsymbol{Q} \boldsymbol{\Gamma} \boldsymbol{V}^\top (\boldsymbol{V} \boldsymbol{y}^{(0)})_{\bar{\mathcal{S}}_{0,t}}$$

$$\Longleftrightarrow \boldsymbol{y}^{(t+1)} - 2\boldsymbol{W} \boldsymbol{y}^{(t)} + \boldsymbol{y}^{(t-1)}$$

$$= \overbrace{\sum_{i=1}^{t} \underbrace{(1+\tilde{\alpha}^2) \tilde{\alpha}^{t-1} \tilde{\alpha}^{-i} \boldsymbol{Q} \boldsymbol{V}^\top (\boldsymbol{V} \boldsymbol{y}^{(i)})_{\mathcal{S}_{i,t}}}_{\boldsymbol{f}_i^{(t)}} + \underbrace{\tilde{\alpha}^{t-1} \boldsymbol{Q} \boldsymbol{\Gamma} \boldsymbol{V}^\top (\boldsymbol{V} \boldsymbol{y}^{(0)})_{\mathcal{S}_{0,t}}}_{\boldsymbol{f}_0^{(t)}}}^{\boldsymbol{f}^{(t)}} .$$

Then, from Lemma D.5

$$\boldsymbol{y}^{(t)} = \boldsymbol{Z}_t \boldsymbol{y}^{(0)} + \sum_{k=1}^{t-1} \boldsymbol{H}_{k,t} \boldsymbol{f}^{(k)} \iff \boldsymbol{V}^\top \boldsymbol{r}^{(t)} = \tilde{\alpha}^t \boldsymbol{Z}_t \boldsymbol{V}^\top \boldsymbol{r}^{(0)} + \tilde{\alpha}^t \sum_{k=1}^{t-1} \boldsymbol{H}_{k,t} \boldsymbol{f}^{(k)}$$

so expanding the error term

$$\tilde{\alpha}^t \boldsymbol{H}_{k,t}\boldsymbol{f}^{(k)} = \sum_{i=1}^{k}\tilde{\alpha}^t \boldsymbol{H}_{k,t}\boldsymbol{f}_i^{(k)} + \tilde{\alpha}^t \boldsymbol{H}_{k,t}\boldsymbol{f}_0^{(k)}$$

$$= (1+\tilde{\alpha}^2)\tilde{\alpha}^{t+k-1-i}\sum_{i=1}^{k}\boldsymbol{H}_{k,t}\boldsymbol{Q}\boldsymbol{V}^\top(\boldsymbol{V}\boldsymbol{y}^{(i)})_{\mathcal{S}_{i,k}}$$

$$+ \tilde{\alpha}^{t+k-1}\boldsymbol{H}_{k,t}\boldsymbol{Q}\boldsymbol{\Gamma}\boldsymbol{V}^\top(\boldsymbol{V}\boldsymbol{y}^{(0)})_{\mathcal{S}_{0,k}}$$

$$= (1+\tilde{\alpha}^2)\tilde{\alpha}^{t+k-1-2i}\sum_{i=1}^{k}\boldsymbol{H}_{k,t}\boldsymbol{Q}\boldsymbol{V}^\top\boldsymbol{r}_{\mathcal{S}_{i,k}}^{(i)} + \tilde{\alpha}^{t+k-1}\boldsymbol{H}_{k,t}\boldsymbol{Q}\boldsymbol{\Gamma}\boldsymbol{V}^\top\boldsymbol{r}_{\mathcal{S}_{0,k}}^{(0)}$$

$$= 2\sum_{i=1}^{k}\tilde{\alpha}^{t+k-2i}\boldsymbol{H}_{k,t}(\frac{1+\alpha}{1-\alpha})\boldsymbol{Q}\boldsymbol{V}^\top\boldsymbol{r}_{\mathcal{S}_{i,k}}^{(i)} + \tilde{\alpha}^{t+k-1}\boldsymbol{H}_{k,t}\boldsymbol{Q}\boldsymbol{\Gamma}\boldsymbol{V}^\top\boldsymbol{r}_{\mathcal{S}_{0,k}}^{(0)}$$

$$\sum_{k=1}^{t-1}\tilde{\alpha}^t \boldsymbol{H}_{k,t}\boldsymbol{f}^{(k)} = \sum_{k=1}^{t-1}(2\sum_{i=1}^{k}\tilde{\alpha}^{t+k-2i}\boldsymbol{H}_{k,t}(\frac{1+\alpha}{1-\alpha})\boldsymbol{Q}\boldsymbol{V}^\top\boldsymbol{r}_{\mathcal{S}_{i,k}}^{(i)} + \tilde{\alpha}^{t+k-1}\boldsymbol{H}_{k,t}\boldsymbol{Q}\boldsymbol{\Gamma}\boldsymbol{V}^\top\boldsymbol{r}_{\mathcal{S}_{0,k}}^{(0)})$$

$$= 2\sum_{k=1}^{t-1}\sum_{j=k}^{t-1}\tilde{\alpha}^{t+j-2k}\boldsymbol{H}_{j,t}(\frac{1+\alpha}{1-\alpha})\boldsymbol{Q}\boldsymbol{V}^\top\boldsymbol{r}_{\mathcal{S}_{k,j}}^{(k)} + \sum_{k=1}^{t-1}\tilde{\alpha}^{t+k-1}\boldsymbol{H}_{k,t}\boldsymbol{Q}\boldsymbol{\Gamma}\boldsymbol{V}^\top\boldsymbol{r}_{\mathcal{S}_{0,k}}^{(0)}$$

which when simplified, gives the relation proposed. $\qquad\square$

## D.3 Convergence of LOCHB of Proof of Theorem D.8

**Corollary D.7.** *Define*

$$\beta_{k,t} \triangleq \|\boldsymbol{u}_{k,t}\|_2/\|\boldsymbol{r}^{(k)}\|_2, \qquad \beta_k \triangleq \max_t \beta_{k,t}.$$

*Then the upper bound of $\|\boldsymbol{r}^{(t)}\|_2$ can be characterized as*

$$\|\boldsymbol{r}^{(t)}\|_2 \le \tilde{\alpha}^t \prod_{j=0}^{t-1}(1+\beta_j)y_t, \tag{35}$$

*where $y_{t+1} - 2y_t + y_{t-1}/((1+\beta_{t-1})(1+\beta_t)) = 0$ where $y_0 = y_1 = \|\boldsymbol{r}^{(0)}\|_2$.*

*Proof.* Since $\|\boldsymbol{u}_{k,t}\|_2 \le \beta_k\|\boldsymbol{r}^{(k)}\|_2$, then given the final iterative updates (29)

$$\boldsymbol{V}^\top\boldsymbol{r}^{(t)} = \tilde{\alpha}^t\boldsymbol{Z}_t\boldsymbol{V}^\top\boldsymbol{r}^{(0)} + \tilde{\alpha}^t t\boldsymbol{u}_{0,t} + 2\sum_{k=1}^{t-1}\tilde{\alpha}^{t-k}(t-k)\boldsymbol{u}_{k,t}$$

and since $\|\boldsymbol{Z}_t\|_2 \le 1$ we can bound

$$\|\boldsymbol{r}^{(t)}\|_2 \le \tilde{\alpha}^t\|\boldsymbol{r}^{(0)}\|_2 + \tilde{\alpha}^t t\beta_0\|\boldsymbol{r}^{(0)}\|_2 + 2\sum_{k=1}^{t-1}\tilde{\alpha}^{t-k-1}(t-k)\beta_k\|\boldsymbol{r}^{(k)}\|_2$$

$$\iff \|\boldsymbol{r}^{(t)}\|_2 - 2\sum_{k=1}^{t-1}\tilde{\alpha}^{t-k-1}(t-k)\beta_k\|\boldsymbol{r}^{(k)}\|_2 \le \tilde{\alpha}^t(1+t\beta_0)\|\boldsymbol{r}^{(0)}\|_2, \tag{36}$$

where $t = 0, 1, \ldots, T$. The rest just follows a similar strategy shown in Corollary C.9. We have the following inequalities

$$\boldsymbol{I} - \boldsymbol{V}_L := \begin{pmatrix} 1 & 0 & 0 & \cdots & 0 \\ -v_{21} & 1 & 0 & \cdots & 0 \\ -v_{31} & -v_{32} & 1 & \cdots & 0 \\ \vdots & \vdots & \vdots & \ddots & \vdots \\ -v_{n1} & -v_{n2} & -v_{n3} & \cdots & 1 \end{pmatrix} \begin{pmatrix} \|\boldsymbol{r}^{(1)}\|_2 \\ \|\boldsymbol{r}^{(2)}\|_2 \\ \|\boldsymbol{r}^{(3)}\|_2 \\ \vdots \\ \|\boldsymbol{r}^{(T)}\|_2 \end{pmatrix} \le \begin{pmatrix} \tilde{\alpha}^1(1+\beta_0)\|\boldsymbol{r}^{(0)}\|_2 \\ \tilde{\alpha}^2(1+2\beta_0)\|\boldsymbol{r}^{(0)}\|_2 \\ \tilde{\alpha}^3(1+3\beta_0)\|\boldsymbol{r}^{(0)}\|_2 \\ \vdots \\ \tilde{\alpha}^T(1+T\beta_0)\|\boldsymbol{r}^{(0)}\|_2 \end{pmatrix} := \boldsymbol{c},$$

where $(V_L)_{tk} = 2\tilde{\alpha}^{t-k}(t-k)\beta_k$. Denote each upper bound as $\tau_t = \|r^{(t)}\|_2$, we will have

$$\tau_t = c_t + \sum_{k=1}^{t-1} v_{t,k} = \tilde{\alpha}^t(1+t\beta_0)\tau_0 + 2\sum_{k=1}^{t-1}(t-k)\tilde{\alpha}^{t-k}\beta_k\tau_k$$

$$\tau_{t+1} = c_{t+1} + \sum_{k=1}^{t} v_{t+1,k} = \tilde{\alpha}^{t+1}(1+(t+1)\beta_0)\tau_0 + 2\sum_{k=1}^{t}(t+1-k)\tilde{\alpha}^{t-k+1}\beta_k\tau_k$$

$$= \tilde{\alpha}^{t+1}(1+(t+1)\beta_0)\tau_0 + 2\sum_{k=1}^{t-1}(t-k)\tilde{\alpha}^{t-k+1}\beta_k\tau_k + 2\sum_{k=1}^{t}\tilde{\alpha}^{t-k+1}\beta_k\tau_k$$

$$\tilde{\alpha}\tau_t = \tilde{\alpha}^{t+1}(1+t\beta_0)\tau_0 + 2\sum_{k=1}^{t-1}(t-k)\tilde{\alpha}^{t-k+1}\beta_k\tau_k$$

$$\tau_{t+1} - \tilde{\alpha}\tau_t = \tilde{\alpha}^{t+1}\beta_0\tau_0 + 2\sum_{k=1}^{t}\tilde{\alpha}^{t-k+1}\beta_k\tau_k$$

$$\tau_{t-1} = \tilde{\alpha}^{t-1}(1+(t-1)\beta_0)\tau_0 + 2\sum_{k=1}^{t-2}(t-k-1)\tilde{\alpha}^{t-k-1}\beta_k\tau_k$$

$$\tilde{\alpha}^2\tau_{t-1} = \tilde{\alpha}^{t+1}(1+(t-1)\beta_0)\tau_0 + 2\sum_{k=1}^{t-1}(t-k)\tilde{\alpha}^{t-k+1}\beta_k\tau_k - 2\sum_{k=1}^{t-1}\tilde{\alpha}^{t-k+1}\beta_k\tau_k$$

$$\tilde{\alpha}\tau_t - \tilde{\alpha}^2\tau_{t-1} = \tilde{\alpha}^{t+1}\beta_0\tau_0 + 2\sum_{k=1}^{t-1}\tilde{\alpha}^{t-k+1}\beta_k\tau_k.$$

$\tau_{t+1} - 2\tilde{\alpha}\tau_t + \tilde{\alpha}^2\tau_{t-1} = 2\tilde{\alpha}\beta_t\tau_t$

The above analysis finally leads to $\tau_{t+1} - 2(1+\beta_t)\tilde{\alpha}\tau_t + \tilde{\alpha}^2\tau_{t-1} = 0$. So for

$$\tilde{\alpha}^t\prod_{j=0}^{t-1}(1+\beta_j)y_t = \tau_t$$

we have

$$y_{t+1} - 2y_t + \frac{y_t}{(1+\beta_t)(1+\beta_{t-1})} = 0.$$

Following the same strategy in Corollary C.9, we obtain the upper bound. $\qquad\square$

**Theorem D.8** (Convergence of LOCHB). *Let the geometric mean of $\beta_t$ be $\overline{\beta}_t \triangleq \prod_{j=0}^{t-1}(1+\beta_j)^{1/t}$. Then the upper bound of $\|r^{(t)}\|_2$ can be characterized as*

$$\|r^{(t)}\|_2 \leq \tilde{\alpha}^t\prod_{j=0}^{t-1}(1+\beta_j)y_t, \tag{37}$$

*where $y_{t+1} - 2y_t + y_{t-1}/((1+\beta_{t-1})(1+\beta_t)) = 0$ where $y_0 = y_1 = \|r^{(0)}\|_2$. Assume that there exists a constant $c \in [0,2)$ such that $\beta_t \leq 1 + \frac{c\sqrt{\alpha}}{1-\sqrt{\alpha}}$. Then the total number of iterations can be bounded as*

$$T \leq \left\lceil \frac{1+\sqrt{\alpha}}{(2-c)\sqrt{\alpha}} \right\rceil \ln\left(\frac{y_t}{\epsilon}\right).$$

*Then the total runtime $\mathcal{T}$ is bounded by*

$$\mathcal{T} \leq \Theta\left(\frac{(1+\sqrt{\alpha})\overline{\mathrm{vol}}(\mathcal{S}_T)}{(2-c)\sqrt{\alpha}}\ln\frac{y_t}{\epsilon}\right) = \widetilde{\mathcal{O}}\left(\frac{\overline{\mathrm{vol}}(\mathcal{S}_T)}{(2-c)\sqrt{\alpha}}\right),$$

*where $\widetilde{\mathcal{O}}$ hides $\ln(y_t/\epsilon)$.*

*Proof.* The first part follows from Corollary of D.7. The rest follows the same strategy of LOCCH as in Theorem 4.2. (See also Theorem 4.2.)

$\qquad\square$

Table 2: Examples of sparse linear systems

| Original Linear system | Our target $Qx = b$ $Q = \Lambda - \sigma D^{-1/2}AD^{-1/2}$ | $[\mu, L]$ | Ref. |
|---|---|---|---|
| $(I - \sigma W)x = \alpha e_s$ [1.] | $\Lambda = I, \sigma = 1 - \alpha$ $b = \alpha e_s$ | $[\alpha, 2 - \alpha]$ | [27] [53] |
| $\left(\alpha I + \frac{1-\alpha}{2}\mathcal{L}\right)x = \alpha D^{-1/2}e_s$ [2.] $\mathcal{L} = I - D^{-1/2}AD^{-1/2}$ | $\Lambda = I, \sigma = \frac{1-\alpha}{1+\alpha}$ $b = 2\alpha D^{-1/2}s/(1+\alpha)$ | $\left[\frac{2\alpha}{1+\alpha}, \frac{2}{1+\alpha}\right]$ | [2] [13] [12] [37] |
| $\left(I - (1-\alpha)AD^{-1}\right)y = \alpha e_s$ [3.] | $\Lambda = I, \sigma = 1 - \alpha$ $b = \alpha D^{-1/2}e_s, \alpha \in (0,1)$ $x = D^{-1/2}y$ | $[\alpha, 2 - \alpha]$ | [7] [56] |
| $\left(\frac{\lambda}{n}I_n + D - A\right)y = 2\lambda e_s$ [4.] | $\Lambda = \frac{\lambda}{n}D^{-1} + I_n, \sigma = 1$ $b = 2\lambda D^{-1/2}e_s, \lambda \in [1, n]$ $x = D^{1/2}y$ | $\left[\frac{\lambda}{nd_{\max}}, \frac{\lambda+2n}{n}\right]$ | [41] [57] |

## D.4 Implementation of LOCHB

We present the implementation of LOCHB as follows: Recall the updates of LOCHB is

$$x^{(t+1)} = x^{(t)} + (1 + \tilde{\alpha}^2)r_{\mathcal{S}_t}^{(t)} + \tilde{\alpha}^2\left(x^{(t)} - x^{(t-1)}\right)_{\mathcal{S}_t}, \quad r^{(t+1)} = 2\tilde{\alpha}Wr^{(t)} - \tilde{\alpha}^2 r^{(t-1)}.$$

The corresponding local updates are

$$\mathbf{\Delta}^{(t)} = (1 + \tilde{\alpha}^2)r^{(t)} + \tilde{\alpha}^2\tilde{x}^{(t)}$$

$$x^{(t+1)} = x^{(t)} + \mathbf{\Delta}_{\mathcal{S}_t}^{(t)}$$

$$r^{(t+1)} = r^{(t)} - \mathbf{\Delta}^{(t)} + \frac{1-\alpha}{1+\alpha}W\mathbf{\Delta}^{(t)}$$

$$\tilde{x}^{(t+1)} = \tilde{x}^{(t)} + \mathbf{\Delta}^{(t)} - \mathbf{\Delta}^{(t-1)},$$

where if we choose $x^{(0)} = x^{(1)} = \tilde{x}^{(1)} = 0$, $r^{(0)} = r^{(1)} = b$ and $\mathbf{\Delta}^{(0)} = 0$.

# E Instances of Sparse Linear Systems

## E.1 Table of Popular Graph-induced Linear Systems

This section presents most commonly used graph-induced linear system as the following

$$\underbrace{\Lambda - \sigma D^{-1/2}AD^{-1/2}}_{Q}x = b,$$

where $Q$ is the generalized version of the perturbed normalized graph Laplacian matrix with perturbation parameter $\sigma > 0$, and $b$ is a sparse vector. A typical example of $Q = I - \frac{1-\alpha}{1+\alpha}D^{-1/2}AD^{-1/2}$ with $\Lambda = I$ and $b = 2\alpha D^{-1/2}e_s/(1+\alpha)$

The detailed parameters are:

- 1. $W = \tilde{D}^{-1/2}\tilde{A}\tilde{D}^{-1/2}$ and $\tilde{A} = I + A$ is the adjacency matrix defined on $\mathcal{G}(\mathcal{V}, \mathcal{E})$ by adding self-loops for all nodes, and $\tilde{D} = I + D$ is defined as the augmented degree matrix by adding self-loops. $\alpha \in (0,1)$ and usually $\alpha < 0.5$, The $I - (1 - \alpha)W$ is the perturbed augmented normalized Laplacian with perturbed parameter $\alpha$.

- 2. $Q = D^{-1/2}\left(D - \frac{1-\alpha}{2}(D + A)\right)D^{-1/2} = \alpha I + \frac{1-\alpha}{2}\mathcal{L} > 0$ and $Q = \frac{1-\alpha}{1+\alpha}D^{-1/2}AD^{-1/2}$. This is known as the lazy random-walk version of PPR vectors.

- 3. $x = \alpha\left(I - (1 - \alpha)AD^{-1}\right)^{-1}e_s$ This is the standard Personalized PageRank vectors widely used for graph embeddings and graph neural network designing [7]. It is also used for decoupling for large-scale GNNs [56].

| Dataset ID | Dataset Name | n | m |
|---|---|---|---|
| $\mathcal{G}_1$ | as-skitter | 1,694,616 | 11,094,209 |
| $\mathcal{G}_2$ | cit-patent | 3,764,117 | 16,511,740 |
| $\mathcal{G}_3$ | com-dblp | 317,080 | 1,049,866 |
| $\mathcal{G}_4$ | com-lj | 3,997,962 | 34,681,189 |
| $\mathcal{G}_5$ | com-orkut | 3,072,441 | 117,185,083 |
| $\mathcal{G}_6$ | com-youtube | 1,134,890 | 2,987,624 |
| $\mathcal{G}_7$ | ogbn-arxiv | 169,343 | 1,157,799 |
| $\mathcal{G}_8$ | ogbn-mag | 1,939,743 | 21,091,072 |
| $\mathcal{G}_9$ | ogbn-products | 2,385,902 | 61,806,303 |
| $\mathcal{G}_{10}$ | ogbn-proteins | 132,534 | 39,561,252 |
| $\mathcal{G}_{11}$ | soc-lj1 | 4,843,953 | 42,845,684 |
| $\mathcal{G}_{12}$ | soc-pokec | 1,632,803 | 22,301,964 |
| $\mathcal{G}_{13}$ | wiki-talk | 2,388,953 | 4,656,682 |
| $\mathcal{G}_{14}$ | ogbl-ppa | 576,039 | 21,231,776 |
| $\mathcal{G}_{15}$ | wiki-en21 | 6,216,199 | 160,823,797 |
| $\mathcal{G}_{16}$ | com-friendster | 65,608,366 | 1,806,067,135 |
| $\mathcal{G}_{17}$ | ogbn-papers100m | 111,059,433 | 1,614,061,934 |

Table 3: Dataset Statistics

- 4. Graph kernel computation for online learning. Each computed vector serves as semi-supervised learning feature vectors [24] or as online node labeling learning vectors [41, 57]. Note the target linear system when $\sigma = 1$

$$\boldsymbol{D}^{-1/2}\left(\frac{\lambda}{n}\boldsymbol{I}_n + \boldsymbol{D} - \boldsymbol{A}\right)\boldsymbol{D}^{-1/2}\boldsymbol{D}^{1/2}\boldsymbol{y} = 2\lambda\boldsymbol{D}^{-1/2}\boldsymbol{e}_s$$

$$\left(\frac{\lambda}{n}\boldsymbol{D}^{-1} + \boldsymbol{I}_n - \boldsymbol{D}^{-1/2}\boldsymbol{A}\boldsymbol{D}^{-1/2}\right)\boldsymbol{D}^{1/2}\boldsymbol{y} = 2\lambda\boldsymbol{D}^{-1/2}\boldsymbol{e}_s.$$

Hence, we have $\boldsymbol{\Lambda} = \frac{\lambda}{n}\boldsymbol{D}^{-1} + \boldsymbol{I}_n, \sigma = 1, \boldsymbol{b} = 2\lambda\boldsymbol{D}^{-1/2}\boldsymbol{e}_s$.

# F   Experimental Details and Missing Results

## F.1   Datasets and Preprocessing

Following Leskovec et al. [34], we treat all 17 graphs as undirected with unit weights. We remove self-loops and keep the largest connected component when the graph is disconnected. After preprocessing, the graphs range from $169,343$ nodes in ogbn-proteins to $111,059,433$ in ogbn-papers100M, as presented in Table 3.

## F.2   Problems Settings and Baseline Methods

For solving Equation (3), we randomly select 50 source nodes $s$ from each graph. The damping factor is fixed at 0.1, i.e., $\alpha = 0.1$ for all experiments, and it varies within the range $\{0.005, 0.01, 0.05, 0.1, 0.15, 0.2, 0.25, 0.3\}$ for others. The $\epsilon$ is chosen from the range $\left[2\alpha/((1+\alpha)d_s), 10^{-4}/n\right]$.

For solving the local clustering problem, we follow the greedy strategy from Andersen et al. [2], where a local cluster is identified by examining the top magnitudes in PPR vectors. Specifically, we denote the boundary of $\mathcal{S}$ as $\partial(\mathcal{S}) = \{(u,v) \in \mathcal{E} : u \in \mathcal{S}, v \notin \mathcal{S}\}$. The conductance of $\mathcal{S}$ is defined as

$$\Phi(\mathcal{S}) \triangleq \frac{|\partial(\mathcal{S})|}{\min(\mathrm{vol}(\mathcal{S}), 2m - \mathrm{vol}(\mathcal{V}\setminus\mathcal{S}))}.$$

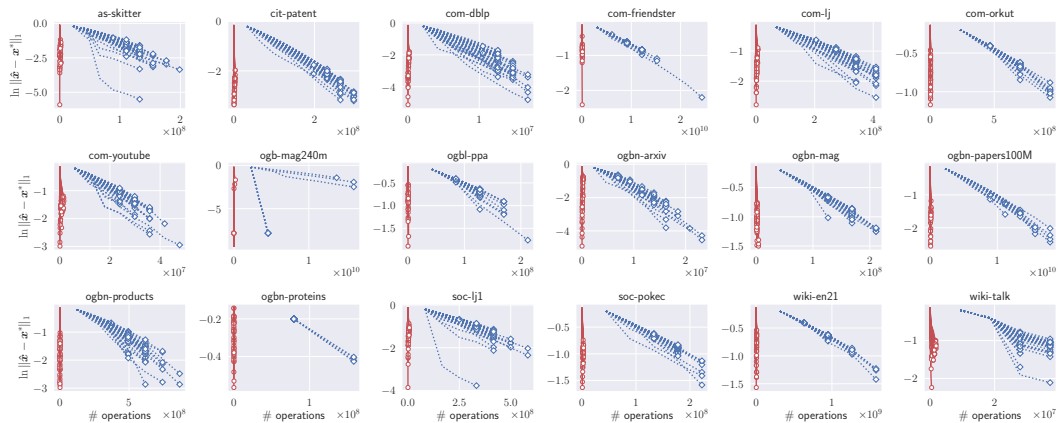

Figure 7: The LOCSOR method compared with CGM over 18 graphs.

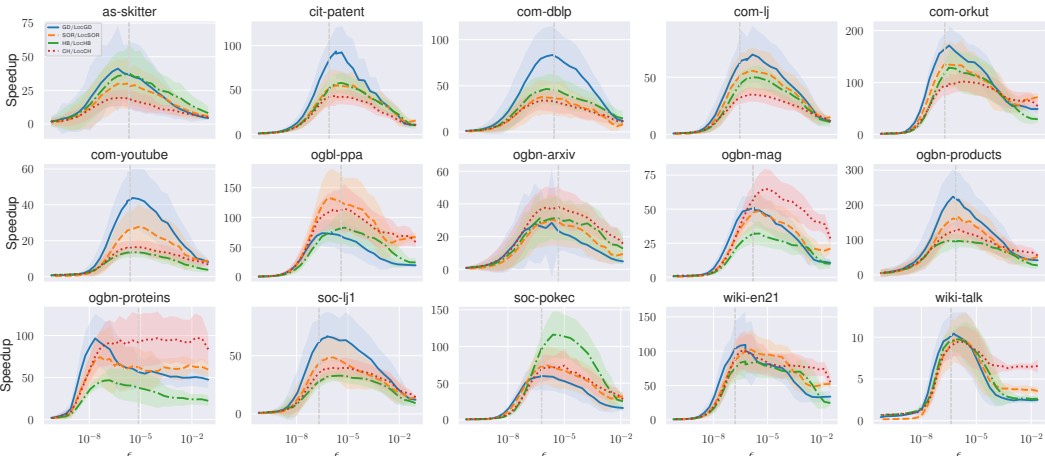

Figure 8: The speedup of local solvers compared with their standard counterparts.

The goal of local clustering is to obtain PPR vectors using these local methods and then apply clustering algorithms to find clusters with low conductance. For the sorting process, given the approximate PPR vector $\tilde{\pi}$, we sort $D^{-1/2}\tilde{\pi}$ in decreasing order of magnitudes. Let the ordered nodes be $v_1, v_2, \ldots, v_t$; the local clustering algorithm iteratively checks the conductance reduction by $v_1, v_2, \cdots, v_k$ where $k = 1, 2, \ldots, t$, and after completing all checks, it returns a subset $v_1, v_2, \ldots, v_{k'}$ that has the minimal conductance among all examined subsets.

**Parameter settings of baselines.** For the local ISTA method [13], the precision parameter is set to $\hat{\epsilon} = 0.5$ for all experiments. According to the algorithm's description of ISTA, the corresponding $\rho$ value is given by $\epsilon/(1 + \hat{\epsilon})$. For LOCSOR, the parameter $\omega$ is calculated as $2(1 + \alpha)/(1 + \sqrt{\alpha})^2$. For the local FISTA, as demonstrated in [22], we adopt the same settings as for ISTA and follow its implementation guidelines. We also include preliminary results on ASPR [37]. The algorithm incorporates a parameter, $\hat{\epsilon}$, to control the number of iterations in the nested Accelerated Projected Gradient Descent (APGD). We adjust $\hat{\epsilon}$ from low precision, $\hat{\epsilon} = 0.1/n$, to high precision, $\hat{\epsilon} = 10^{-4}/n$, to ensure the identification of a good approximation.

For our experiment, we used a server powered by an Intel(R) Xeon(R) Gold 5218R CPU, which features 40 cores (80 threads). The system is equipped with 256 GB of RAM.

## F.3 Full results of Fig. 15 4 5 6

In all 15 graphs, we set $\alpha = 0.1$ and $\epsilon = 0.1/n$. For each of the testing graphs, we randomly select 50 nodes and run LOCSOR and CGM.

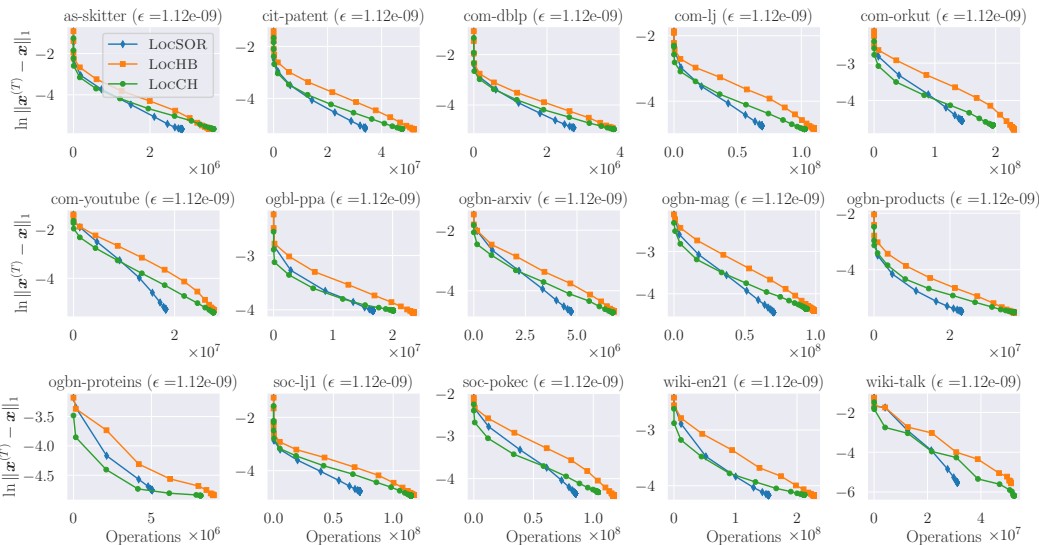

Figure 10: Comparison of three local solvers over 15 graphs.

Fig. 8 presents all speedup tests on 15 datasets. It is evident that these standard linear solvers can be localized effectively.

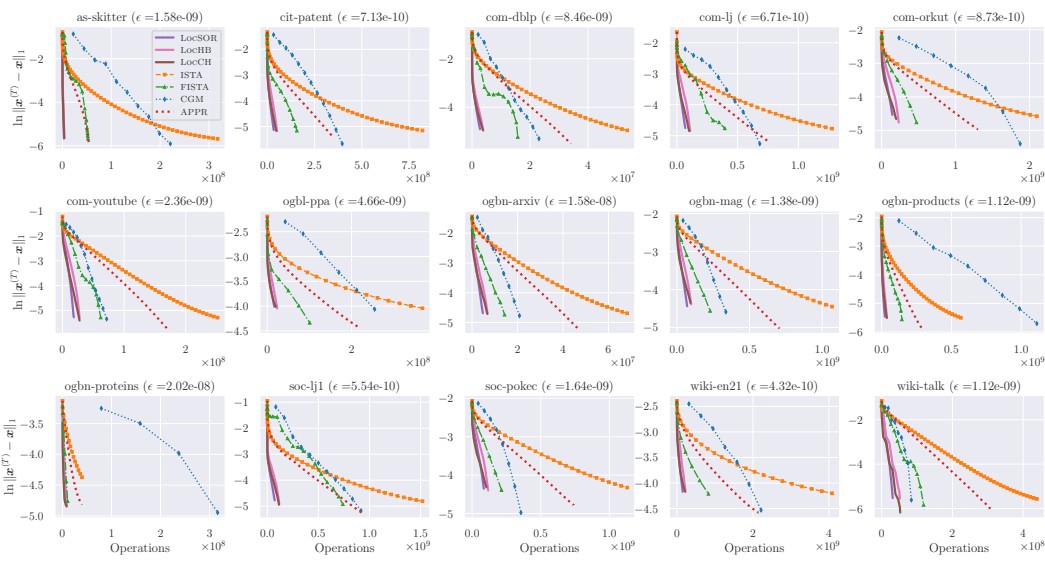

Figure 9: The estimation error reduction tests on 7 solvers including our LOCSOR, LOCHB, and LOCCH. The experiments were conducted on 15 datasets.

Fig. 9 presents the missing results on the estimation error reduction for 15 datasets. Compared with the global solver CGM, all local methods show significant speedup in the early stages. To compare our three local solvers, we zoom in on our results and present them in Fig. 10. Empirically, LOCSOR is the fastest algorithm when the parameter $\omega$ is chosen optimally.

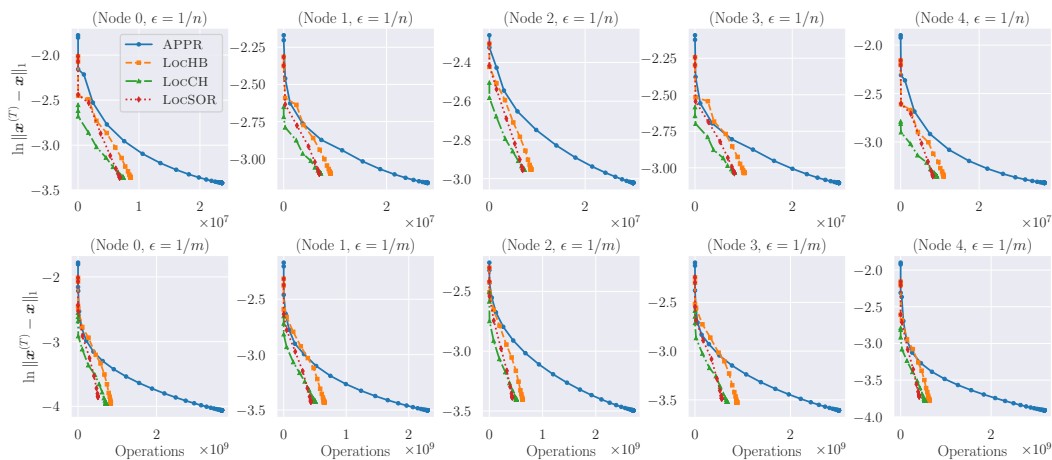

Figure 11: Estimation error as a function of the number of operations on com-friendster. We randomly select 5 different nodes and use $\epsilon = 1./n$ and $\epsilon = 1/m$.

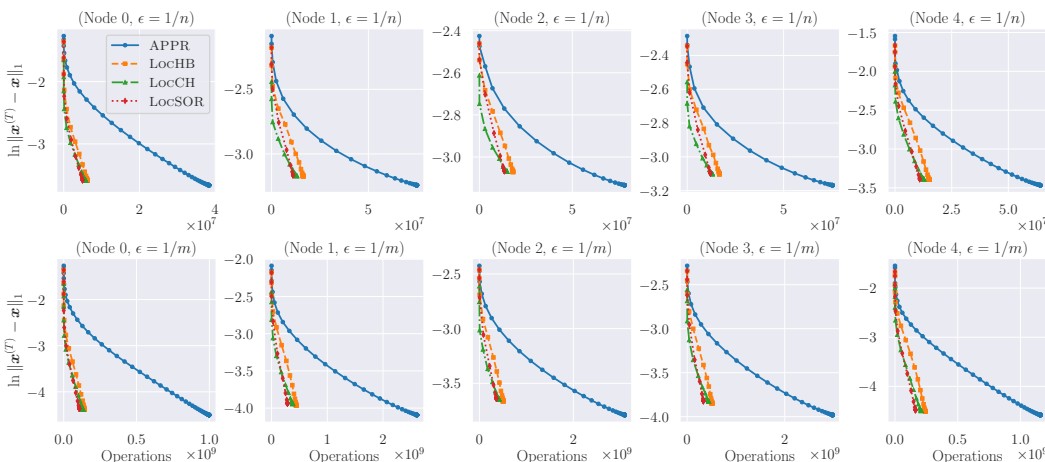

Figure 12: Estimation error as a function of the number of operations on ogbn-papers100m. We randomly select 5 different nodes and use $\epsilon = 1./n$ and $\epsilon = 1/m$.

| - | Run time (seconds) | | | Number of operations | | |
|---|---|---|---|---|---|---|
| Dataset | LocSOR | LocCH | CGM | LocSOR | LocCH | CGM |
| as-skitter | $0.350 \pm 0.054$ | $0.567 \pm 0.095$ | $2.626 \pm 0.551$ | 7.827e+05 | 1.026e+06 | 1.524e+08 |
| cit-patent | $0.966 \pm 0.120$ | $1.609 \pm 0.189$ | $14.660 \pm 2.249$ | 1.804e+06 | 2.298e+06 | 2.873e+08 |
| com-dblp | $0.068 \pm 0.058$ | $0.104 \pm 0.059$ | $0.469 \pm 0.112$ | 8.222e+04 | 1.166e+05 | 1.562e+07 |
| com-friendster | $15.29 \pm 1.89$ | $26.54 \pm 3.52$ | $508.50 \pm 99.12$ | 7.027e+07 | 8.063e+07 | 1.442e+10 |
| com-lj | $0.802 \pm 0.148$ | $1.410 \pm 0.234$ | $7.593 \pm 2.361$ | 2.604e+06 | 3.271e+06 | 4.122e+08 |
| com-orkut | $0.455 \pm 0.158$ | $0.815 \pm 0.308$ | $13.343 \pm 7.951$ | 2.965e+06 | 3.221e+06 | 9.220e+08 |
| com-youtube | $0.290 \pm 0.070$ | $0.501 \pm 0.099$ | $1.314 \pm 0.257$ | 7.617e+05 | 9.323e+05 | 3.561e+07 |
| ogb-mag240m | $85.14 \pm 16.05$ | $108.86 \pm 7.994$ | $549.51 \pm 367.2$ | 4.541e+07 | 5.820e+07 | 9.426e+09 |
| ogbl-ppa | $0.116 \pm 0.019$ | $0.202 \pm 0.040$ | $5.624 \pm 1.108$ | 6.117e+05 | 6.445e+05 | 1.682e+08 |
| ogbn-arxiv | $0.039 \pm 0.060$ | $0.058 \pm 0.063$ | $0.239 \pm 0.104$ | 1.158e+05 | 1.354e+05 | 1.473e+07 |
| ogbn-mag | $0.520 \pm 0.079$ | $0.921 \pm 0.128$ | $6.136 \pm 0.974$ | 1.804e+06 | 1.947e+06 | 2.050e+08 |
| ogbn-papers100M | $27.20 \pm 3.94$ | $41.99 \pm 6.17$ | $750.96 \pm 110.45$ | 8.893e+07 | 1.095e+08 | 1.604e+10 |
| ogbn-products | $0.695 \pm 0.236$ | $1.059 \pm 0.249$ | $31.907 \pm 4.961$ | 1.777e+06 | 2.385e+06 | 7.071e+08 |
| ogbn-proteins | $0.021 \pm 0.057$ | $0.025 \pm 0.057$ | $0.910 \pm 0.306$ | 7.941e+04 | 6.610e+04 | 1.488e+08 |
| soc-lj1 | $1.040 \pm 0.282$ | $1.751 \pm 0.487$ | $7.102 \pm 3.410$ | 3.263e+06 | 4.045e+06 | 4.833e+08 |
| soc-pokec | $0.210 \pm 0.027$ | $0.368 \pm 0.049$ | $1.568 \pm 0.207$ | 2.020e+06 | 2.225e+06 | 2.160e+08 |
| wiki-en21 | $1.436 \pm 2.708$ | $1.794 \pm 0.215$ | $19.658 \pm 4.576$ | 5.996e+06 | 6.659e+06 | 1.329e+09 |
| wiki-talk | $0.251 \pm 0.049$ | $0.455 \pm 0.091$ | $0.642 \pm 0.071$ | 1.090e+06 | 1.290e+06 | 4.284e+07 |

Table 4: Summary of runtime and operations for 15 datasets. ($\epsilon = 10^{-6}$)

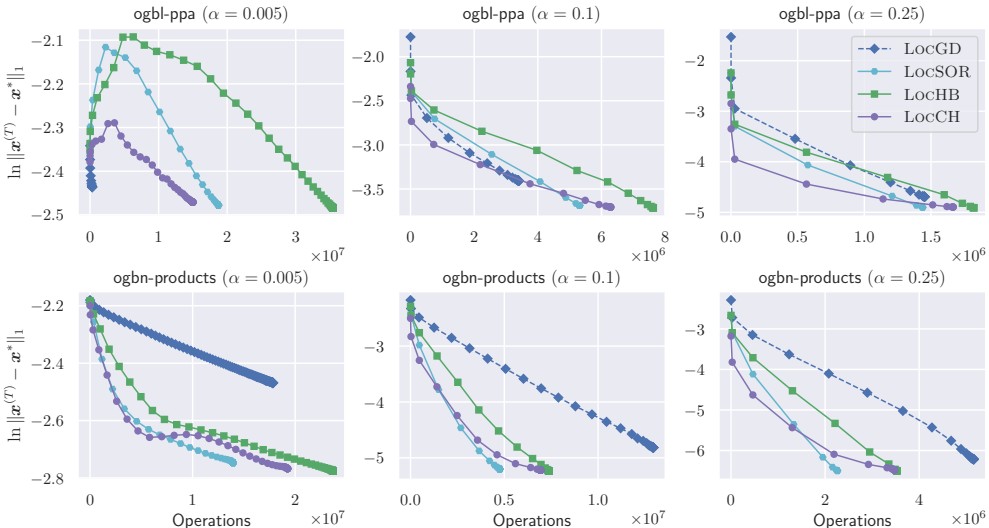

Figure 13: Estimation error as a function of operations needed. For $\alpha = 0.005$, $\alpha = 0.1$, and $\alpha = 0.25$.

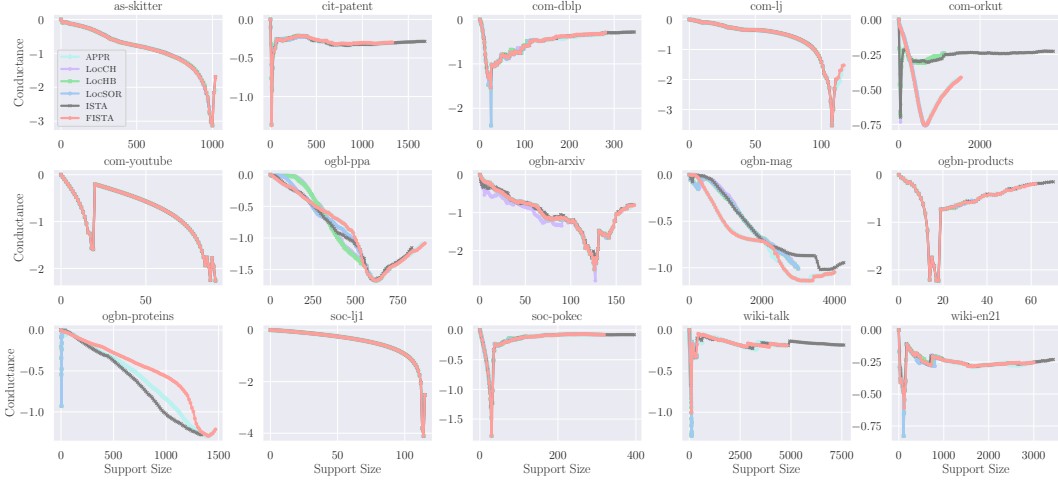

Figure 14: The graph conductance found by local graph clustering method using different local approximate methods. Experiments ran on 15 graphs. ($\epsilon = 10^{-6}$)

| Dataset | APPR | LocCH | LocHB | LocSOR | ISTA | FISTA |
|---------|------|-------|-------|--------|------|-------|
| as-skitter | 5.90e-04 | 5.90e-04 | 5.90e-04 | 5.90e-04 | 5.90e-04 | 5.90e-04 |
| cit-patent | 4.23e-02 | 4.23e-02 | 4.23e-02 | 4.23e-02 | 4.23e-02 | 4.23e-02 |
| com-dblp | 4.12e-03 | 4.12e-03 | 4.12e-03 | 4.12e-03 | 4.12e-03 | 4.12e-03 |
| com-lj | 2.94e-04 | 2.94e-04 | 2.94e-04 | 2.94e-04 | 2.94e-04 | 2.94e-04 |
| com-orkut | 1.75e-01 | 1.76e-01 | 1.76e-01 | 1.75e-01 | 1.76e-01 | 1.75e-01 |
| com-youtube | 5.46e-03 | 5.46e-03 | 5.46e-03 | 5.46e-03 | 5.46e-03 | 5.46e-03 |
| ogbn-arxiv | 2.11e-02 | 2.11e-02 | 2.11e-02 | 2.12e-02 | 2.14e-02 | 2.11e-02 |
| ogbn-mag | 3.18e-03 | 1.59e-03 | 3.18e-03 | 3.18e-03 | 4.74e-03 | 3.18e-03 |
| ogbn-products | 7.26e-02 | 1.02e-01 | 9.68e-02 | 9.65e-02 | 9.51e-02 | 7.24e-02 |
| ogbn-proteins | 5.92e-03 | 5.92e-03 | 5.92e-03 | 5.92e-03 | 5.92e-03 | 5.92e-03 |
| soc-lj1 | 5.07e-02 | 1.18e-01 | 1.18e-01 | 1.18e-01 | 5.25e-02 | 5.07e-02 |
| soc-pokec | 7.80e-05 | 7.80e-05 | 7.80e-05 | 7.80e-05 | 7.80e-05 | 7.80e-05 |
| wiki-talk | 1.64e-02 | 1.64e-02 | 1.64e-02 | 1.64e-02 | 1.64e-02 | 1.64e-02 |
| ogbl-ppa | 5.13e-02 | 5.13e-02 | 5.13e-02 | 5.13e-02 | 5.13e-02 | 5.13e-02 |
| wiki-en21 | 1.31e-01 | 1.31e-01 | 1.31e-01 | 1.31e-01 | 1.31e-01 | 1.31e-01 |

Table 5: The local conductance for six local solvers tested on 15 graphs datasets. ($\epsilon = 10^{-6}$)

| Dataset | APPR | LocCH | LocHB | LocSOR | ISTA | FISTA |
|---------|------|-------|-------|--------|------|-------|
| as-skitter | 0.127 | 0.147 | 0.144 | 0.043 | 0.323 | 0.093 |
| cit-patent | 0.362 | 0.516 | 0.457 | 0.125 | 0.939 | 0.308 |
| com-dblp | 0.069 | 0.033 | 0.028 | 0.014 | 0.297 | 0.042 |
| com-lj | 0.357 | 0.440 | 0.664 | 0.175 | 0.493 | 0.229 |
| com-orkut | 0.072 | 0.141 | 0.139 | 0.055 | 0.108 | 0.084 |
| com-youtube | 0.128 | 0.176 | 0.131 | 0.040 | 0.682 | 0.102 |
| ogbl-ppa | 0.102 | 0.047 | 0.091 | 0.027 | 0.146 | 0.102 |
| ogbn-arxiv | 0.042 | 0.013 | 0.014 | 0.006 | 0.237 | 0.032 |
| ogbn-mag | 0.068 | 0.137 | 0.091 | 0.035 | 0.253 | 0.090 |
| ogbn-products | 0.072 | 0.121 | 0.117 | 0.045 | 0.135 | 0.090 |
| ogbn-proteins | 0.005 | 0.005 | 0.005 | 0.002 | 0.005 | 0.004 |
| soc-lj1 | 0.239 | 0.376 | 0.350 | 0.103 | 0.512 | 0.194 |
| soc-pokec | 0.067 | 0.096 | 0.059 | 0.033 | 0.222 | 0.051 |
| wiki-talk | 0.197 | 0.314 | 0.274 | 0.109 | 0.508 | 0.176 |
| wiki-en21 | 0.121 | 0.290 | 0.279 | 0.106 | 0.197 | 0.147 |

Table 6: Runtime (seconds) for six local solvers tested on 15 graphs datasets. ($\epsilon = 10^{-6}$)

## F.4 Results on local clustering

| Dataset | APPR | LOCCH | LOCHB | LOCSOR | ISTA | FISTA |
|---|---|---|---|---|---|---|
| as-skitter | 6.9e+05 | 7.6e+04 | 8.1e+04 | 6.5e+04 | 2.9e+06 | 5.7e+05 |
| cit-patent | 6.7e+05 | 1.0e+05 | 1.1e+05 | 8.9e+04 | 2.3e+06 | 4.4e+05 |
| com-dblp | 4.3e+05 | 4.8e+04 | 5.0e+04 | 3.5e+04 | 1.9e+06 | 2.9e+05 |
| com-lj | 5.7e+05 | 8.9e+04 | 1.0e+05 | 7.6e+04 | 2.8e+06 | 4.4e+05 |
| com-orkut | 5.4e+05 | 8.9e+04 | 1.1e+05 | 9.0e+04 | 1.3e+06 | 5.0e+05 |
| com-youtube | 5.3e+05 | 6.5e+04 | 6.6e+04 | 5.7e+04 | 3.8e+06 | 4.4e+05 |
| ogbl-ppa | 6.7e+05 | 9.6e+04 | 1.1e+05 | 9.3e+04 | 1.5e+06 | 6.0e+05 |
| ogbn-arxiv | 6.8e+05 | 8.7e+04 | 9.7e+04 | 7.6e+04 | 2.7e+06 | 5.2e+05 |
| ogbn-mag | 5.2e+05 | 8.2e+04 | 9.4e+04 | 8.3e+04 | 2.4e+06 | 4.4e+05 |
| ogbn-products | 9.0e+05 | 1.1e+05 | 1.3e+05 | 9.8e+04 | 2.1e+06 | 7.6e+05 |
| ogbn-proteins | 5.3e+05 | 4.0e+04 | 5.5e+04 | 5.0e+04 | 6.8e+05 | 6.2e+05 |
| soc-lj1 | 5.6e+05 | 8.9e+04 | 1.0e+05 | 7.7e+04 | 2.8e+06 | 4.4e+05 |
| soc-pokec | 5.9e+05 | 1.1e+05 | 1.3e+05 | 1.1e+05 | 2.5e+06 | 5.0e+05 |
| wiki-talk | 4.4e+05 | 5.5e+04 | 5.2e+04 | 4.6e+04 | 1.8e+06 | 3.4e+05 |
| wiki-en21 | 5.2e+05 | 8.0e+04 | 9.7e+04 | 8.1e+04 | 1.8e+06 | 4.9e+05 |

Table 7: Operations Needed for six local solvers tested on 15 graphs datasets. ($\epsilon = 10^{-6}$)

# G   Related work

Many graph applications [2, 7, 29, 15, 27, 36, 30, 44, 25, 45, 51, 48] only require solving Equ. (1) approximately. The reasons could be either the most energies of $\pi$ are among a small set of nodes forming small subgraphs, or one wants to study large graphs by checking them locally. Given a graph $\mathcal{G}$ with $n$ nodes and $m$ edges, there are two main types of iterative solvers for Equ. (1) as follows:

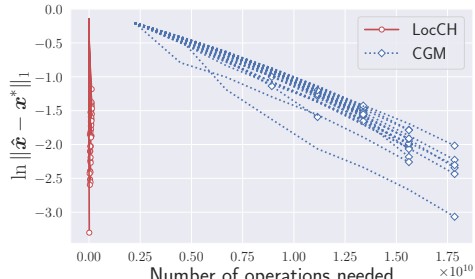

Figure 15: Comparison of the error reduction between the proposed LOCCH and the standard CGM on the papers100M dataset [23], in terms of the number of operations required.

**Standard iterative methods.** Methods for solving linear systems have been well-established over the past decades (see textbooks of Saad [43], Golub & Van Loan [19], Young [55]). The fastest linear solver for solving the symmetrized version of Equ. (1) is the Conjugate Gradient Method (CGM) with runtime complexity $\tilde{\mathcal{O}}(m/\sqrt{\alpha})$ where $m$ is the number of edges in the graph. It costs $\Theta(m)$ to access the entire graph at each iteration; hence, it is much slower than local solvers, as demonstrated in Fig. 15. The symmetric diagonally dominant (SDD) solvers advance CGM further to have complexity $\tilde{\mathcal{O}}\left(m \log^c n \log(1/\epsilon)\right)$ [31, 46]. Anikin et al. [4] considered the PageRank problem and proposed an algorithm with runtime depending on $\mathcal{O}(n)$. This paper focuses on local algorithms where the goal is to avoid the dominant factor $m$ or $n$ by avoiding the full $\boldsymbol{Q}\boldsymbol{x}$ operation.

**Local algorithms.** Local solvers, in contrast to standard counterparts, leverage the fact that the energy of $\pi$ lives in a small portion of the graph and hence do not require $\mathcal{O}(m)$ or $\mathcal{O}(n)$ per iteration. They are advantageous for huge-scale graphs demonstrated in Fig. 15. Andersen et al. [2] used APPR to obtain an approximate of $\pi$ for local clustering. Quite similar algorithms were developed in Berkhin [6] (*bookmark-coloring* algorithm) and Kloster & Gleich [28] (Gauss-Southwell procedure).

Under the same stopping condition as APPR, Fountoulakis et al. [13] demonstrated that APPR is equivalent to coordinate descent via variational characterization, with a runtime of $\tilde{\mathcal{O}}(1/(\alpha\epsilon))$ using ISTA where the monotonicity and conservation properties remain. Hence, it is nature to ask whether $\tilde{\mathcal{O}}(1/(\sqrt{\alpha}\epsilon))$ could be achieved by FISTA [5] in Fountoulakis & Yang [12]. However, the difficulty is that FISTA violates the monotonicity property where the volume accessed of per-iteration cannot be bounded properly. To overcome this, Martínez-Rubio et al. [37] proposed a nested accelerated projected gradient descent (APGD) and gradually expanding solutions so that the monotonicity property still holds. However, nested APGD requires solving subproblems accurately, which in practice may be cumbersome if the precision requirement of the inner problem is too stringent. All current local methods rely on some monotonicity property of variables to guarantee locality, which does not exist in most accelerated frameworks; thus, developing an accelerated method that is guaranteed to preserve intermediate variable sparsity remains challenging.

It is worth mentioning that local methods are also closely related to sublinear time and local computational algorithms [42, 1]. From the optimization perspective, the equivalence between Gauss-Seidel and coordinate descent has been considered [49, 35, 39, 50] but does not focus on local analysis.

