# OpenReview forum: "Iterative Methods via Locally Evolving Set Process"
_NeurIPS.cc/2024/Conference — NeurIPS 2024 poster_

### Official Review · Reviewer_N8Gt · 2024-06-24

**Soundness:** 3
**Presentation:** 3
**Contribution:** 3
**Rating:** 6
**Confidence:** 3

**Summary:**

This paper considers the study of local algorithms for graph clustering which is an important problem in the field of graph data analysis. In particular this paper is considers the task of computing Personalized Page Rank (PPR) vectors for a given graph. In this problem the algorithm is given a graph in the form of its adjacency and degree matrices, the goal is to approximate the Personalized Page Rank vector for a given starting vertex and dampening factor $\alpha$ up to precision $\epsilon$ without accessing the entire graph. The classical algorithm of Andersen, Chung and Lang runs in time $O(1/\alpha \epsilon)$, which independent of the graph size. The central question posed by subsequent works is whether the dependence on $\alpha$ can be improved to $1/\sqrt{\alpha}$. The main contribution of the paper is to propose a new algorithmic framework based on the locally evolving set process. Under this framework they are able to implement existing algorithms such as Andersen et al.'s APPR algorithm as well as localized implementation of standard gradient descent. They are also able to develop localized versions of chebyshev and heavy ball methods that do achieve the $1/\sqrt{\alpha}$ dependence for some fixed constant value of $\epsilon$. Finally they show that on several large scale graphs, their new localized chebyshev and heavy ball methods do outperform APPR and related methods empirically.

**Strengths:**

The main strengths of this paper are to develop a new algorithmic framework that can not only encompass existing algorithms but lead to the development of better ones that overcome previously known limitations for designing local graph clustering algorithms. They also back their theoretical analysis with the practical implementation of their method which is also shown to be superior to previous algorithms.

**Weaknesses:**

One weakness is that the paper is only able to obtain a quadratic improvement in the dependence on the parameter $\alpha$, obtained by the local implementation of the Chebyshev and Heavy-ball method, only for a value of $\epsilon$ and not for all.

**Questions:**

One question is that what is the core reason for not obtaining convergence result for accelerated methods for all $\epsilon>0$.

**Limitations:**

Authors have addressed limitations.

---

> ### Author Rebuttal · Authors · 2024-08-06
>
> Thank you for taking the time and effort to review our paper carefully. We appreciate the positive perspective on our work. Your concern on the range of $\epsilon$ can be effectively addressed as follows:
>
> **Q1.** What is the core reason for not obtaining convergence results for accelerated methods for all $\epsilon > 0$?
>
> **A:** To clarify, our results cover all cases for   $0 \leq \epsilon \leq 1/d_s$, including Theorems 3.3, 3.5, and 4.2. When $\epsilon > 1/d_s$, the algorithm terminates before taking a single step, e.g. $T = 0$. In other words, we assume that the precision parameter satisfies $\epsilon \leq 1/d_s$ in our theorems because $\epsilon \leq 1/d_s$ ensures all local solvers run at least one local iteration, so that $T \geq 1$. All theorems hold when $1/d_s < \epsilon$. We thank the reviewer for pointing out if this was unclear in the manuscript and will revise it.
>
> We are happy to have further discussions if needed!

---

> > ### Comment · Reviewer_N8Gt · 2024-08-12
> > **Response to rebuttal**
> >
> > Thanks to the authors for answering the question. My score remains the same.

---

### Official Review · Reviewer_oXbQ · 2024-07-05

**Soundness:** 4
**Presentation:** 4
**Contribution:** 3
**Rating:** 9
**Confidence:** 4

**Summary:**

This paper uses the evolving set procedure to give a local PageRank algorithm whose dependence on \alpha (the reset probability) is \sqrt{\alpha}.

It proposes accelerated local iterative methods with coefficients given by Chebyshev iteration. The convergence of this algorithm in both graph theoretic and general sparse linear systems settings are analyzed in detail. Discussions of the relations between this method and other local iterative algorithms are also given in detail.

The method was implemented and tested on a range of graphs, mostly coming from social networks. This includes two large ones with edges in the billions. On moderate ranges of \alpha (reset probability), the experiments show significant speedups (factor of about 3) and convergences (factor of 10) on most graphs.

**Strengths:**

Local algorithms are widely used in graph analytics. The question studied is natural, and has been proposed before.

The method is theoretically well-founded, and has significant technical depth.

The experiments are thorough and well documents, and clearly demonstrate the advantages of this method in multiple parameter regimes.

**Weaknesses:**

The gains only kick in at a relatively large number of steps: it's not clear to me that these are the parameter regimes in which local algorithms actually get used.

Ideally for the empirical works I'd also like to see comparisons of downstream tasks and effects on overall accuracies (e.g. F-1 score), but the paper itself has already covered a lot of ground.

**Questions:**

Is the dependence on \epsilon optimal? Aka. have methods with \sqrt{\eps} (or even \log(1 / \eps)) dependences been ruled out?

**Limitations:**

yes, limitations have been addressed, and are entirely theoretical w.r.t. some graph parameter regimes.

---

> ### Author Rebuttal · Authors · 2024-08-06
>
> Thank you for taking the time and effort to review our paper carefully. Your positive perspective on our work is so inspiring. We also believe our work is novel and some new interesting problems are worth to explore. Your main concerns and our responses are as follows:
>
> ---
>
> **Q1.** The gains only kick in at a relatively large number of steps: it's unclear that these are the parameter regimes in which local algorithms get used.
>
> **A:** Thank you for this excellent point. We assume you mean in what range of $\epsilon$ is useful in practice. Let's take typical examples to see when $\epsilon$ is enough/useful in practice. For the local clustering algorithm, in Fountoulakis's paper (see Fountoulakis' work in [1]), they used $\epsilon = 10^{-5}$ where the number of nodes in graphs is in the range $[2\times 10^6, 3\times 10^6]$. This is around $\epsilon \approx 1/n$ (roughly corresponding to the largest speedup as shown in Figure 4, dashed vertical lines are $\epsilon = 1/n$). For learning GNN models, PPRs are used to train the PPRGo model (see Bojchevski's work in [2]); the parameter $\epsilon$ used is $10^{-4}$ where the number of nodes in graphs is in the range $[1.87\times 10^4, 1.05 \times 10^7]$. This means $\epsilon \leq 1/n$ in all graphs. We found similar parameter settings in graph embedding and online graph node classification. The effective range of $\epsilon$ is $\epsilon \leq 10^{-4} / n$ where the speedup is significant, as presented in Fig. 4. This clearly indicates that our design demonstrates significant speedup, especially around $\epsilon=1 / n$.
>
> If you mean the number of local iterations needed, then it is true that the number of local iterations will be slightly larger than that of standard solvers. The key point is the combination of both the number of iterations and local volumes, and then the runtime of local solvers is much smaller than that of standard solvers. Whether there is an optimal tradeoff between a local number of iterations and local volumes is an interesting problem.
>
> ---
>
> **Q2.** Ideally, for the empirical works, I'd also like to see comparisons of downstream tasks and effects on overall accuracies (e.g., F-1 score), but the paper itself has already covered a lot of ground.
>
> **A:** Yes, the downstream tasks, such as training good GNN models that have decent F1 scores, are typical examples of the methods that our proposed methods could apply. Any promising results will make our paper even stronger.  However, due to space limits, we will consider these tasks in our future work. And we are looking forward to downstream applications.
>
> ---
>
> **Q3.** Is the dependence on $\epsilon$ or $\alpha$ optimal?
>
> **A:** Let us recall that the runtime bound is optimal for the standard Chebyshev method. Specifically, the *first-order optimal methods (such as Chebyshev) for linear system* have the runtime bound $\mathcal O (m/\sqrt{\alpha} \cdot \log 1/\epsilon)$. This runtime bound is optimal for finding a solution of a sparse linear system defined on a chain graph (see page Theorem 3.15 of Bubeck's work in [1] for more details). We will discuss both parameters $\alpha$ and $\epsilon$.
>
> ---
> **References**
>
> - [1] Fountoulakis, K., Roosta-Khorasani, F., Shun, J., Cheng, X., \& Mahoney, M. W. (2019). Variational perspective on local graph clustering. Mathematical Programming, 174, 553-573.
>
> - [2] Bojchevski, Aleksandar, Johannes Gasteiger, Bryan Perozzi, Amol Kapoor, Martin Blais, Benedek Rózemberczki, Michal Lukasik, and Stephan Günnemann. "Scaling graph neural networks with approximate PageRank." In Proceedings of the 26th ACM SIGKDD International Conference on Knowledge Discovery \& Data Mining, pp. 2464-2473. 2020.

---

> > ### Comment · Reviewer_oXbQ · 2024-08-12
> > **thank you**
> >
> > Thank you for the detailed comments / clarifications.
> >
> > It's good to know the \epsilon regime being considered here. However, my worry is that in such parameter regimes, non-local things with some dependencies on n might start winning. This is, of course, beyond the scope of this work, or even this discussion. So I'll keep my review/score unchanged.

---

### Official Review · Reviewer_cyZp · 2024-07-13

**Soundness:** 3
**Presentation:** 3
**Contribution:** 3
**Rating:** 7
**Confidence:** 2

**Summary:**

This paper considers the approximate personalized page rank. Classical results for this problem have a runtime that is linear in $1/\alpha\epsilon$ where $\alpha$ is the damping factor and $\epsilon$ is the error parameter. The authors show that APPR is simply a local variant of Gauss-Seidel Successive Overrelaxation. Using this connection, the authors derive new run time bounds for APPR and also propose a new algorithm based on Gradient Descent. The execution time for both these are, in the worst-case, identical to the previous bounds. However, they are more sensitive to the state of execution of the algorithms (depend on the active nodes) and seem to mirror the actual performance of these algorithms. Also, under certain assumptions, they improve the worst-case execution time.

**Strengths:**

The paper addresses an important problem, provides deeper insights into an existing algorithm, provides a new algorithm and also reanalyzes the algorithm in a more fine-grained way. All of this is done via connection to GSSOR which seems to be new.

I find the result quite interesting. However, I am not very familiar with recent work on personalized page rank.  For this reason, I recommend accepting but with a low confidence.

**Weaknesses:**

NA

**Questions:**

Can you explain how weak/strong are the assumptions that you make to achieve the improvement for the local Chebyshev method? I couldn't quite gauge the usefulness of this result.

---

> ### Author Rebuttal · Authors · 2024-08-06
>
> Thank you for taking the time and effort to review our paper carefully. Your positive perspective on our work is inspiring. The main concern on the assumption we made is addressed as follows:
>
> **Q1.** Comments on the assumptions we made in analyzing the local Chebyshev method.
>
> **A:** (Ignore this answer if you read the general response section). Let us recall that our Theorem C.6 (Standard CH) provides a runtime bound for the standard Chebyshev method. The total number of operations is bounded by
>
> $$
> \mathcal{T}_{\mathrm{CH}} \leq m \cdot \left\lceil\frac{1+\sqrt{\alpha}}{2 \sqrt{\alpha}} \ln \frac{2}{\epsilon}\right\rceil,
> $$
>
> where $m$ is the number of edges. Our key lemma, Lemma 4.1 (Line 246), captures the evolving process of the local Chebyshev method. By considering the geometric mean reduction on $\beta_t$ and using Lemma 4.1, LocCH has the following runtime bound:
>
> $$
> \mathcal{T} _{\text{LocCH}} \leq \overline{\operatorname{vol}}({\mathcal S } _T) \cdot \left\lceil\frac{1+\sqrt{\alpha}}{(2-c) \sqrt{\alpha}} \ln \frac{2 y_T}{\epsilon} \right\rceil
> $$
>
>
> where $\overline{\operatorname{vol}} ({\mathcal S } _T)$ is the averaged volume and $y_T$ is defined by $y _ {t+1}-2 y _t+y _{t-1} / ( (1+\beta _{t-1}) (1+\beta _t ) )=0$. Ideally, when $\epsilon \rightarrow 0$, it roughly indicates the local solver is getting closer to the global one. This leads to $y_T \rightarrow 1$, $\overline{\operatorname{vol}} ({\mathcal S } _T) \rightarrow m$, and $c \rightarrow 0$ where $c$ is the parameter defined via the geometric mean of $\beta_t$. However, verifying how strong or weak this assumption is can be difficult. The main reason is the convergence of the second-order difference equation itself is complicated and makes the analysis harder. We are investigating and exploring alternative directions now. A more reasonable approach might be to consider a typical example on a chain graph, where one can obtain a much simpler formulation for $y_t$. We will develop a refined analysis for specific graph types in future work.
>
> We are happy to have further discussions if needed.

---

### Author Rebuttal · Authors · 2024-08-06

**General Responses**

We thank all reviewers for their time and effort in carefully reading our paper. We are very happy that you like our work. Some general concerns are worth discussing as follows:

---

**Q1.** Comments on the assumption of the local Chebyshev (LocCH) method.

**A:** Let us recall that our Theorem C.6 (Standard CH) provides a runtime bound for the standard Chebyshev method. The total number of operations is bounded by

$$
\mathcal{T}_{\mathrm{CH}} \leq m \cdot \left\lceil\frac{1+\sqrt{\alpha}}{2 \sqrt{\alpha}} \ln \frac{2}{\epsilon}\right\rceil,
$$

where $m$ is the number of edges. Our key lemma, Lemma 4.1 (Line 246), captures the evolving process of the local Chebyshev method. By considering the geometric mean reduction on $\beta_t$ and using Lemma 4.1, LocCH has the following runtime bound:

$$
\mathcal{T} _{\text{LocCH}} \leq \overline{\operatorname{vol}}({\mathcal S } _T) \cdot \left\lceil\frac{1+\sqrt{\alpha}}{(2-c) \sqrt{\alpha}} \ln \frac{2 y_T}{\epsilon} \right\rceil
$$


where $\overline{\operatorname{vol}} ({\mathcal S } _T)$ is the averaged volume and $y_T$ is defined by $y _ {t+1}-2 y _t+y _{t-1} / ( (1+\beta _{t-1}) (1+\beta _t ) )=0$. Ideally, when $\epsilon \rightarrow 0$, it roughly indicates the local solver is getting closer to the global one. This leads to $y_T \rightarrow 1$, $\overline{\operatorname{vol}} ({\mathcal S } _T) \rightarrow m$, and $c \rightarrow 0$ where $c$ is the parameter defined via the geometric mean of $\beta_t$. However, verifying how strong or weak this assumption is can be difficult. The main reason is the convergence of the second-order difference equation itself is complicated and makes the analysis harder. We are investigating and exploring alternative directions now. A more reasonable approach might be to consider a typical example on a chain graph, where one can obtain a much simpler formulation for $y_t$. We will develop a refined analysis for specific graph types in future work.

---

**Q2.** Comments on the parameter regimes in which local algorithms get used.


**A:** The range of $\epsilon$ that is useful in real-world applications depends on the specific task. However, as far as we know, the speedup over standard solvers can be applied to a wide range of downstream tasks. Let's look at typical examples to see when $\epsilon$ is sufficient or useful in practice. For the local clustering algorithm, in Fountoulakis's paper (see Fountoulakis's work in [1]), they used $\epsilon = 10^{-5}$ where the number of nodes in graphs is in the range $[2\times 10^6, 3\times 10^6]$. This is around $\epsilon \approx 1/n$ (roughly corresponding to the largest speedup shown in Figure 4, where the dashed vertical lines are $\epsilon = 1/n$). For learning GNN models, PPRs are used to train the PPRGo model (see Bojchevski's work in [2]); the parameter $\epsilon$ used is $10^{-4}$ where the number of nodes in graphs is in the range $[1.87\times 10^4, 1.05 \times 10^7]$. This means $\epsilon \leq 1/n$ in all graphs. We found similar parameter settings in graph embedding and online graph node classification.

Overall, the effective range of $\epsilon$ is $\epsilon \leq 10^{-4} / n$, where the speedup is significant, as presented in Fig. 4. This should cover most interesting downstream tasks where PPR is a crucial tool.

---

**Q3.** Is the dependence on $\epsilon$ optimal?

**A:** Let us recall that the runtime bound is optimal for the standard Chebyshev method. Specifically, the *first-order optimal methods (such as Chebyshev) for linear systems* have the runtime bound $\mathcal{O}(m/\sqrt{\alpha} \cdot \log 1/\epsilon)$. This runtime bound is optimal for finding a solution of a sparse linear system defined on a chain graph (see Theorem 3.15 of Bubeck's work in [3] for more details).

We believe this is true in terms of the **optimal local first-order methods** for local solvers. However, it remains interesting to see whether the lower bound can be identified, where one may find that it matches the bound conjectured in Line 295 of our manuscript.


---
**References**

- [1] Fountoulakis, K., Roosta-Khorasani, F., Shun, J., Cheng, X., & Mahoney, M. W. (2019). Variational perspective on local graph clustering. Mathematical Programming, 174, 553-573.

- [2] Bojchevski, Aleksandar, Johannes Gasteiger, Bryan Perozzi, Amol Kapoor, Martin Blais, Benedek Rózemberczki, Michal Lukasik, and Stephan Günnemann. "Scaling graph neural networks with approximate pagerank." In Proceedings of the 26th ACM SIGKDD International Conference on Knowledge Discovery \& Data Mining, pp. 2464-2473. 2020.

- [3] Sébastien Bubeck, Convex Optimization: Algorithms and
Complexity, https://arxiv.org/pdf/1405.4980.

---

### Author Response · Authors · 2024-08-14
**Thank all reviewers for their time and effort in the discussion and for helping us.**

We sincerely thank all reviewers for their time and effort in the discussion and for helping us; we will keep improving our submission.

---

### Decision · Program_Chairs · 2024-09-25

**Decision:**

Accept (poster)

**Comment:**

This paper introduces a novel approach to efficiently compute approximate personalized PageRank (and more generally for sparse linear systems) based on a locally evolving set process. It further develops localized versions of chebyshev and heavy ball methods that achieve a $1/\sqrt{\alpha}$ dependence in the number of iterations necessary for approximating pagerank with dampening factor $\alpha$, in contrast with the $1/\alpha$ dependence for the classic algorithm by Andersen-Chung-Lang. This dependence is optimal, and the authors also present simulations that suggest an empirical advantage for small enough target accuracy.

The reviewers were positive about the paper, highlighting the novel technical contributions. There were some concerns about the empirical advantage being realized only for very small approximations, but I think this is an overall solid contribution.